# Social identity correlates of social media engagement before and after the 2022 Russian invasion of Ukraine

Yara Kyrychenko [1] ✉, Tymofii Brik [2], Sander van der Linden [1] &
Jon Roozenbeek [1,3] ✉

Despite the global presence of social media platforms, the reasons why people like and share content are still poorly understood. We investigate how group identity mentions and expressions of ingroup solidarity and outgroup hostility in posts correlate with engagement on Ukrainian social media (i.e., shares, likes, and other reactions) before and after the 2022 Russian invasion of Ukraine. We use a dataset of 1.6 million posts from Ukrainian news source pages on Facebook and Twitter (currently X) and a geolocated sample of 149 thousand Ukrainian tweets. Before the 2022 Russian invasion, we observe that outgroup mentions in posts from news source pages are generally more strongly associated with engagement than negative, positive, and moral-emotional language. After the invasion, social identity mentions become less strongly associated with engagement. Moreover, post-invasion ingroup solidarity posts are strongly related to engagement, whereas posts expressing outgroup hostility show smaller associations. This is the case for both news and non-news social media data. Our correlational results suggest that signaling solidarity with one's ingroup online is associated with more engagement than negativity about outgroups during intense periods of intergroup conflicts, at least in the context of the Russian-Ukrainian war.

With social media platforms projected to connect roughly 70% of people worldwide by 2027[1,2], what content gains traction on social media has real-world implications across domains ranging from politics[3] to healthcare[4]. These implications can be especially high-stakes during intense intergroup conflicts such as highly polarized elections or wars. Previous studies conducted mainly in the US have identified negative emotions, moral outrage, and group identity as the primary correlates of social media engagement[5–7]. In this paper, we extend these findings to a non-WEIRD context of severe intergroup conflict by investigating whether content expressing ingroup solidarity and outgroup hostility is associated with increased engagement six months before and after the 2022 Russian invasion of Ukraine.

Social media platforms run algorithms that utilize user signals such as shares, likes, and other reactions, known as engagement

metrics, to show users content that maximizes these metrics[8]. The interplay between algorithmic content recommendation and human behavior makes social media a complex system with emergent properties[8,9]. Although social media platforms limit access to their recommender algorithms, researchers can investigate what characteristics of human behavior (e.g., the linguistic patterns in user posts) tend to generate more engagement. For instance, a 2021 study by Rathje et al.[7] found that mentions of the opposite political party in the United States (e.g., Democratic politicians talking about Republicans and vice versa) were the strongest correlates of engagement with political content on Facebook and Twitter (currently X), suggesting that social identity cues may underlie what gains popularity on social media. Social identity theory (SIT)[10] postulates that people derive part of their self-identity from salient memberships in social

[1]Department of Psychology, University of Cambridge, Cambridge, UK. [2]Kyiv School of Economics, Kyiv, Ukraine. [3]Department of War Studies, King's College London, London, UK. ✉e-mail: yk408@cam.ac.uk; jjr51@cam.ac.uk

groups and start comparing themselves (i.e., the ingroup) to others (i.e., the outgroup), which makes SIT helpful in explaining intergroup conflict and affective polarization[11].

Although SIT has been studied for many decades[12], less is known about the correlates of social identity on digital and social media. Some studies suggest that social media might exacerbate political polarization by, for example, making social identity more salient[13–15]. However, the influence of specific social identity processes online appears to hinge on the intergroup context, as indicated by varying polarization effects observed in social media studies across different countries[16–19]. Moreover, research about the role of media and polarization in non-WEIRD (i.e., Western, Educated, Industrialized, Rich, and Democratic[20]) contexts is lacking, with as much as 86% of studies coming from the US[21]. In the US context, negative outgroup cues seem more effective than positive ingroup cues in persuading partisans[7,22,23]. More generally, major conflicts such as World War II have usually been explained by focusing on outgroup derogation and dehumanization[24,25]. Nevertheless, some scholars of social identity theory have theorized that ingroup-favoring motivations should matter more than outgroup derogation[26–28]. Indeed, many studies suggest that ingroup solidarity might be more relevant to the group than outgroup hostility (broadly defined) after severe shocks or in the presence of outgroup threats[29–34]. For instance, after the November 2015 Paris terrorist attacks, one study[31] found no evidence of an increase in opinions indicating outgroup hostility but instead observed a higher level of views consistent with ingroup solidarity among French citizens. Of course, social media behavior is not only a reflection of group psychology but also of social media algorithms.

Nonetheless, collectively, these findings raise questions about the conditions under which outgroup hostility (or outgroup hate) and ingroup solidarity (or ingroup love)[35] gain traction on social media. Given the key role of emotions in shaping the diffusion of (news) content on social media[5,6], our investigation draws on insights from intergroup emotions theory (IET)[35,36]. This theory extends SIT into the realm of (intergroup) emotions, suggesting that salient group identifications can influence emotions whereby events (such as a violent conflict) are appraised regarding their implications for the ingroup and outgroup. IET highlights the multidirectionality of causal processes such that emotions can influence intergroup behavior, but intergroup behaviors can, in turn, influence the expression of identity and emotions[35]. Notably, although IET itself does not specify whether ingroup solidarity is more engaging than outgroup hostility during any particular stage of intergroup conflict, its proponents do suggest to evaluate further the hypothesis that ingroup positivity can foster conflict processes more than outgroup negativity[35, p. 27]. In fact, few studies have conceptually distinguished ingroup solidarity from outgroup hostility to better understand their unique impacts during intergroup conflicts[7,35]. In our work, we study whether social media posts gain more engagement if they express ingroup solidarity and outgroup hostility after the 2022 full-scale invasion of Ukraine to explore the broader theoretical prediction that ingroup solidarity can be associated with more engagement than outgroup hostility[26,28,35] and to advance research on the correlates of social media engagement in non-WEIRD contexts.

In late 2013 and early 2014, Ukraine's Euromaidan revolution ousted the pro-Russian then-president Viktor Yanukovych, and the Ukrainian peninsula of Crimea was annexed by the Russian Federation shortly afterward. Paramilitary groups in the eastern region of Donbas declared independence from Ukraine and began a Russia-supported insurgency against the Ukrainian state[37]. Although not frozen in terms of military activity, the conflict remained mostly at a stalemate until 24 February 2022, when Russia launched a full-scale military invasion of Ukraine. This war has so far cost hundreds of thousands of lives and continues into the present day, with Russia gaining and then losing control of swaths of territory in Southern and Eastern Ukraine. The war

is also being fought in the information space, as Russia considers social media part of the broader (kinetic and informational) battlefield under the Gerasimov Doctrine[37].

A large body of research has studied Ukrainian ethnic, civic, and linguistic identities and their interplay with each other and with pro-Russian identity, i.e., support for and attachment to Russia[37–44]. After the Euromaidan revolution, Ukrainian identity shifted to become more anti-Russian in all regions of Ukraine except for the eastern region of Donbas, where people began to identify more with regional and pro-Russian identities[39]. As Fig. 1 illustrates, after the 2022 invasion, as few as 2% of Ukrainians and 23% of Russians expressed a positive attitude toward the other country, a dramatic decrease from 83% and 74% positive opinions just a decade earlier[45–47], indicating increased intergroup tensions and polarization. The 2022 Russia-Ukraine war is, thus, a high-profile example of affective polarization culminating in the outbreak of a full-scale war, fully captured on social media.

In this work, we investigate social media engagement (operationalized as the sum of all platform-specific reactions) with expressions of identity mentions (Studies 1 and 2) and ingroup solidarity and outgroup hostility (Studies 2 and 3) on Facebook and Twitter before and after the 2022 invasion. Given the complicated historical and socio-political underpinnings of the Russia-Ukraine war[37], we do not have specific a priori hypotheses about the primacy of outgroup hostility or ingroup solidarity either before or after the full-scale invasion (with the exception of our replication of Rathje et al., 2021, i.e., Study 1[7]). Rather, we view this work as a natural exploration of how engagement with expressions of intergroup emotions covaries with different stages of a major intergroup conflict. Our results suggest that, in the context of the Russia-Ukraine war, ingroup solidarity becomes the strongest factor associated with engagement on Ukrainian social media after Russia's invasion in February 2022, compared to outgroup hostility. These effects are stronger among Facebook and Twitter posts by news sources but also present in Twitter posts by regular social media users geolocated to Ukraine. Our findings add nuance to the ongoing scientific debate about why content goes viral.

## Results
### Study 1: news sources before the invasion
Study 1 is a generalization of Rathje et al.[7] to the context of pro-Ukrainian and pro-Russian news sources in Ukraine before the 2022 invasion. This study aims to test whether the correlates of engagement on US social media identified by Rathje et al.[7] (i.e., mentions of ingroup and outgroup identity) hold up in a very different affectively polarized national context. Like Rathje et al.[7], we collected data from Facebook and Twitter. Facebook is the most popular social media platform in Ukraine, while Twitter ranks fifth, behind YouTube, Instagram, and Telegram[48]. We collected posts by the most popular pro-Ukrainian and pro-Russian news sources in Ukraine posted between 12 July 2021, and 24 February 2022, in Ukrainian or Russian (see Methods and Supplementary Information for details and the temporal distribution of the data as not all Twitter posts were collected). Following the original study, we classified a post as pro-Ukrainian (or pro-Russian) if it came from a pro-Ukrainian (or pro-Russian) news source and counted how many words in each post referred to Ukrainian and Russian identity and used negative, positive, or moral-emotional language[6]. We created dictionaries with mentions of Ukrainian and Russian identity that contained matter-of-fact references to the two nations: country name, capital, currency, decision-making center (i.e., Bankova and Kremlin), the 10 largest cities and ten most well-known politicians, and their language-specific morphological derivations. We used previously validated affective dictionaries (negative and positive language[49,50] and an existing dictionary of moral emotions[6], which we translated into Ukrainian and Russian). Finally, we fit mixed effects linear regressions predicting log-transformed engagement (i.e., the log of the sum of all platform-specific reactions + 1) based on mentions of the ingroup and

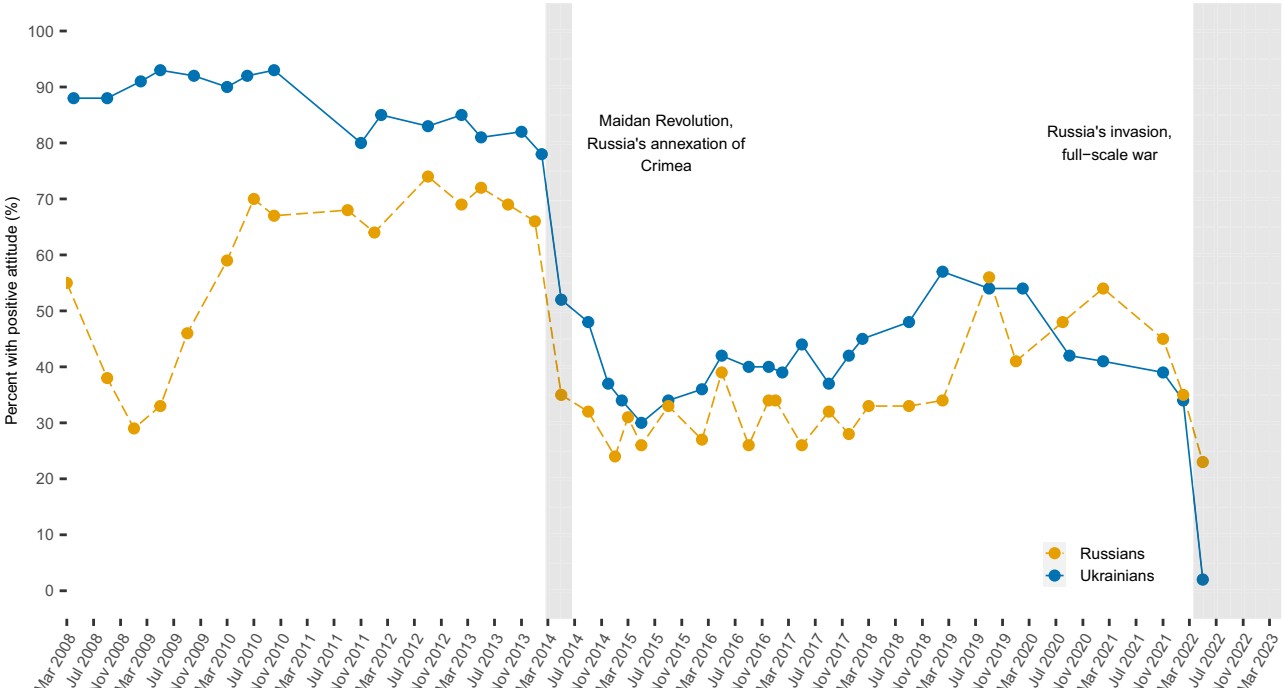

**Fig. 1 | Timeline of Ukrainians' positive attitudes to Russia and Russians' positive attitudes to Ukraine.** The opinions of Russians and Ukrainians about the other country have worsened dramatically over the last decade. In Nov 2013, 82% of Ukrainians viewed Russia positively. This dropped to about 52% after the Maidan revolution and Russia's annexation of Crimea. Opinions remained relatively stable, hovering around 40–45%, until after Russia's invasion in February 2022 when approval dropped to about 2%. A similar pattern can be seen for Russians' opinions about Ukraine. Source: Kyiv International Institute of Sociology and Levada-Center[45–47].

outgroup and negative, positive, and moral-emotional language, with news source as the random effect. We controlled for the follower count, URL, and media attachments, word count, and whether a tweet was a retweet. Like the original study, we hypothesized that mentions of the outgroup would have the strongest association with engagement across platforms ($N = 468,310$ and $N = 114,557$ posts, and $N = 182,053$ and $N = 46,705$ posts for pro-Ukrainian and pro-Russian Facebook, and pro-Ukrainian and pro-Russian Twitter, respectively). Across the three studies, statistical tests are two-tailed and the $p$-values and Cohen's $d$ are calculated using Satterthwaite d.f. Our results are robust to dictionary variations (Supplementary Table 23).

Similar to Rathje et al.[7], the use of outgroup mentions in a social media post was the strongest predictor of engagement compared to ingroup mentions and emotional language in all data sets, except for pro-Russian Twitter. Controlling for all other factors, each additional outgroup word increased engagement by 4% to 23%. Ingroup language was associated with a 4% to 16% increase in engagement. Outgroup mentions predicted more engagement than ingroup mentions in three out of four datasets, with the exception of pro-Russian Twitter. Also replicating previous work[6,7,13], moral-emotional language significantly increased engagement by 3% to 6% across data sets. Positive and negative language predicted a 2% to 5% increase in engagement in all datasets, with the exception of negative language on pro-Russian Twitter, where we observed no effect. Our findings are broadly in line with the results reported by Rathje et al.[7]: using terms referring to the outgroup was strongly predictive of engagement on Ukrainian Facebook and Twitter, more so than ingroup terms and emotional language categories, before the 2022 invasion. A visual representation of the results can be found in Fig. 2, and complete regression tables are available in Supplementary Table 1.

## Study 2: pro-Ukrainian news sources after the invasion
In Study 2, we explored how the correlates of social media engagement vary with time and during critical events – in this case, the outbreak of

a full-scale war. Specifically, we investigated (1) whether the correlates of engagement identified by Rathje et al.[7], i.e., ingroup and outgroup mentions, become more or less strongly associated and (2) whether ingroup solidarity or outgroup hostility[35] was associated with more engagement after the 2022 invasion (25 February 2022 - 13 September 2022; overall, $N = 1,011,171$ posts and $N = 399,555$ posts for Facebook and Twitter, respectively). We only studied pro-Ukrainian data, as Facebook and Twitter were banned in Russia shortly after the invasion, making the data limited and unlikely to be representative of authentic engagement patterns[51].

After the start of the 2022 invasion, Ukrainian and Russian identity mentions became less strongly associated with engagement. Controlling for all else, outgroup mentions dropped from predicting a 16% increase, $\exp(\beta) = 1.16$, $t(466964) = 63.89$, $p < 0.001$, $d = 0.19$, 95% CI=[1.15, 1.17], to 7%, $\exp(\beta) = 1.07$, $t(535719) = 41.71$, $p < 0.001$, $d = 0.11$, 95% CI = [1.07, 1.08], on Facebook and from a 23% increase, $\exp(\beta) = 1.23$, $t(182021) = 44.93$, $p < 0.001$, $d = 0.21$, CI=[1.22, 1.24]), to 6%, $\exp(\beta) = 1.056$, $t(217172) = 17.36$, $p < 0.001$, $d = 0.08$, 95% CI = [1.05, 1.06], on Twitter (per each additional word and controlling for all other variables). Similarly, the effects of ingroup terms dropped slightly from an 11% increase, $\exp(\beta) = 1.11$, $t(468248) = 62.27$, $p < 0.001$, $d = 0.18$, 95% CI = [1.11, 1.12], to 4%, $\exp(\beta) = 1.04$, $t(535714) = 26.27$, $p < 0.001$, $d = 0.07$, 95% CI = [1.04, 1.05], on Facebook and from a 16% increase, $\exp(\beta) = 1.16$, $t(182032) = 46.66$, $p < 0.001$, $d = 0.22$, 95% CI = [1.15, 1.16], to 11%, $\exp(\beta) = 1.11$, $t(217180) = 35.99$, $p < 0.001$, $d = 0.15$, 95% CI=[1.11, 1.12], on Twitter. Facebook and Twitter posts were likely to gain 4–5% more engagement for each additional moral emotional word and 3-7% for each additional positive word. Negative words, however, had no significant effect on both social media platforms after the invasion. See Fig. 3a for a visual depiction and Supplementary Table 2 for all coefficients.

Next, we explored what may have replaced the group identity mentions as a predictor of engagement after the invasion. We first manually inspected a series of Facebook posts with high amounts of

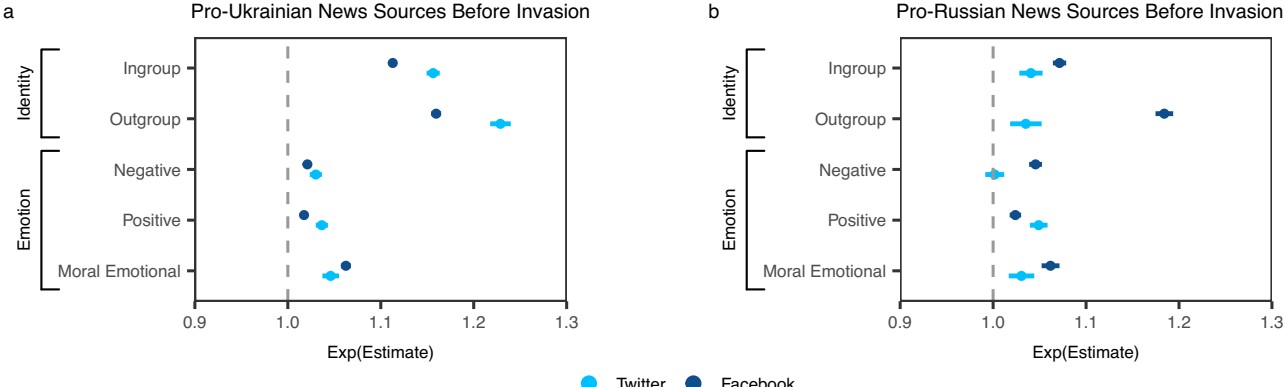

**Fig. 2 | Identity and emotional words as predictors of engagement before the 2022 invasion.** Before Russia's 2022 invasion of Ukraine, mentions of the outgroup were the strongest predictor of engagement on Ukrainian social media, as compared to ingroup mentions and negative, positive, and moral emotional words. The only exception was pro-Russian Twitter, where outgroup mentions had the same effect as ingroup language. **a** Predictors of engagement with posts by pro-Ukrainian news sources for each additional word in the corresponding dictionary ($N = 468{,}310$ Facebook posts; $N = 182{,}053$ tweets). **b** Predictors of engagement with posts by pro-Russian news sources for each additional word in the corresponding dictionary ($N = 114{,}557$ Facebook posts; $N = 46{,}705$ tweets). Data presented as $\exp(\beta)$ estimates with error bars representing 95% CIs. See Results Study 1 and Supporting Table 1 for full information.

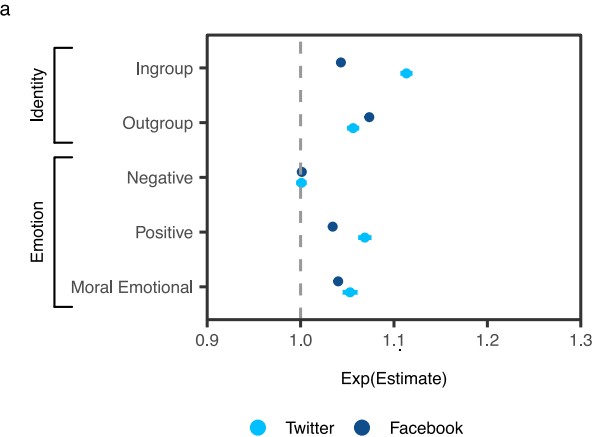

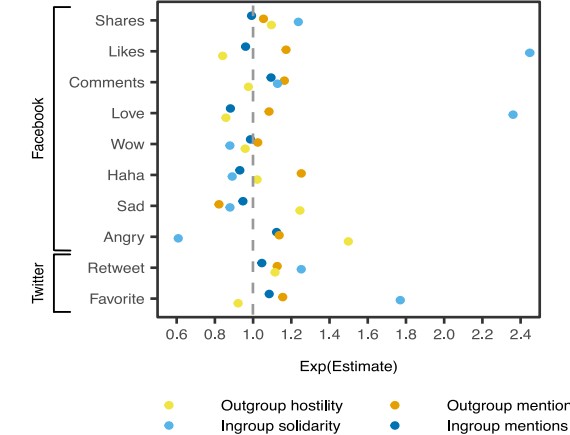

**Fig. 3 | Identity and emotional words as predictors of engagement after the 2022 invasion. a** Ingroup and outgroup mentions and negative, positive, and moral emotional language as predictors of engagement after the invasion on pro-Ukrainian Facebook and Twitter. **b** Regression coefficients for each of the binary categories of group identity language predicting the corresponding platform-specific reaction after the invasion on pro-Ukrainian Facebook and Twitter. Data presented as $\exp(\beta)$ estimates with error bars representing 95% CIs; both figures are based on $N = 535{,}797$ Facebook posts and $N = 217{,}245$ tweets. See Results Study 2 and Supporting Tables 1, 23, and 25 for full information.

engagement to see whether there were any obvious patterns in terms of what went viral. Based on this and drawing on recommendations from intergroup emotions theory to conceptually and empirically distinguish ingroup love from outgroup hate[35], we created two classifiers of emotionally-charged group identity: ingroup solidarity and outgroup hostility (note that these two are not mutually exclusive with group mentions)[32,35,52].

We trained two binary classifiers of ingroup solidarity and outgroup hostility (see Table 1 for post examples)[35]. We operationalized ingroup solidarity in this context as expressing solidarity, liking, or unity of Ukraine or Ukrainians; for example, if a post praises Ukraine, mentions Ukrainians as competent, good people, or Ukraine as a great, strong, or united nation. Outgroup hostility was operationalized as expressing hostility, derogation, or dislike of Russia or Russians; for example, if a post criticizes Russia, mentions Russians as incompetent, immoral people, or the Russian Federation as a bad, weak, or failing nation. To create the models, the first author manually labeled 2000 posts from the datasets and used 1600 of them for training (see Supplementary Information for the code

book). We then fine-tuned ingroup solidarity and outgroup hostility multilingual Natural Language Inference DeBERTa models (BERT-NLI models) following Laurer et al.[53], who showed that inference models require much less labeled data to achieve good classification results compared to traditional fine-tuning of BERT-style models and other supervised machine learning. Another advantage of this method is that we were able to provide the models with explicit definitions of ingroup solidarity and outgroup hostility. Both classifiers attained an accuracy of ~0.87 and an F1-macro of 0.80. We then tested how expressions of intergroup emotions and the binarized ingroup and outgroup mentions predicted engagement. We also conducted the same analysis with dictionaries instead of classifiers, yielding very similar results (see Methods and Supplementary Figs. 6 and 7 and Table 29). Our ingroup solidarity dictionary contained words that positively reference the ingroup (e.g., defenders, heroes), while the outgroup hostility dictionary had words that negatively refer to the outgroup (e.g., occupiers, invaders). We investigated the change in engagement patterns overall and using a sliding window time series approach. For each 14-day interval in the pro-Ukrainian data starting

**Table 1 | Examples of social media posts from news sources with ingroup and outgroup mentions, ingroup solidarity, and outgroup hostility**

| Post type | Original | Translation |
|---|---|---|
| Ingroup mentions (Ukraine) | З кінця лютого до України повернулися майже 2,5 мільона осіб - ООН (UA) | Since the end of February, almost 2.5 million people have returned to Ukraine - UN |
| | Германия передала Украине ещё 4 самоходных зе-нитных установки Gepard – правительство ФРГ (RU) | Germany has provided Ukraine with another 4 self-propelled Gepard anti-aircraft guns - German government |
| Outgroup mentions (Russia) | Ворог намагається вийти на кордони Луганської області [3 Backhand Index Pointing Down Emojis] Російські війська наступають на Лиман та Сєвєродонецьк — Генштаб (UA) | The enemy is trying to reach the borders of Luhansk region [3 Backhand Index Pointing Down Emojis] Russian troops are advancing on Liman and Severodonetsk—General Staff |
| | Ситуация по российскому вторжению #stoprussia Российские окку-пационные войска продолжают перегруппировку с целью возобновления наступа-тельных действий в направлении г.Бровары Ки-евской области (RU) | The situation with the Russian invasion #stoprussia. Russian occupying troops continue to regroup in order to resume offensive operations in the direction of the city of Brovary, Kyiv region. |
| Ingroup solidarity (pro-Ukrainian emotions) | Ви просто подивіться на це [3 Surprise Emojis] ЗСУ - герої... Немає слів... (UA) | Just look at this [3 Surprise Emojis] the ZSU (Ukrainian Armed Forces) are heroes... There are no words... |
| | Девочки, тачки и бабки - это футбол по-украински (RU) | Girls, cars and money: this is Ukrainian football |
| Outgroup hostility (anti-Russian emotions) | Що, російська мерзото, "Херсонська народна ре-спубліка"не вилу-пилася? (UA) | What, Russian scumbags, it didn't work out with the "Kherson People's Republic"? |
| | Черепахе спасибо за четкую позицию! Огромнаячерепаха покусала россиянку в Турции (RU) | Thank you, turtle, for taking such a clear position! A huge turtle bit a Russian woman in Turkey |

These categories are not mutually exclusive; e.g., a post can be classified as mentioning ingroup and as ingroup solidarity if it explicitly mentions Ukraine and expresses solidarity with Ukrainians. Posts are presented in the original language (UA for Ukrainian and RU for Russian) and translated into English.

on the first day in our sample (i.e., between 12 July 2021 and 13 September 2022), we fit a mixed-effects linear regression predicting log-transformed engagement based on the binary categories of expressing ingroup solidarity, expressing outgroup hostility, binarized mentions of Ukrainian and Russian identity (i.e., binary indicators of whether a post contained a word referring to a group from Study 1), and negative, positive, and moral emotional word counts and control variables. We plotted the resulting time series of estimates, shifting it by half the window length (seven days) to accurately capture the timing of the effects.

Before the invasion, ingroup solidarity was associated with more engagement than outgroup hostility (Fig. 4). On both Facebook and Twitter, outgroup hostility briefly increased right before the invasion and dropped back down afterward. Ingroup solidarity, however, remained a strong predictor after the start of the invasion, hovering around a 95% increase on Facebook and a 65% increase on Twitter (Supplementary Fig. 2). Fitting one regression model to all of the pro-Ukrainian data from after the invasion, we find that, controlling for all else, posts that were classified as ingroup solidarity were likely to gain 92% more engagement, $\exp(\beta) = 1.92$, $t(533141) = 151.95$, $p < 0.001$, $d = 0.42$, 95% CI = [1.91, 1.94], on Facebook and 68% more engagement, $\exp(\beta) = 1.68$, $t(214803) = 96.31$, $p < 0.001$, $d = 0.42$, 95% CI = [1.67, 1.70]) on Twitter overall (see Supplementary Table 3; Fig. 3b depicts how these language categories are associated with specific reactions). Meanwhile, outgroup hostility was likely to increase engagement by just 1%, $\exp(\beta) = 1.01$, $t(533140) = 3.32$, $p = 0.001$, $d = 0.01$, 95% CI = [1.01, 1.02]), on Facebook and had no statistically significant effect on Twitter, $\exp(\beta) = 0.99$, $t(214803) = -1.11$, $p = 0.265$, $d = 0.01$, 95% CI=[0.98, 1.01]). In other words, ingroup solidarity became the strongest correlate of social media engagement with news-related content on both Facebook and Twitter after the invasion and remained so for at least half a year. Descriptively, ingroup solidarity and outgroup hostility made up around 15% and 10% of content, respectively, on Facebook and 9% and 6% on Twitter before the invasion. After the invasion, the proportions jumped to 31% and 35% on Facebook and 21% and 25% on Twitter, respectively (Supplementary Tables 5 and 6).

**Study 3: geolocated pro-Ukrainian Twitter after the invasion**
In Study 3, we tested whether the post-invasion engagement patterns from Study 2 generalize to non-news social media data. We collected a dataset of original, non-replies tweets geolocated to Ukraine spanning July 2021 to September 2022. Unlike the previous two studies, where we used the orientation of the account of origin to classify posts into pro-Ukrainian or pro-Russian, in this study we used a fine-tuned RoBERTa model trained to classify posts into pro-Ukrainian or pro-Russian point of view about the war on 30 thousand manually labeled examples from seven social media platforms, including Twitter, achieving an accuracy of 0.86 and F1-macro of 0.81. Here, pro-Ukrainian does not mean positive about Ukraine or Ukrainians, but rather can mean any expression in support of Ukraine or opposition of Russia, including expressions that are negative about Russia or Russians (e.g., a sentence such as I hate Russia, for instance, would also be classified as pro-Ukrainian by this particular classifier). We then only analyzed the posts identified as pro-Ukrainian and posted after the invasion ($N = 148,959$ tweets). As the RoBERTa classifier was trained and evaluated exclusively on the data from after the invasion and thus does not necessarily generalize to the prior period, we cannot make any claims for the period before the invasion, but we provide this data for completeness (Supplementary Tables 4 and 8). We further fine-tuned our ingroup solidarity and outgroup hostility Study 2 BERT-NLI models on 1000 posts from pro-Ukrainian geolocated Twitter, achieving an F1-macro of 0.75 and 0.82, respectively. We fit a mixed effects linear regression predicting log-transformed engagement based on the same variables as in Study 2 (excluding whether a post had media attachments, which was not available) and whether the user was verified. Follower count and verified information were missing for some of the accounts in our data, but subsetting to only the data where we have full user information produces largely the same results (see Supplementary Table 22). This study is not intended to replicate the findings regarding changes from pre-invasion to post-invasion, but only the engagement patterns post-invasion.

We find that the post-invasion results from Study 2 are replicated in a non-news-specific dataset: after the start of the invasion, posts containing ingroup solidarity were likely to get 14%, $\exp(\beta) = 1.14$, $t(147613) = 14.85$, $p < 0.001$, $d = 0.08$, 95% CI = [1.12, 1.16], more engagement, as compared to 7%, $\exp(\beta) = 1.07$, $t(145862) = 8.01$, $p < 0.001$, $d = 0.04$, 95% CI = [1.05, 1.09], for outgroup hostility, and 4%, $\exp(\beta) = 1.04$, $t(146191) = 4.87$, $p < 0.001$, $d = 0.03$, 95% CI = [1.03, 1.06], and 7%, $\exp(\beta) = 1.07$, $t(147015) = 11.00$, $p < 0.001$, $d = 0.06$, 95% CI = [1.06, 1.09], for the outgroup and ingroup mentions, respectively

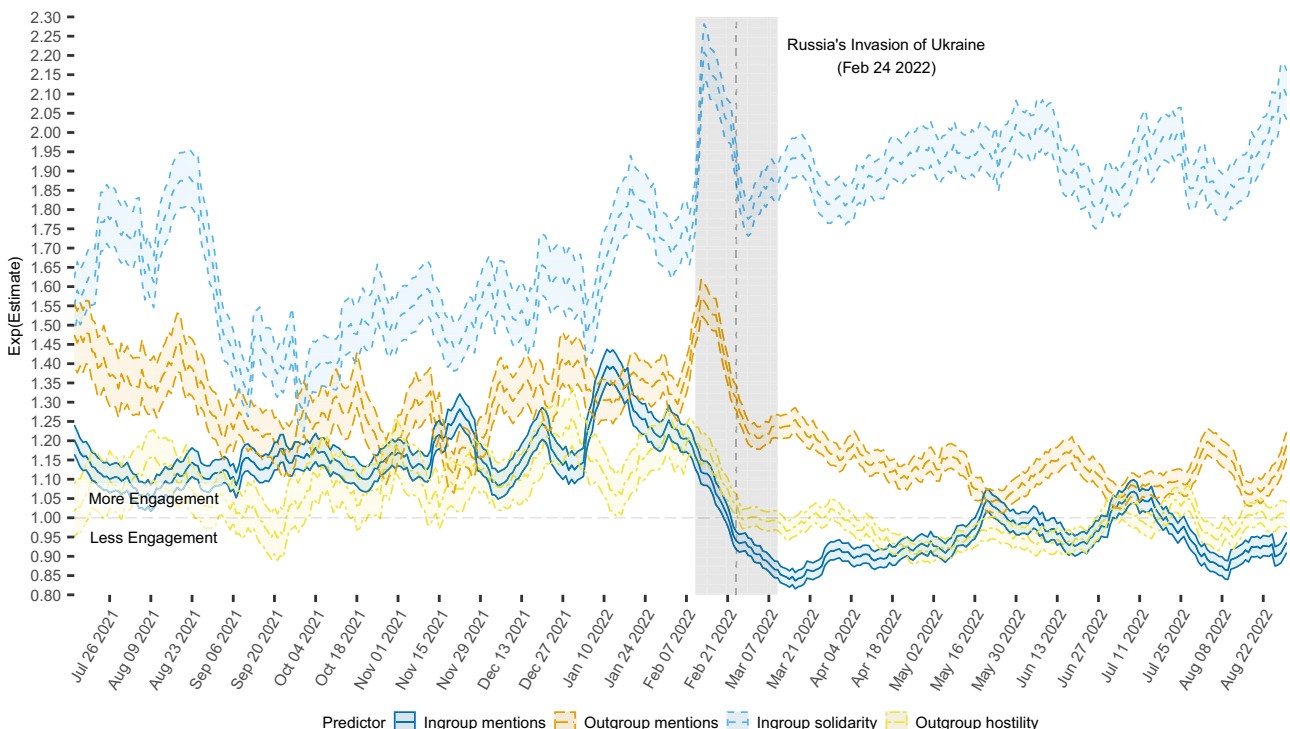

**Fig. 4 | Predictors of engagement over time on Ukrainian Facebook.** Outgroup (Russian) identity mentions consistently predicted more engagement than ingroup (Ukrainian) mentions in our dataset of pro-Ukrainian news media accounts on Facebook before the 2022 invasion. However, after 24 February 2022, outgroup and ingroup mentions started predicting less engagement. Instead, ingroup solidarity emerged as the strongest predictor of engagement by a margin of more than 50%. A similar pattern was seen on Twitter (Supplementary Fig. 2). The dashed vertical line represents the invasion date (24 February 2022), and the content from that day is present in the gray window around it. The peak in ingroup solidarity coefficients around the end of August 2021 coincides with the celebrations of the Ukrainian Independence Day on August 24 – a highly identity-salient event supposed to elicit solidarity. Data presented as time series of exp($\beta$) estimates (central lines) with 95% CIs (shaded regions around the central line); $N = 1,011,171$ Facebook posts.

(see Supplementary Table 4 for all regression coefficients). These results suggest that our findings from Study 2 generalize beyond news content.

## Discussion

In this work, we investigated the factors associated with social media engagement during the lead-up to and the first six months of the 2022 Russian invasion of Ukraine. Drawing on a dataset of about 1.6 million posts from Ukrainian news source pages on Facebook and Twitter and a geolocated sample of 149 thousand Ukrainian tweets, we report several findings. First, in line with Rathje et al.[7], we find that mentions of the outgroup are statistically significantly associated with more social media engagement than ingroup mentions and negative, positive, and moral-emotional language within the context of pre-invasion Ukraine. Consistent with prior US-based research, moral-emotional and positive language were also statistically significantly associated with engagement across all of our news-source data[6,13]. Surprisingly, negative language had no statistically significant effect on engagement among our news-source data after the invasion[5,54]. Second, we find that ingroup solidarity emerged as the strongest and most consistent factor associated with engagement after the full-scale invasion of Ukraine and remained so for at least six months, while outgroup hostility had a weaker association. Specifically, we find that news posts made after the invasion are likely to receive 92% more engagement on Facebook and 68% more engagement on Twitter if they are classified as ingroup solidarity. Meanwhile, outgroup hostility posts receive only 1% more engagement on Facebook and have no statistically significant effect on Twitter. The fact that this pattern holds across platforms and for posts from news sources and arbitrary Twitter users suggests that it might not be solely due to platform algorithm changes but could be an expression of a more general artifact of human behavior or a complex product of the two.

From a methodological perspective, studies like ours that combine social media data and precise natural language models offer external validity that is challenging to obtain with traditional lab experiments and surveys, especially when the participants are involved in active conflict[55]. Although such computational investigations – enabled by the recent advances in artificial intelligence – might help us situate psychological theories within emerging technologies that increasingly encompass our daily lives[56], their interpretation with respect to psychological theories relies on a set of assumptions and has limitations[57] (see below for a discussion).

Nevertheless, our findings have real-world implications, as modern warfare is waged in the digital space as well as on the kinetic battlefield. In particular, Russia has used social media to conduct foreign influence campaigns in Ukraine and the US, at least since the 2014 Crimea annexation[37,58] and the 2016 US elections[59], respectively. In fact, the Russian annexation of Crimea is one of the most significant examples of so-called Hybrid warfare, which involves the skillful use of emerging technologies such as social media[37,58].

We offer several interpretations of our findings. One possible interpretation draws on intergroup emotions theory[35,36], which suggests that one's personal emotional experiences depend on the salient group membership and its associated norms. To advance further research on the topic, Mackie et al.[35] suggested that (1) research should differentiate ingroup solidarity from outgroup hostility both conceptually and empirically so that (2) the emergent hypothesis that ingroup positivity motivates conflict more than outgroup negativity[35,p.27] can be further evaluated (see also refs. 26,28). With regard to our work, users might experience intergroup emotions

through emotional contagion by being exposed to posts that express emotions (e.g., ref. 60), which increases their willingness to perform group affiliative behaviors (e.g., engaging with such posts) and group identification[35,61]. In the context of an intergroup conflict, ingroup solidarity might increase the sense of group identity and willingness to perform affiliative or group-beneficial behaviors more than outgroup hostility, resulting in higher engagement rates with ingroup posts. Indeed, we find correlational evidence that ingroup solidarity is associated with more engagement than outgroup hostility on social media, especially during active conflict. However, we cannot make any claims about the existence or direction of any causal effect, as further experimental studies are required to isolate the underlying processes.

Nonetheless, the increase in engagement with pro-Ukrainian content relative to anti-Russian content coincided with a marked decrease in positive sentiments towards Russians among Ukrainians (Fig. 1). Despite outgroup hostility becoming much more commonplace on Ukrainian Facebook and Twitter after the invasion as compared to before (see Supplementary Fig. 10), people did not engage with it much. Specifically, while mentioning the outgroup is associated with more social media engagement (i.e., group members talking about other groups, but not necessarily in a negative manner) when no outright intergroup conflict is present, outgroup hostility is associated with less social media engagement than other variables tested (especially during intergroup conflict). This contradicts expectations that outgroup hostility is strongly associated with social media engagement based on Rathje et al.[7]. Since the United States is a highly polarized country where polarization is primarily driven by negative partisanship[22,23], our results suggest that the types of identity language that correlate with online engagement may be context-dependent and not universal across cultures[62]. In the Ukrainian context of high polarization (Fig. 1), solidarity is associated with more engagement than hostility on social media, even though negative emotions have been the primary focus of past studies[6,7]. Therefore, our study highlights the need for future research to create social media interventions that independently target ingroup solidarity and outgroup hostility to causally test how each type of content increases or decreases the levels of affective polarization experienced on these platforms.

Although one could argue that during full-scale war, outgroup derogation and dehumanization of victims would seem more common, especially among the aggressor (i.e., Russia), it is possible that these dynamics differ pre- and post-invasion. However, unfortunately, we were not able to examine Russian Facebook and Twitter data as they were banned in Russia, but we encourage future research to build on our work, when possible, and compare correlates of engagement with Russian social media content (e.g., on Telegram or VKontakte). In Ukraine, we find that ingroup solidarity has the strongest association with Facebook and Twitter engagement during a violent intergroup conflict. We note that ingroup solidarity became more strongly associated with engagement post-invasion, as compared to pre-invasion, potentially signaling a further consolidation of Ukrainian national identity. This is in line with other research showing that the 2014 Euromaidan revolution boosted Ukrainian national (civic) identity[37,39,40]. High ingroup solidarity and support for the Ukrainian president Zelensky[63] is also consistent with the rally-round-the-flag effect found in the US literature, where presidential approval ratings soar in response to foreign threats[33,64].

On the other hand, it is possible that especially hostile content is algorithmically discouraged by social media platforms[9], or has become so over time. But if this is true, then this would have to be the case for both Facebook and Twitter to a similar extent, at least in the context of our study. Nonetheless, even though our study explored the correlates of social media engagement over time, making the effects less dependent on specific news events, we cannot know if and when the underlying algorithms have changed and how that influenced the results. For instance, a recent experimental study suggests that

Facebook's algorithmic feed might have led people in the US to see more content classified as uncivil or containing slurs during the 2020 elections but had little effect on measures of issue and affective polarization[65]. Moreover, the Facebook algorithm valued angry and other reactions as worth five likes in early 2018 and changed the weight of an angry reaction to zero in 2020[66]. However, Facebook and Twitter have been reported to demote and ban Ukrainian pages posting content about the war[67]. Another possibility is that Ukrainian social media users are reluctant to share explicitly derogatory content aimed at the Russian people, as opposed to aimed at the Russian government[45], or that people are afraid to express hostility for personal safety reasons. However, this does not explain why outgroup hostility was associated with a lot of angry reactions after the start of the invasion but became gradually less associated starting in May 2022 (Supplementary Fig. 9). At the same time, expressions of ingroup solidarity remained statistically significantly associated with engagement throughout, potentially sustaining the group's efforts toward victory[35]. Overall, the fact that ingroup solidarity gains engagement during conflict, more so than outgroup hostility, may be important for social media users as it may contribute to a more positive atmosphere[60,68] and for social media content creators to write posts that get traction. Moreover, it is important for social media platforms as a way to gauge if their algorithms are promoting unnecessarily hostile content.

There are several limitations to our study. Studies 2 and 3 were primarily exploratory; we did not create our classifiers and dictionaries prior to data collection but rather did so after we had already obtained the data (as we collected our data initially to replicate the findings by Rathje et al.[7] in the context of Ukraine). We thus recommend caution with the interpretation of our correlational findings and encourage further exploration and experimental testing. Our interpretation of our work with respect to psychological literature relies on the assumptions that (1) social media users experience emotions after reading relevant emotional posts (i.e., emotional contagion) and that (2) they behave (i.e., engage with the posts) because of their emotional experiences. Although these assumptions are somewhat supported by previous studies (see ref. 60 for (1) and ref. 15 for (2)), there are alternative explanations for why a user would engage with a post. In fact, a user liking an ingroup solidarity post may reflect a range of emotional states, including the user's current state, the user's desired state, an attempt to boost morale through motivated behavior, a change in online versus offline behavioral patterns, or it could be unrelated to the post's emotional content. Future experimental research in controlled laboratory and field settings is necessary to validate and clarify our findings further. We also do not know which users produced the recorded engagement and for what reason; it is possible that artificial users (i.e., bots) were partly responsible for driving these dynamics, although if this is the case, then it must (again) be true to a similar degree for both Facebook and Twitter, which have different bot policies and populations. Moreover, the populations of social media users across platforms are not representative of the general public and might be particularly skewed for platforms with small local audiences like Twitter[69]. It is also hard to say how much our results generalize beyond the Ukraine-Russia context, in part because of the skillful use of social media by the Ukrainian government to foster international solidarity[70]. Future research could compare the effects of ingroup solidarity and outgroup hostility across platforms, countries, conflicts, and affected communities (such as politicians, soldiers, activists, and civilians). Moreover, collaborations of researchers with social media platforms could help shed light on how different moderation strategies and content distribution systems influence engagement patterns, as the outcomes of socio-technical systems are often emergent[71]. Nevertheless, whether due to platform algorithms and moderation or human psychology (or both), our findings suggest that expressions of ingroup solidarity were likely to receive more engagement than outgroup hostility on Ukrainian social

media during the first six months of the 2022 Russian invasion of Ukraine.

## Methods

This study was approved by the University of Cambridge Research Ethics Committee (PRE.2022.022) and complies with all relevant ethical regulations. We selected news sources to include all of the most popular political online news media in Ukraine appearing in three independent top-100 and top-50 online rankings and previous studies (see Supplementary Methods for more details). These sources were classified as pro-Russian if they were based in Russia or banned in Ukraine for expressing pro-Russian sentiments as of August 2021. The rest were classified as pro-Ukrainian (see Supplementary Methods for more details). In total, our sample contained 108 news sources. Of those, 100 had a Facebook page, and 93 had Twitter accounts, 15 of which were pro-Russian for both. No statistical method was used to predetermine sample size. Twitter news sources data was collected at four timepoints (August 2021, and January, June, and September 2022) by retrieving the 3200 most recent posts for each account using the R package rtweet (see the temporal distribution of the data in the Supplementary Fig. 10). Facebook data was collected in bulk using CrowdTangle, a tool owned by Meta provided to researchers in our lab as a part of the Social Science One partnership, using historical download for each page in July 2023. The geolocated Twitter dataset for Study 3 was collected using the R package academictwitteR with the country setting set to Ukraine in April 2023 and resulted in 367,051 original non-reply Ukrainian or Russian-language tweets of which 289,592 were classified as pro-Ukrainian (148,959 posted after the invasion). All Twitter data was gathered in accordance with the Twitter Terms of Service. The distributions of posts and the proportion of posts classified as ingroup solidarity, outgroup hostility, and binary mentions of the ingroup and outgroup over time for each dataset can be seen in Supplementary Fig. 10. The descriptive statistics for all datasets are available in Supplementary Tables 5–14.

We treated Facebook and Twitter data in Ukrainian and Russian as similarly as possible. Post language was determined using the cld2 (v1.2.1) R package, and content in languages other than Ukrainian and Russian was excluded. We used the quanteda R package (v3.2.0) for dictionary analyses. We created a dictionary of descriptive mentions of Ukrainian and Russian identity, which included the country name, capital city, currency, political decision-making center (Bankova for Ukraine and Kremlin for Russia), ten largest cities, ten most popular politicians, and all of their morphological derivations (e.g., Ukrainian, Ukraine's, and so on for Ukraine). For instance, the Ukrainian dictionary contains words like Ukraine, Kyiv, Hryvnia, and Zelensky, while the Russian dictionary has Russia, Moscow, Ruble, and Putin. We did not include cities in the Donetsk or Luhansk regions or Crimea to keep the dictionary as neutral as possible and because both countries claim those cities to be part of their territory (e.g., Mariupol, Donetsk, Luhansk, etc.). For the moral-emotional language dictionary, we translated the dictionary used by Brady et al.[6] into Russian and Ukrainian. We used the Linguistic Inquiry and Word Count (LIWC) 2015 Ukrainian[49] and 2007 Russian[50] versions for the negative and positive affect dictionaries. All reported effects are robust to variations in the identity dictionaries, like not including cities and politicians (Supplementary Table 23), and the most frequent words from these dictionaries in our data are country names (Supplementary Fig. 4).

We counted the number of words from each dictionary present in a post to calculate its score in that category. We calculated engagement as the sum of all reactions a post has received (e.g., engagement on Twitter is the sum of retweets and favorites). We used overall engagement, as opposed to retweets or shares, as our main dependent variable because we are interested primarily in engagement and not sharing patterns. In addition, this metric allows us to better compare engagement across platforms. The results are similar for retweets or shares (Supplementary Tables 17–20). All analyses were performed in R, version 4.3.1. All variables were mean-centered using the R package jtools (v2.2.2), and outcome variables were log-transformed to correct for skewness.

For Study 1, we followed the analysis plan used by Rathje et al.[7] as closely as possible. We fit mixed effects linear regressions to predict log-transformed engagement based on descriptive in- and outgroup mentions, negative, positive, and moral-emotional words and adding account as the random effect. Wherever possible, we controlled for account follower count, whether the post contained a URL, whether it contained media, whether the post was a retweet (in the case of Twitter data), and the total word count. Across all three studies, all statistical tests are two-tailed, and the $p$-values and Cohen's $d$ are calculated using Satterthwaite d.f. The normality and equal variances checks for the main models presented in the paper can be found in the Supplementary Information. Although we note the presence of slight deviations from the assumptions, such deviations are common in real-world data sets and linear mixed models have been shown to be robust to distributional assumption violations[72,73]. We also conducted robustness checks with robust mixed effects models for the main regressions, which produced very similar results.

We used two different natural language processing approaches – custom dictionaries and fine-tuned BERT-style models – to measure ingroup solidarity (specific for Ukraine) and outgroup hostility (specific to how Ukrainians view Russians). We created the BERT-NLI models by manually labeling 1000 posts from before the invasion and 1000 from after, half from Facebook and half from Twitter, for ingroup solidarity and outgroup hostility (see Supplementary Methods for the code book; all labeling throughout the paper was done by the first author). We then fine-tuned ingroup solidarity and outgroup hostility multilingual Natural Language Inference DeBERTa models following Laurer et al.[53] on a subset of 1600 labeled posts and used the remaining 400 for testing. Both classifiers achieved an accuracy of ~0.87 and F1-macro of 0.80. We also developed ingroup solidarity and outgroup hostility dictionaries that were based on our observations of the Facebook data in this sample and domain knowledge. The dictionaries do not contain neutral Ukrainian or Russian words like Russian but, instead, have implicit references that reflect positively on Ukraine or Ukrainians (for ingroup solidarity) and negatively on Russia or Russians (for outgroup hostility). We also used words containing dehumanizing language, drawing on dictionaries used in prior work[74] in the outgroup hostility dictionary. For instance, the ingroup solidarity dictionary contains words like glory, defender, Cossack, and Maidan, while the outgroup derogation dictionary contains Moskal, (a derogatory word for Russian) occupant, rashist (a portmanteau of Russia and fascist), and animal (see Supplementary Fig. 5 for word clouds of top words from the dictionaries in our data). The performance of ingroup solidarity and outgroup hostility dictionaries was worse than the BERT-NLI performance: accuracy of 0.78 and 0.80 and F1-macro of 0.58 and 0.66, respectively (Supplementary Tables 15 and 16). However, our results are largely the same for the dictionaries and the BERT-NLI models (Supplementary Figs. 6 and 7).

Finally, we calculated the estimates for each engagement predictor over time with a sliding window approach. Starting at the first date in our Facebook sample (13 July 2021), we used the data over the first 14 days to fit a mixed effects linear regression model predicting log-engagement, then we shifted the time window by one day and fit a new regression, and so on. We repeated the procedure until there were no more days to shift over. We performed the same sliding window procedure with the BERT-NLI models and dictionaries for pro-Ukrainian Facebook and Twitter, controlling for the same variables as in Study 1 (see Supplementary Figs. 6 and 7 and Table 29 for dictionary results). To make the coefficients between BERT-NLI binary classifications and the descriptive mentions of ingroup and outgroup identity comparable, we turned the latter into binary indicators of

whether a post contained at least one word from the corresponding dictionary. The binarized dictionaries of descriptive ingroup and descriptive outgroup mentions achieved good performance: accuracy of 0.76 and 0.9 and F1-macro of 0.76 and 0.87, respectively (Supplementary Tables 15 and 16). Our results stay largely the same depending on the window size (Supplementary Fig. 5).

For Study 3, we first labeled each tweet as expressing a pro-Ukrainian or pro-Russian point of view by using a multilingual RoBERTa model[75] fine-tuned on a dataset of 30 thousand social media posts related to the war manually labeled as expressing pro-Ukrainian or pro-Russian view by trained annotators at the Center for Content Analysis in Ukraine(see ref. 76 for the development of labeling strategy). Although the training data spanned May 2022 – February 2023, the model achieved high performance on a set of 400 Twitter posts from our data: accuracy of 0.86 and F1-macro of 0.81. We excluded all posts identified as pro-Russian (21.1%). We then further fine-tuned our BERT-NLI models on a set of 1000 Twitter posts and increased the classification threshold of the models to 0.999 probability as the models qualitatively seemed to overclassify unrelated content (this decision does not influence our results, see Supplementary Table 30). The geolocated Twitter ingroup solidarity and outgroup hostility models had an accuracy of 0.94 and 0.96 and an F1-macro of 0.75 and 0.82, respectively. We fit a mixed effects linear regression predicting log-transformed engagement based on the same variables as in Study 2 and whether the user was verified using only the data from after the invasion. Across all three studies, regressions without the descriptive mention variables show largely the same coefficients for ingroup solidarity and outgroup hostility as regressions with the variables (Supplementary Table 21).

### Reporting summary
Further information on research design is available in the Nature Portfolio Reporting Summary linked to this article.

## Data availability
The raw social media data are protected and are not available due to data privacy laws. Please reach out to the corresponding authors for the Twitter post IDs and CrowdTangle page IDs. The processed (anonymized, partial) social media data are available on OSF: 10.17605/OSF.IO/RMC3E.

## Code availability
The code needed to replicate all analyses is available on OSF: 10.17605/OSF.IO/RMC3E.

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

## Acknowledgements

We want to extend our gratitude to the Center for Content Analysis in Ukraine, in particular to Artem Zakharchenko and his team, including Yevhen Luzan, Olena Zakharchenko, Olexiy Rogalyov, Olena Zinenko, Yuliia Maksymtsova, Maryna Fursenko, Valeriia Molotsiian, and Anhelika Machula. We are grateful to Eliot Smith and Diane Mackie for their helpful and insightful comments. We would also like to thank Steve Rathje, Jay Van Bavel, Andrea Jones-Rooy, and the members of the Cambridge Social Decision-Making Lab and the New York University Social Identity and Morality Lab for helpful feedback. This work was supported by a Gates Cambridge Scholarship (#OPP1144) awarded to Y.K. and funding from the British Academy (#PF21\210010) awarded to J.R.

## Author contributions

Y.K., T.B., and J.R. conceptualized the study. Y.K. and J.R. performed research. Y.K. analyzed data. Y.K., S.v.d.L., and J.R. wrote the paper.

## Competing interests

The authors declare no competing interests.
