## [Peer Review File · Nature Communications]

Reviewers' Comments:

Reviewer #1:

Remarks to the Author:

I like this piece a great deal and am favorably inclined towards seeing it published in Nature Communications. The authors seek to replicate previous research regarding the drivers of engagement with social media posts in two new contexts: first, outside the confines of the United States (where the vast majority of previous research of this type have been done); and second, before and after a major shock to the political environment, in this case the Russian invasion of Ukraine for engagement with posts by Ukrainian media publications. It has a clever research design to identify posts by pro-Russian and pro-Ukrainian media sources among Ukrainian media sources (although, as I'll note below, this is not laid out as clearly as it could be), and it has very interesting findings. Whereas prior to the war engagement seems to be fairly similar to what has been seen in the United States (posts with out-group references engender more interaction than in-group references, and moral-emotional language boosts engagement, although there are some exceptions), after the war "ingroup solidarity" posts get the most engagement, and much more than "outgroup hostility". The former finding is interesting because most prior research in the US has focused on partisan, as opposed to ethnic/national in-groups/out-groups, while the latter is interesting because it is a novel finding about what happens post shock to society (and in wartime). For the most part, the findings are clearly laid out, and I think the paper will add to our knowledge of what drives engagement with social media. Equally interesting, it seems to be a nice illustration of what we can learn from digital trace data about what is important to people during wartime.

I have one major question about the paper, one recommendation that I think is important for the authors to address, and then two suggestions that I think will make the paper more accessible to readers.

Here's my major question: why are the authors relying on dictionaries when the state of the art for classifying social media posts is supervised machine learning, and especially large language transformer models (e.g., BERT, RoBERTa, etc.)? I understand the draw to use the moral-emotional dictionary from Brady et al. b/c the hypotheses there are actually about the presence of particular words in posts, and I can see why it is easier to try to use LIWC for positive and negative words, but I don't understand why the new categories of ingroup solidarity and outgroup hostility are relegated to dictionaries when the authors could use supervised ML. The paper does not present any validation of these dictionaries (nor of the descriptive dictionaries, for that matter), and it just makes one wonder about the robustness of findings to what was included in the dictionary. For starters, I would like to see the lists of words in these dictionaries translated into English (all I see right now are word clouds in Ukrainian and Russian in Figures S3 and S4, which makes any sort of validation difficult). But more importantly, I would like to see actual examples of posts produced by these dictionaries to see if they match what one would expect for in-group solidarity and out-group hostility to see if they actually accurately capture these concepts. As best I can tell from the methods and materials section, the analysis is relying on the number of words in a post from these dictionaries and not a post-level dichotomous indicator as being in a category or not. So, for example, a curse word followed by "Russia", as best I can tell, would enter the analysis as one descriptive word (Russia), and probably one negative word (the curse) and maybe a moral emotional word (the curse?), but that would not be counted as out-group hostility. But clearly "F*\$k Russia" in the title of a news story should be coded as out group hostility, no? And this is exactly what we would expect supervised ML to pick up. So I guess my bottom line here is that I would like a lot more information about why we should actually trust that the in-group solidarity and out-group hostility classifiers are actually picking up news stories/posts that would be classified in those terms. Alternatively, if all the analysis is about word used in posts, then I would like that to be much clearer, and I'd like the authors to provide the dictionaries translated into English for out-group hostility and in-group solidarity that are novel for the point of this paper.

The recommendation I have is that there is nothing in the paper about contextualizing FB and Twitter usage in Ukraine. From reading the current version of the paper, one might expect readers to think FB and Twitter usage in Ukraine is similar to in the United States. But there is a history of other (Russia-based) platforms being used in Ukraine, and although these were banned in Ukraine after the 2014 annexation of Crimea, they were still being used in 2021 at similar numbers to Twitter (<https://www.statista.com/statistics/1278407/most-popular-social-media-ukraine/>). So it might be useful to say something about the popularity of these platforms in Ukraine, and who uses Facebook and Twitter in Ukraine.

But also more generally – what do the authors mean by “Ukrainian Facebook and Twitter”. A literal read of this would likely mean the use of Facebook and Twitter by people living in Ukraine. However, I think what it means here is reactions on Facebook and Twitter to posts by Ukrainian media outlets. Do the authors know if these reactions are coming from Ukrainians, or from people living in Ukraine? If so (or if not), this should definitely be made explicit.

I also found it someone strange that, unless I missed this, there is no acknowledgement of the fact that Ukrainian media publish both in Russian and in Ukrainian. I would have expected to see controls for (or interactions with?) the language of the post in the regression analyses, but yet I don't see anything along these lines in the supplemental appendices. Why is that?

My two suggestions for the authors are as followed. First, as currently laid out, it is quite difficult to use the tables in the supplementary materials that are designed to show the robustness of findings to actually do so because the tables are all on different pages. Could you simply put all robustness tests of a finding on the same table, so rows would be variables (as it is now) and columns would be different specifications? This would make it much easier to visibly inspect claims about the robustness of findings.

Second, I think it needs to be much clearer that your categories of pro-Russian and pro-Ukrainian – at least at best as I understand this – are based on publication level classifications, not article level classifications. In this way, actually, study 2 is much clearer, because the pro-Russian category is simply dropped. But for study 1, it would be useful if the authors made this distinction clearer, because the draft often reads as if it is the posts about the articles themselves that have a pro-Russian characterization, as opposed to coming from pro-Russian publications. And with that in mind, it would be useful to know if it is actually the case that pro-Russian publications are always providing pro-Russian articles, especially once the war starts? Given that these are Ukrainian publications and post-war most Russian speakers in Ukraine were very pro-Ukraine in terms of the war, I would like to know if that characterization (that the publications continued to produce pro-Russian articles) was indeed correct. But more generally, in the effort to tie everything in Study 1 to the Rathje et al. article, this distinction feels like it is getting lost. Indeed, it might be useful to provide a brief summary of the Rathje et al. approach – did that publication also rely on source level classification of a publication as pro-left or pro-right, or did it rely on post-level classification?

Reviewer #2:

Remarks to the Author:

The manuscript tests how group-based language and emotional language predict engagement of news headlines across different contexts (pre and post war Russia/Ukraine). The paper has solid theoretical framing based in SIT and it should be commended for extending previous work by examining a different culture and going through the work to translate new lexicons. In general, the manuscript has potential but there are several methodological limitations that are concerning, and the authors also overstate conclusions throughout.

Study 1

1 – The “descriptive” ingroup/outgroup language seems psychologically opaque. When a news headline mentions one’s own country or another enemy country, surely it increases attention because it involves some salient current event, but it is unclear what exactly it is because current events are so heterogeneous. For example, it is some group-threat headline or is it simply a headline about Ukraine’s weather. Both are motivationally relevant and could draw engagement but one doesn’t have anything to do with group identity per se (yet both are categorized the same by the authors’ method). The paper could do a much better job at investigating what these highly viral “descriptive group” language categories are actually tracking in the headlines. I would like to see some content analysis and a discussion of what the results yield so that the authors can link the content to some psychological concept of interest.

2 – The authors state that they test how word counts predict sum of engagement metrics by using OLS regression. Generally, engagement sums produce highly skewed distributions that are typically not appropriate for OLS regression even after they have log-transformed. The methods do not show any distributions so I couldn’t judge if this is the case. The OSF link provided in the methods is dead (“Resource Deleted”) so I could not see their code. The authors should include greater details in the method (not everyone is going to examine their code) and demonstrate that they are modeling the outcome correctly, either using negative binomial model that is appropriate for over-dispersed count data or demonstrating the log-transformation fixes the skew appropriately. My assumption is that since the authors are looking at popular news headlines that get more engagement than normal, the log-transformation has a better chance of working out, but the authors should demonstrate this.

Study 2

1 – I was not convinced that the new measure of “group solidarity and hostility” are well-defined. The authors state they added words like “defenders,” “heroes,” and “occupiers.” These words simply appear to be war-related emotional words, which the authors study in the context of war so of course it will skew the category to have an engagement advantage. It is also worth noting that these words have overlap with several words cited in the “moral-emotional” lexicon (also see comment in “General” below).

General

1 – To summarize the above methodological concerns, my biggest concern with the manuscript is that the group-based language categories were not carefully constructed, not based on theory, and were not validated in any way either to contrast against other categories they measured or to confirm they are measuring the psychological construct of interest. However, it is crucial to do so since it is the main construct of interest for the manuscript. The papers are replicating appear to do some version of content analysis and validation if they would like to see an example of how to proceed.

2- The authors chose to study engagement to popular news headlines rather than messages by ordinary users or political elites. This choice constrains the results and the ability of the authors to compare it to other work.

First, in Study 2 the authors leave out Russia headlines due to propaganda, but is there any evidence that Ukraine national news groups are not engaging in propaganda strategies? Whether in a corrupt sense or just a response to the context, of course war time will result in popular headlines supporting the country vis-à-vis the war. The main concern is that war-time motives for propaganda type news headlines is a competing explanation for the results of Study 2 (as opposed to some group-level shift in attraction to group solidarity). Both could explain the results, but can the authors point to their preferred explanation over the other?

Second it is important to highlight further that because the authors chose to study popular news headlines, it significantly limits the variation in language, especially intense emotional language

that is used compared to ordinary users who may be having conflicts on the platform. The authors mention this briefly on pg. 4 but it should be highlighted when making a claim about how the study does or does not replicate previous results. If anything, it extends rather than replicates previous results in a new context of news headlines.

This means the authors should more clearly highlight that their studies are about news headlines in the abstract and title. The paper is not one about social media engagement in general, it is about news headline engagement which restricts the claims the authors can make.

3 – Related: the authors make a needlessly generalized claim in the abstract: “Our results suggest that signaling solidarity with one’s in-group may be more important than displaying negativity about out-groups during periods of intense conflict.” The statement should include a caveat that the study was about popular news headlines and that the findings are specific to Ukraine / Russia.

4 – The authors’ main conclusion is that group-based language engagement is subject to exogenous shocks / context. This is interesting but it also appears that the moral-emotional category had the most consistent results across both texts. The authors should highlight this finding since they are comparing several categories of language.

I also thought there is room to better highlight the inconsistent negative affect findings to a greater extent, because several recent studies have made claims about how negative headlines always draw more engagement. For instance in NHB the study “Negativity drives online news consumption” by Van Bavel et al.

Conclusion

The authors should be commended for the cross-cultural comparison and extension of previous work in a war context. I think there is a version of this paper that is a nice contribution to the psychology of news headlines engagement, but it is not there yet.

Reviewer #3:

Remarks to the Author:

Review of NCOMMS-23-01909. This manuscript describes associations between the content of Facebook and Twitter posts made by Pro-Ukrainian and Pro-Russian media outlets, and engagement with these posts, in the context of the Ukraine-Russia war. Prior findings that moral-emotional words are associated with more engagement are replicated. Results for other classes of words are more varied across platforms and country. Furthermore, changes over time in engagement with posts containing “ingroup solidarity” and “outgroup hostility” words are described. The authors frame these findings in terms of affective polarization and interpret the findings as advancing the understanding of inter-group dynamics.

Studying these associations in the context of the Ukraine-Russia conflict is interesting, as it provides a unique documentation of the correlates of full-blown international conflict as it occurs, and because I agree that most of the literature in this area is too US-centric. We indeed need more evidence from other social and cultural settings. As such, I welcome this manuscript. However I think there is too little scientific advancement offered in this manuscript to be published in Nature Communications.

1. My key concern is that the manuscript essentially repeats the descriptive approach of prior work (enhanced by introducing two new classes of words, see validity concerns below), but that it continues to leave the meaning of the observed patterns unexplained. This would not be a problem if the patterns were unambiguous, or if the patterns had clear meaning in terms of understanding human psychology/behavior in the context of inter-group relations. Unfortunately neither applies to the current manuscript. E.g., the substantial between-platform and between-country variation

(e.g., the outgroup words in Pro-Russian media in Fig 2) calls the generalizability of these findings into question, and makes it unclear what these findings really mean. Are we looking at artefacts of social media algorithms, or true human interaction patterns?

2. More insight into why certain posts invite more or less engagement could help here, but the methodological approach is too limited for this. Essentially it is word counting, leaving the meaning of these words and engagement with them unexplained. E.g., what does engagement with a post by a pro-Ukrainian outlet with an outgroup reference such as 'Moscow' 'Kremlin' and 'Putin' reflect? In the manuscript these words are interpreted as identity language, but I have difficulty seeing how mentioning the capital of another country helps define or build an identity. My hunch is that these words are likely to occur in informational posts about politics (we are looking at posts by news outlets, after all), so retweeting/sharing these posts could simply serve to spread information about current and important developments in international relations. Extending that logic, it makes sense that the Ukrainian public would keep a close eye on what was happening in Moscow in the run-up to the invasion, explaining why outgroup-related words would invite more engagement (Fig 2A) – without this having any bearing on identity. While combining classes of words (eg the positive x outgroup and negative x outgroup interactions) might shed some more light on what the word counts actually reflect (spreading negative rumor, saying positive things, etc), there is no way of distinguishing between interpretations based on the current evidence.

3. In light of a growing body of evidence documenting such correlations between words and engagements, I feel we need more context to these correlations to really get a handle on what they mean for affective polarization and understanding inter-group relations. For instance, the word counting approach might be complemented with a second methodological approach to provide a more in-depth interpretation of what the patterns actually mean and reflect in terms of human psychology, to merit publication in an outlet such as this (compare, e.g., a paper by Ashokkumar & Pennebaker, 10.1126/sciadv.abg7843, who combine their content analysis of reddit posts with survey data to better understand what references to different groups mean). Such an approach has the potential to really shed light on which motives drive engagement with 'ingroup solidarity' words: bolstering one's online reputation, recasting intergroup relations, or simply sharing the available information about the ongoing war, which likely all contains references to one's own and the other country?

Aside from these concerns about contribution, I have some additional concerns about the validity of some aspects of the approach:

4. The measure of outgroup hostility and ingroup solidarity words relies purely on face validity, and no further validation steps were taken. While the dictionary approach is already quite a crude approach even when using a well-validated word list such as LIWC, there is no way of knowing whether the words that were identified to reflect solidarity or hostility actually map onto these underlying psychological constructs. Moreover, the words that are used here (to the extent that they are translated into English and reported in the ms.) seem to have meaning only in the context of a war. This is important because the same word list is applied prior to the invasion. But what does, e.g., an 'ingroup solidarity' word such as 'hero' mean at that point in time? And would people engage with it for the same reason as after the invasion? Given that the introduction of this concept is half of the contribution of this manuscript (the replication of prior findings in the Ukraine-Russia war being the other half), I think a more careful measurement approach is called for.

5. Related to this, I looked for, but could not find a report on the baserates of the various classes of words. Reporting baserates could help shed more light on, or even falsify the assumption implied in the previous paragraph: that all bellicose words are more frequent after the outbreak than before, simply because there is a war going on at that point.

6. Another validity concern is that, while the possibility of bots contributing to the dataset is

mentioned in a few places, no attempt seems to have been taken to remove bot activity from the dataset. Since the influence of bots is explicitly mentioned in the interpretation of the strikingly 'deviant' engagement pattern on Pro-Russian Twitter (Fig 2A), I wonder to what extent other findings may actually be artefacts of bot activity.

7. A final concern is that I find the connection between the findings and the broader literature of affective polarization somewhat underdeveloped. In particular it is unclear whether the results are regarded as a cause, consequence, or symptom of affective polarization? What is the connection between the various classes of words (even if we go by the labels assigned to them) and the theory?

8. As a minor sidenote, while I can see why the authors argue that a broader perspective on affective polarization than just based on partisanship is needed to fully understand it, I think this also makes the need for affective polarization as a separate construct superfluous. Affective polarization is, at least to my knowledge, a term introduced by political scientists to apply the broader concepts of ingroup favoritism and outgroup hostility derogation from Social Psychology / Social Identity Theory to the (US) political partisan context. By broadening affective polarization to include international relations, it seems to me that we are back at the original concepts from Social Identity Theory.

In conclusion, while I think the manuscript indeed adds to the existing body of evidence on these findings, and while I welcome the broadening of the US-centric scope of this literature, I think the manuscript does not really advance the understanding of why these patterns occur. While I in principle like the research and welcome the findings, I think the methodological approach and interpretation are currently too limited for this outlet.

Reviewer #1

I like this piece a great deal and am favorably inclined towards seeing it published in Nature Communications. The authors seek to replicate previous research regarding the drivers of engagement with social media posts in two new contexts: first, outside the confines of the United States (where the vast majority of previous research of this type have been done); and second, before and after a major shock to the political environment, in this case the Russian invasion of Ukraine for engagement with posts by Ukrainian media publications. It has a clever research design to identify posts by pro-Russian and pro-Ukrainian media sources among Ukrainian media sources (although, as I'll note below, this is not laid out as clearly as it could be), and it has very interesting findings. Whereas prior to the war engagement seems to be fairly similar to what has been seen in the United States (posts with out-group references engender more interaction than the in-group references, and moral-emotional language boosts engagement, although there are some exceptions), after the war "ingroup solidarity" posts get the most engagement, and much more than "outgroup hostility". The former finding is interesting because most prior research in the US has focused on partisan, as opposed to ethnic/national in-groups/out-groups, while the latter is interesting because it is a novel finding about what happens post shock to society (and in war-time). For the most part, the findings are clearly laid out, and I think the paper will add to our knowledge of what drives engagement with social media. Equally interesting, it seems to be a nice illustration of what we can learn from digital trace data about what is important to people during wartime.

- Thank you for your kind words and your constructive comments; we have now made major changes to the manuscript, including the recommended classifier-based approach, which we believe have significantly improved the quality of our manuscript and substantially strengthened our findings. Please find our responses to your comments below.

I have one major question about the paper, one recommendation that I think is important for the authors to address, and then two suggestions that I think will make the paper more accessible to readers.

Here's my major question: why are the authors relying on dictionaries when the state of the art for classifying social media posts is supervised machine learning, and especially large language transformer models (e.g., BERT, RoBERTa, etc.)? I understand the draw to use the moral-emotional dictionary from Brady et al. b/c the hypotheses there are actually about the presence of particular words in posts, and I can see why it is easier to try to use LIWC for positive and negative words, but I don't understand why the new categories of ingroup solidarity and outgroup hostility are relegated to dictionaries when the authors could use supervised ML. The paper does not present any validation of these dictionaries (nor of the descriptive dictionaries, for that matter), and it just makes one wonder about the robustness of findings to what was included in the dictionary.

- Thank you for pointing this out, and we agree. In our initial manuscript, we opted to follow the approach laid out by Rathje et al. (2021) as closely as possible, but we

agree that there are better ways to analyse our data. In line with your suggestion, we have therefore fine-tuned BERT-style ingroup solidarity and outgroup hostility classifiers following the framework of Laurer et al. (2023; <https://doi.org/10.1017/pan.2023.20>). In particular, we fine-tuned a state-of-the-art multilingual Natural Language Inference DeBERTa v3 model trained on over 100 languages (including Russian and Ukrainian; the model is accessible at <https://huggingface.co/MoritzLaurer/mDeBERTa-v3-base-xnli-multilingual-nli-2mil7>). This approach lets us achieve high performance with little labelled data and is particularly suited for data with imbalanced class proportions (which is the case here). An additional benefit of this approach is that we were able to give an explicit definition of ingroup solidarity and outgroup hostility to the models. We manually annotated 2000 posts from our pro-Ukrainian data (1000 before the invasion and 1000 after, half from Twitter and half from Facebook, randomly sampled) for ingroup solidarity and outgroup hostility (see the Code Book on OSF and the definition in the main text and below). A sample of 400 of these was used as the test set (not seen by the model at any stage of training). We performed hyperparameter search and fine tuning in two Google Colab notebooks (on OSF). The models will be made publicly available upon publication. Both ingroup solidarity and outgroup hostility achieve high F1 macro (.79 and .805, resp.) and balanced accuracy (.772 and .819, resp.) on the test sets. F1 macro, which is the average of the F1 scores for each class, is the most appropriate metric in this context as the data classes (e.g., solidarity vs. not) are highly imbalanced (base rates of around 10% solidarity/hostility before the invasion and 30% after, see Tables S4-6 in the SI for more descriptive statistics). We kept the dictionary classifiers for the descriptive ingroup and outgroup categories (e.g., in Study 2 a post was classified as reflecting descriptive ingroup if it contained one or more words from the ingroup dictionary) because their performance was already high (see the tables below, which are the same as Table S14 and S15 in the SI).

"Solidarity with Ukraine or Ukrainians": "The quote is expressing solidarity, liking or unity of Ukraine or Ukrainians, for example it praises Ukraine, mentions Ukrainians as competent, good people or Ukraine as a great, strong or united nation."

"Hostility towards Russia or Russians": "The quote is expressing hostility, derogation or disliking of Russia or Russians, for example it criticises Russia, mentions Russians as

incompetent, immoral people or the Russian Federation as a bad, weak or failing nation.”

Table S14. Classifier validation results (Facebook and Twitter news sources). Results are based on a dataset of 400 posts stratified by class (different for ingroup solidarity and outgroup hostility).

	F1 Macro (average of F1 per class)	Balanced Accuracy (average of recall per class) (imbalanced; proportion of correctly classified)	Accuracy (proportion of correctly classified)
Ingroup Identity			
Dictionary	0.759	0.788	0.759
Untrained Ukrainian and Russian speaker	0.832	0.82	0.847
Outgroup Identity			
Dictionary	0.872	0.848	0.9
Untrained Ukrainian and Russian speaker	0.893	0.92	0.905
Ingroup Solidarity			
Dictionary	0.576	0.569	0.782
Untrained Ukrainian and Russian speaker	0.665	0.634	0.842
BERT-NLI	0.79	0.772	0.868
Outgroup Hostility			
Dictionary	0.66	0.654	0.795
Untrained Ukrainian and Russian speaker	0.731	0.689	0.868
BERT-NLI	0.805	0.819	0.873

Table S15. Classifier validation results (geolocated Twitter). Results for everything but pro-Ukrainian classifier are based on the data classified as pro-Ukrainian by the model (309 posts).

	F1 Macro (average of F1 per class)	Balanced Accuracy (average of recall per class) (imbalanced; proportion of correctly classified)	Accuracy (proportion of correctly classified)
Pro-Ukrainian RoBERTa	0.812	0.798	0.86
Ingroup Identity (Dictionary)	0.833	0.802	0.877
Outgroup Identity (Dictionary)	0.83	0.786	0.958
Ingroup Solidarity			
Dictionary	0.602	0.676	0.854
BERT-NLI (finetuned)	0.748	0.748	0.942
Outgroup Hostility			
Dictionary	0.692	0.683	0.929
BERT-NLI (finetuned)	0.821	0.836	0.955

For starters, I would like to see the lists of words in these dictionaries translated into English (all I see right now are word clouds in Ukrainian and Russian in Figures S3 and S4, which makes any sort of validation difficult).

- Thank you! We have now added translations to the supplement and the OSF (see screenshot below).

Descriptive ingroup dictionary (translated):

Ukraine, Kyiv, Kiev, Kharkiv, Lviv, Lvov, Zaporizhzhia, Odessa, Dnipro, Mykolaiv, Vinnytsia, Krivyyi Rih, Zhytomyr, Ivano-Frankivsk, Sumy, Hryvna, Bankova, Zelensky, Avakov, Yermak, Medvedchuk, Poroshenko, Shefir, Venediktov, Shmygal, Razumkov, Kosiuk.

Descriptive outgroup dictionary (translated):

Russia, RF, Moscow, Saint Petersburg, Peter, Novosibirsk, Ekaterinburg, Samara, Kazan, Omsk, Chelyabinsk, Rostov, Ufa, Novgorod, Volgograd, Krasnoyarsk, Perm, Voronezh, Khabarovsk, Pyatigorsk, Ruble, Kremlin, Putin, Mishustin, Vaimo, Medvedev, Shoigu, Kiriyenko, Sobyenin, Sechin, Lavrov, Miller.

Ingroup solidarity dictionary (translated):

Glory, defender*, hero*, fighter*, battle*, light*, eternal, memory, we stand, defend*, defense, native, own land, homeland, warrior*, anthem, vyshyvanka, coat of arms, flag*, sich, 2014, viburnum*, mother, cossack*, spirit, volunteer*, legend*, perish*, cherish*, care*, protect*, victory, veteran*, cyborg*, borsch, palanytsya, heavenly hundred, maiden.

Outgroup hostility dictionary (translated):

Moskal*, katsap*, vatnik*, traitor*, fascist*, occupant*, occupier*, hostile, enemy, enemies, killer*, bully*, executioner*, rapist*, raped, scum, orc*, fool*, nonhuman*, revenge, punish, rashist*, invader*, backward, uncultured, cage, beast*, cockroach*, cold-blooded*, dominate, addict*, sick*, rude, creature, exterminate, extinct, wild, greedy, illogical, immoral, irrational, contagious, infection, irresponsible*, lazy, monkey*, monster*, neanderthal*, parasite*, pig*, poison*, predator*, mad*, rat*, outlaw, outcasts, spineless, subhuman, ignorant, illiterate, cancer, uncultured, undeveloped, ungrateful, stupid, venom*, pest*, vermin*, nest*, flea*, primitive*, bastard*, stranger*, alien*, blindly, sneaky, heartless, ruthless, materialistic, rob*, thief*, marauder*, passive, tough*, unsophisticated, unemotional, unreliable.

But more importantly, I would like to see actual examples of posts produced by these dictionaries to see if they match what one would expect for in-group solidarity and out-group hostility to see if they actually accurately capture these concepts.

- This makes sense; we have now added Table 1 in the main body, which shows examples of posts reflecting ingroup solidarity and outgroup hostility.

As best I can tell from the methods and materials section, the analysis is relying on the number of words in a post from these dictionaries and not a post-level dichotomous indicator as being in a category or not. So, for example, a curse words followed by “Russia”, as best I can tell, would enter the analysis as one descriptive word (Russia), and probably one negative word (the curse) and maybe a moral emotional word (the curse?), but that would not be counted as out-group hostility. But clearly “F*\$k Russia” in the title of a news story should be coded as out group hostility, no? And this is exactly what we would expect supervised ML to pick up. So I guess my bottom line here is that I would like a lot more information about why we should actually trust that the in-group solidarity and out-group hostility classifiers are actually picking up news stories/posts that would be classified in those terms. Alternatively, if all the analysis is about word used in posts, then I would like that to be much clearer, and I’d like the authors to provide the dictionaries translated into English for out-group hostility and in-group solidarity that are novel for the point of this paper.

- Thank you, and this makes a lot of sense. We have now validated our initial findings using a classifier-based approach (see above), which we have moved to the main text due to its improved robustness compared to the dictionary-based approach; we moved the latter to the supplement (see Figures S6 and S7 and Table S29). Both approaches, which rely on entirely different assumptions and methods, show exactly the same finding, namely that ingroup solidarity became by far the best predictor of engagement on both Facebook and Twitter after the full-scale invasion.
- Additionally, we have now validated our findings on a dataset of geolocated Ukrainian Twitter data published between July 2021 and September 2022 (and not just data from news outlets). Doing so again validates our findings, even with such a noisy dataset. This has added substantial confidence to our conclusions.

The recommendation I have is that there is nothing in the paper about contextualizing FB and Twitter usage in Ukraine. From reading the current version of the paper, one might expect readers to think FB and Twitter usage in Ukraine is similar to in the United States. But there is a history of other (Russia-based) platforms being used in Ukraine, and although these were banned in Ukraine after the 2014 annexation of Crimea, they were still being used in 2021 at similar numbers to Twitter (<https://www.statista.com/statistics/1278407/most-popular-social-media-ukraine/>). So it might be useful to say something about the popularity of these platforms in Ukraine, and who uses Facebook and Twitter in Ukraine.

- Thank you! This is a good point; we have now added an explanation of the use of Facebook and Twitter in Ukraine, see page 3.

But also more generally – what do the authors mean by “Ukrainian Facebook and Twitter”. A literal read of this would likely mean the use of Facebook and Twitter by people living in Ukraine. However, I think what it means here is reactions on Facebook and Twitter to posts by Ukrainian media outlets. Do the authors know if these reactions are coming from Ukrainians, or from people living in Ukraine? If so (or if not), this should definitely be made explicit.

- Thank you; you are right in pointing out that in the data we used in our initial manuscript, we were unable to speak to this. We have now added an additional dataset of geolocated Twitter data published from within Ukraine. Here, again, the results are the same as those from our earlier draft and in our other analyses (i.e., ingroup solidarity becomes the best predictor of engagement), although there are some nuances. See Introduction (page 4) and Results (page 6).

I also found it someone strange that, unless I missed this, there is no acknowledgement of the fact that Ukrainian media publish both in Russian and in Ukrainian. I would have expected to see controls for (or interactions with?) the language of the post in the regression analyses, but yet I don’t see anything along these lines in the supplemental appendices. Why is that?

- Thanks for raising this point; media in Ukraine indeed publish in both Ukrainian and Russian, but it is important not to conflate language with identity. For instance, most Russian-language outlets in Ukraine are pro-Ukrainian, not pro-Russian. We therefore thought it preferable to use a language-agnostic approach, and use different dictionaries for the two languages (i.e., translations of each other). Furthermore, to address this point in more detail, our new BERT-style classifiers are multilingual and have been trained using both Russian and Ukrainian-language posts; they are thus able to detect ingroup solidarity and outgroup hostility in both languages.

My two suggestions for the authors are as followed. First, as currently laid out, it is quite difficult to use the tables in the supplementary materials that are designed to show the robustness of findings to actually do so because the tables are all on different pages. Could you simply put all robustness tests of a finding on the same table, so rows would be variables (as it is now) and columns would be different specifications? This would make it much easier to visibly inspect claims about the robustness of findings.

- Thank you; we appreciate the suggestion and are sorry for the inconvenience. We tried putting all the robustness checks into the same table but it quickly became extremely large so we were forced to have multiple tables. For the variations in descriptive identity dictionaries, we did manage to put the main results into one table (Table S21). We also added a Table of Contents to the SI, which hopefully should make comparison easier. Tables S1-S3 report the results from the main analysis in the manuscript. Tables S4-13 have descriptive statistics. Tables S16-S29 in the supplement provide the robustness checks. These results all show the same results, which has added to our confidence in our findings.

Second, I think it needs to be much clearer that your categories of pro-Russian and pro-Ukrainian – at least at best as I understand this – are based on publication level classifications, not article level classifications. In this way, actually, study 2 is much clearer, because the pro-Russian category is simply dropped. But for study 1, it would be useful if the authors made this distinction clearer, because the draft often reads as if it is the posts about the articles themselves that have a pro-Russian characterization, as opposed to coming from pro-Russian publications. And with that in mind, it would be useful to know if it is actually the case that pro-Russian publications are always providing pro-Russian articles, especially once the war starts? Given that these are Ukrainian publications and post-war most Russian speakers in Ukraine were very pro-Ukraine in terms of the war, I would like to know if that characterization (that the publications continued to produce pro-Russian articles) was indeed correct. But more generally, in the effort to tie everything in Study 1 to the Rathje et al. article, this distinction feels like it is getting lost. Indeed, it might be useful to provide a brief summary of the Rathje et al. approach – did that publication also rely on source level classification of a publication as pro-left or pro-right, or did it rely on post-level classification?

- Thank you for raising this point. We indeed chose a publication-level classification approach because that was the approach used by Rathje et al. (2021); we now explain this on page 3. For the geolocated Twitter data (Study 3), we have validated our approach by using a classifier that can detect pro-Russian and pro-Ukrainian language in social media content, not just news content. This is a substantial improvement compared to the original approach, as machine learning techniques are very well-suited for these kinds of classification tasks. Furthermore, doing so allows us to look beyond our initial classification of news sources as pro-Ukrainian and pro-Russian; we have now added Study 3, which looks at Twitter data geolocated to Ukraine (and not limited to news outlets); here, we again find very similar results as in Studies 1 and 2.

Reviewer #2

The manuscript tests how group-based language and emotional language predict engagement of news headlines across different contexts (pre and post war Russia/Ukraine). The paper has solid theoretical framing based in SIT and it should be commended for extending previous work by examining a different culture and going through the work to translate new lexicons. In general, the manuscript has potential but there are several methodological limitations that are concerning, and the authors also overstate conclusions throughout.

- Thank you for your very thoughtful and insightful comments. We have now made major changes to our manuscript, most notably by adopting a classifier-based (rather than a dictionary-based) approach as our primary mode of analysis. In addition, we have validated our findings with a series of additional robustness checks, which has substantially increased our confidence in our conclusions. Regardless of the methodological approach used, we continue to find that ingroup solidarity became by far the strongest predictor of engagement on Ukrainian social media after the start of the 2022 full-scale invasion. Please find detailed responses to your comments below.

Study 1

1 – The “descriptive” ingroup/outgroup language seems psychologically opaque. When a news headline mentions one’s own country or another enemy country, surely it increases attention because it involves some salient current event, but it is unclear what exactly it is because current events are so heterogeneous. For example, it is some group-threat headline or is it simply a headline about Ukraine’s weather. Both are motivationally relevant and could draw engagement but one doesn’t have anything to do with group identity per se (yet both are categorized the same by the authors’ method). The paper could do a much better job at investigating what these highly viral “descriptive group” language categories are actually tracking in the headlines. I would like to see some content analysis and a discussion of what the results yield so that the authors can link the content to some psychological concept of interest.

- Thank you for pointing this out, and we largely agree; Study 1 aimed to replicate the findings by Rathje et al. (2021) as closely as possible within the context of Ukraine. While doing so, we noticed that, even though the Rathje et al. (2021) study is called “Outgroup animosity drives engagement on social media”, the authors did not explicitly measure outgroup animosity but rather what we have dubbed “descriptive” outgroup language; mere mentions of the other group, without necessarily using affective or negative-emotional language (e.g., Democratic politicians on Twitter talking about Republicans). We agree that this method is somewhat crude and does not allow for a granular analysis of what types of language (and sentiment or affect, et cetera) actually drive engagement. However, in Study 1, we did not want to deviate from the original study too much, as our main goal was to see if Rathje et al.’s (2021) findings replicated in Ukraine. We therefore addressed these issues in Studies 2 and 3, where we now use a classifier-based approach to delve into the concepts of ingroup solidarity and outgroup hostility in more detail.

2 – The authors state that they test how word counts predict sum of engagement metrics by using OLS regression. Generally, engagement sums produce highly skewed distributions that are typically not appropriate for OLS regression even after they have log-transformed. The methods do not show any distributions so I couldn't judge if this is the case. The OSF link provided in the methods is dead ("Resource Deleted") so I could not see their code. The authors should include greater details in the method (not everyone is going to examine their code) and demonstrate that they are modeling the outcome correctly, either using negative binomial model that is appropriate for over-dispersed count data or demonstrating the log-transformation fixes the skew appropriately. My assumption is that since the authors are looking at popular news headlines that get more engagement than normal, the log-transformation has a better chance of working out, but the authors should demonstrate this.

- Thank you for pointing this out. Apologies for the OSF link: we could not anonymize our first OSF repository (the one you had the link to) because it was linked to a GitHub account. We then deleted that OSF and updated the manuscript with the new one, but the reviewers accidentally received the wrong version. We have now ensured that the OSF is correct and accessible. Our main analyses are now done by predicting log-transformed engagement with mixed effects linear regressions allowing the intercept to vary by account the post came from. The log transformation fixes the skew of the data overall and for most accounts and the random effect allows to control for the variation in how users might engage with different pages and the non-independence in the data due to many posts coming from the same page (see descriptive statistics in the SI Tables S4-7, there are also distribution plots for log-engagement for every page on the OSF in the plots/descriptive/ folder files ending with eng_hist.png, where we can see that for most pages the log-transformation fixes the skew completely). We have added a more detailed description of the log transformed linear regression to the methods section (page 3 and 8). As per your suggestion, we also attempted using mixed effects negative binomial regression (with non-log-transformed engagement as the DV); however, the residuals of the models were extremely skewed (see QQ plot 1 below) as compared to the mixed effects linear model with log-transformation (see QQ plot 2, which shows far less skew). We therefore decided to use the mixed effects linear model as our primary method of analysis.

Negative binomial QQ plot (the y-axis goes up to 600,000):

Linear with log-transformation QQ plot:

Study 2

1 – I was not convinced that the new measure of “group solidarity and hostility” are well-defined. The authors state they added words like “defenders,” “heroes,” and “occupiers.” These words simply appear to be war-related emotional words, which the authors study in the context of war so of course it will skew the category to have an engagement advantage. It is also worth noting that these words have overlap with several words cited in the “moral-emotional” lexicon (also see comment in “General” below).

- Thank you, and you are completely right that our initial manuscript did not do a thorough enough job at validating these concepts. To address this, we have therefore fine-tuned BERT-style ingroup solidarity and outgroup hostility classifiers following the framework of Laurer et al. (2023; <https://doi.org/10.1017/pan.2023.20>). In particular, we fine-tuned a state-of-the-art multilingual Natural Language Inference DeBERTa v3 model trained on over 100 languages (including Russian and Ukrainian; the model is accessible at <https://huggingface.co/MoritzLaurer/mDeBERTa-v3-base-xnli-multilingual-nli-2mil7>). This approach lets us achieve high performance with little labelled data and is particularly suited for data with imbalanced class proportions

(which is the case here). An additional benefit of this approach is that we were able to give an explicit definition of ingroup solidarity and outgroup hostility to the models. We manually annotated 2000 posts from our pro-Ukrainian data (1000 before the invasion and 1000 after, half from Twitter and half from Facebook, randomly sampled) for ingroup solidarity and outgroup hostility (see the Code Book on OSF and the definition in the main text and below). A sample of 400 of these was used as the test set (not seen by the model at any stage of training). We performed hyperparameter search and fine tuning in two Google Colab notebooks (on OSF). The models will be made publicly available upon publication. Both ingroup solidarity and outgroup hostility achieve high F1 macro (.79 and .805, resp.) and balanced accuracy (.772 and .819, resp.) on the test sets. F1 macro, which is the average of the F1 scores for each class, is the most appropriate metric in this context as the data classes (e.g., solidarity vs. not) are highly imbalanced (base rates of around 10% solidarity/hostility before the invasion and 30% after, see Tables S4-6 in the SI for more descriptive statistics). We kept the dictionary classifiers for the descriptive ingroup and outgroup categories (e.g., in Study 2 a post was classified as reflecting descriptive ingroup if it contained one or more words from the ingroup dictionary) because their performance was already high (see the tables below, which are the same as Table S14 and S15 in the SI).

"Solidarity with Ukraine or Ukrainians": "The quote is expressing solidarity, liking or unity of Ukraine or Ukrainians, for example it praises Ukraine, mentions Ukrainians as competent, good people or Ukraine as a great, strong or united nation."

"Hostility towards Russia or Russians": "The quote is expressing hostility, derogation or disliking of Russia or Russians, for example it criticises Russia, mentions Russians as

incompetent, immoral people or the Russian Federation as a bad, weak or failing nation.”

Table S14. Classifier validation results (Facebook and Twitter news sources). Results are based on a dataset of 400 posts stratified by class (different for ingroup solidarity and outgroup hostility).

	F1 Macro (average of F1 per class)	Balanced Accuracy (average of recall per class) (imbalanced; proportion of correctly classified)	Accuracy (proportion of correctly classified)
Ingroup Identity			
Dictionary	0.759	0.788	0.759
Untrained Ukrainian and Russian speaker	0.832	0.82	0.847
Outgroup Identity			
Dictionary	0.872	0.848	0.9
Untrained Ukrainian and Russian speaker	0.893	0.92	0.905
Ingroup Solidarity			
Dictionary	0.576	0.569	0.782
Untrained Ukrainian and Russian speaker	0.665	0.634	0.842
BERT-NLI	0.79	0.772	0.868
Outgroup Hostility			
Dictionary	0.66	0.654	0.795
Untrained Ukrainian and Russian speaker	0.731	0.689	0.868
BERT-NLI	0.805	0.819	0.873

Table S15. Classifier validation results (geolocated Twitter). Results for everything but pro-Ukrainian classifier are based on the data classified as pro-Ukrainian by the model (309 posts).

	F1 Macro (average of F1 per class)	Balanced Accuracy (average of recall per class) (imbalanced; proportion of correctly classified)	Accuracy (proportion of correctly classified)
Pro-Ukrainian RoBERTa	0.812	0.798	0.86
Ingroup Identity (Dictionary)	0.833	0.802	0.877
Outgroup Identity (Dictionary)	0.83	0.786	0.958
Ingroup Solidarity			
Dictionary	0.602	0.676	0.854
BERT-NLI (finetuned)	0.748	0.748	0.942
Outgroup Hostility			
Dictionary	0.692	0.683	0.929
BERT-NLI (finetuned)	0.821	0.836	0.955

General

1 – To summarize the above methodological concerns, my biggest concern with the manuscript is that the group-based language categories were not carefully constructed, not based on theory, and were not validated in any way either to contrast against other categories they measured or to confirm they are measuring the psychological construct of interest. However, it is crucial to do so since it is the main construct of interest for the manuscript. The papers are replicating appear to do some version of content analysis and validation if they would like to see an example of how to proceed.

- Thank you for raising this, and we agree that our initial construction of the ingroup solidarity and outgroup hostility constructs were imperfect. We have addressed this by applying an entirely new methodology which is able to capture these constructs very well (see also above). Interestingly, this new approach and the one we used in our original manuscript show the same results, despite relying on entirely different

assumptions, which we believe substantially strengthens our findings. Finally, we have now added a paragraph on pages 1 and 2 where we discuss the theoretical underpinnings of ingroup solidarity and outgroup hostility, primarily drawing on realistic group conflict theory and intergroup emotions theory.

2- The authors chose to study engagement to popular news headlines rather than messages by ordinary users or political elites. This choice constrains the results and the ability of the authors to compare it to other work.

First, in Study 2 the authors leave out Russia headlines due to propaganda, but is there any evidence that Ukraine national news groups are not engaging in propaganda strategies? Whether in a corrupt sense or just a response to the context, of course war time will result in popular headlines supporting the country vis-à-vis the war. The main concern is that war-time motives for propaganda type news headlines is a competing explanation for the results of Study 2 (as opposed to some group-level shift in attraction to group solidarity). Both could explain the results, but can the authors point to their preferred explanation over the other?

- Thank you! We wanted to offer one minor clarification: the reason we could not use the Russian data in Study 2 is because Twitter and Facebook were banned in Russia after the 2022 invasion. You are right that it is possible that engagement on Ukrainian social media was driven in part by the fact that Ukrainian news outlets started producing more pro-Ukrainian headlines after February 2022. To address this point, we have now run the same analyses on a dataset of Ukrainian geolocated Twitter data, which includes regular social media users as well as news producers. The results are very similar to our original analyses, which has reaffirmed our idea that ingroup solidarity became a more important driver of social media engagement post- February 2022 (see the discussion of the results from Study 3). Moreover, the proportion of ingroup solidarity and outgroup hostility after the invasion increased to roughly the same levels (around 30% in Facebook news sources, see Figure S10 in the SI), but only solidarity posts gained more engagement.

Second it is important to highlight further that because the authors chose to study popular news headlines, it significantly limits the variation in language, especially intense emotional language that is used compared to ordinary users who may be having conflicts on the platform. The authors mention this briefly on pg. 4 but it should be highlighted when making a claims about how the study does or does not replicate previous results. If anything, it extends rather than replicates previous results in a new contexts of news headlines. This means the authors should more clearly highlight that their studies are about news headlines in the abstract and title. The paper is not one about social media engagement in general, it is about news headline engagement which restricts the claims the authors can make.

- Thank you, this is a really good point. We chose to study news headlines on social media because Rathje et al (the study we aimed to replicate originally) did so. To capture the variation in social media context we also collected and analysed the geolocated tweets from Ukraine (the new Study 3), where we found a similar effect; this shows that our results generalise well to non-news content.

3 – Related: the authors make a needlessly generalized claim in the abstract: “Our results suggest that signaling solidarity with one’s in-group may be more important than displaying negativity about out-groups during periods of intense conflict.” The statement should include a caveat that the study was about popular news headlines and that the findings are specific to Ukraine / Russia.

- Thank you, and you are right that it is better to hedge our conclusions; we do note, however, that we have now added an additional dataset of Twitter posts (Study 3) which allows us to generalise our findings beyond merely content posted by news media. We have now changed the wording in the abstract as follows: “Our results suggest that, at least in the context of the Russo-Ukrainian war, signaling solidarity with one’s ingroup may be more important than expressing negativity about outgroups during a period of intense intergroup conflict.”

4 – The authors’ main conclusion is that group-based language engagement is subject to exogenous shocks / context. This is interesting but it also appears that the moral-emotional category had the most consistent results across both texts. The authors should highlight this finding since they are comparing several categories of language.

- Thank you for mentioning this! Moral-emotional language does seem to be a consistent predictor across datasets and we have now highlighted that on pages 4, 5 and 7 of the manuscript (Study 1 - results and Discussion).

I also thought there is room to better highlight the inconsistent negative affect findings to a greater extent, because several recent studies have made claims about how negative headlines always draw more engagement. For instance in NHB the study “Negativity drives online news consumption” by Van Bavel et al.

- This is a great point, thank you! We have highlighted that on pages 4 , 5 and of 7 the new manuscript.

Conclusion

The authors should be commended for the cross-cultural comparison and extension of previous work in a war context. I think there is a version of this paper that is a nice contribution to the psychology of news headlines engagement, but it is not there yet.

- Thank you for these comments! We agree that the initial version could have improved, which we have now done. We look forward to your comments and really appreciate your time and effort.

Reviewer #3

Review of NCOMMS-23-01909. This manuscript describes associations between the content of Facebook and Twitter posts made by Pro-Ukrainian and Pro-Russian media outlets, and engagement with these posts, in the context of the Ukraine-Russia war. Prior findings that moral-emotional words are associated with more engagement are replicated. Results for other classes of words are more varied across platforms and country. Furthermore, changes over time in engagement with posts containing “ingroup solidarity” and “outgroup hostility” words are described. The authors frame these findings in terms of affective polarization and interpret the findings as advancing the understanding of inter-group dynamics.

Studying these associations in the context of the Ukraine-Russia conflict is interesting, as it provides a unique documentation of the correlates of full-blown international conflict as it occurs, and because I agree that most of the literature in this area is too US-centric. We indeed need more evidence from other social and cultural settings. As such, I welcome this manuscript. However I think there is too little scientific advancement offered in this manuscript to be published in Nature Communications.

- Thank you kindly for your very constructive comments. We have now made substantial changes to our manuscript, both methodological and theoretical, to strengthen both its contribution and our confidence in our conclusions. We have built entirely new language classifiers that not only offer a major methodological advance in psychological research, but also add substantial validity to our findings. In addition, we have added additional data from Twitter (not limited to news content only, but rather Twitter posts geolocated to Ukraine during the relevant time period), which further adds to the scope and robustness of our study. Using various methodological approaches rooted in entirely different assumptions, the conclusion that ingroup solidarity became the best predictor of engagement on Russian social media.

1. My key concern is that the manuscript essentially repeats the descriptive approach of prior work (enhanced by introducing two new classes of words, see validity concerns below), but that it continues to leave the meaning of the observed patterns unexplained. This would not be a problem if the patterns were unambiguous, or if the patterns had clear meaning in terms of understanding human psychology/behavior in the context of inter-group relations. Unfortunately neither applies to the current manuscript. E.g., the substantial between-platform and between-country variation (e.g., the outgroup words in Pro-Russian media in Fig 2) calls the generalizability of these findings into question, and makes it unclear what these findings really mean. Are we looking at artefacts of social media algorithms, or true human interaction patterns?

- Thank you for pointing this out! We agree that this is a valid concern. To address this issue, we have now used an entirely different, classifier-based approach to investigate the dynamics of engagement on Ukrainian social media (including a new dataset of Twitter data geolocated to Ukraine, and not limited to only news content). Using a classifier we built that can identify language related to ingroup solidarity and outgroup hostility (for the theoretical justification of these constructs, see Materials

and Methods), we find that our initial findings are extremely stable and robust: ingroup solidarity became by far the most important predictor of engagement on Ukrainian social media (not only Twitter but also Facebook) after the 2022 full-scale invasion. Furthermore, we have now clarified the interpretation of our findings in the context of intergroup emotions theory and realistic group conflict theory (see page 7). However, independent of whether the patterns we observe are due to human psychology or social media algorithms (or both), these findings add to our knowledge of what drives engagement on social media and challenge the traditional narrative centered around outgroup animosity.

2. More insight into why certain posts invite more or less engagement could help here, but the methodological approach is too limited for this. Essentially it is word counting, leaving the meaning of these words and engagement with them unexplained. E.g., what does engagement with a post by a pro-Ukrainian outlet with an outgroup reference such as ‘Moscow’ ‘Kremlin’ and ‘Putin’ reflect? In the manuscript these words are interpreted as identity language, but I have difficulty seeing how mentioning the capital of another country helps define or build an identity. My hunch is that these words are likely to occur in informational posts about politics (we are looking at posts by news outlets, after all), so retweeting/sharing these posts could simply serve to spread information about current and important developments in international relations. Extending that logic, it makes sense that the Ukrainian public would keep a close eye on what was happening in Moscow in the run-up to the invasion, explaining why outgroup-related words would invite more engagement (Fig 2A) – without this having any bearing on identity. While combining classes of words (eg the positive x outgroup and negative x outgroup interactions) might shed some more light on what the word counts actually reflect (spreading negative rumor, saying positive things, etc), there is no way of distinguishing between interpretations based on the current evidence.

- Thank you for this, and we fully agree that our initial methodological approach was too limited in the ways you point out. In our initial manuscript, we followed the approach laid out by Rathje et al. (2021) as closely as possible; here, the authors used a dictionary-based approach where outgroup and ingroup language was measured in the way you describe (i.e., not by focusing explicitly on words that reflect solidarity or hostility, but rather descriptive terms, e.g., “Republican” or “Democrat”). To mitigate this, we initially created additional dictionaries that more explicitly reflect solidarity and hostility within the context of group identity in Ukraine (before and after the full-scale invasion). However, we agree that this approach has substantial limitations. For this reason, we have used an entirely new, classifier-based approach. We built two classifiers from scratch that can identify ingroup solidarity and outgroup hostility, fit for purpose for this study. This model performs really well (see SI Table S14 and S15), and the findings remain highly robust: we show (with confidence, in our opinion) that identity-based language, and especially language reflecting ingroup solidarity, becomes the most important predictor of social media engagement only after the outbreak of the 2022 full-scale invasion. Because this finding is so robust to variations in methodology, statistical approaches, datasets (not only for content from news outlets but also from regular social media users), and social media platform

type, we are confident that this is a true finding. We now explain this in more detail in the Discussion section.

3. In light of a growing body of evidence documenting such correlations between words and engagements, I feel we need more context to these correlations to really get a handle on what they mean for affective polarization and understanding inter-group relations. For instance, the word counting approach might be complemented with a second methodological approach to provide a more in-depth interpretation of what the patterns actually mean and reflect in terms of human psychology, to merit publication in an outlet such as this (compare, e.g., a paper by Ashokkumar & Pennebaker, 10.1126/sciadv.abg7843, who combine their content analysis of reddit posts with survey data to better understand what references to different groups mean). Such an approach has the potential to really shed light on which motives drive engagement with ‘ingroup solidarity’ words: bolstering one’s online reputation, recasting intergroup relations, or simply sharing the available information about the ongoing war, which likely all contains references to one’s own and the other country?

- Thank you! We agree; as mentioned above, we have now added a classifier-based approach alongside the dictionary approach (along with a new dataset of geolocated Twitter data, which has allowed us to generalise our findings beyond what our original dataset was able to speak to). The fact that the results from both types of analyses (which rely on entirely different assumptions) are so similar has convinced us that we are looking at an important phenomenon that is robust enough to remain intact for many months after the 2022 invasion. Furthermore, we have done numerous additional robustness checks (for example looking at the base rates of the various language categories, see below), all of which have further strengthened our confidence in our findings. Finally, we have substantially expanded our discussion of the theoretical implications of our findings (see both Introduction and Discussion);

Aside from these concerns about contribution, I have some additional concerns about the validity of some aspects of the approach:

4. The measure of outgroup hostility and ingroup solidarity words relies purely on face validity, and no further validation steps were taken. While the dictionary approach is already quite a crude approach even when using a well-validated word list such as LIWC, there is no way of knowing whether the words that were identified to reflect solidarity or hostility actually map onto these underlying psychological constructs. Moreover, the words that are used here (to the extent that they are translated into English and reported in the ms.) seem to have meaning only in the context of a war. This is important because the same word list is applied prior to the invasion. But what does, e.g., an ‘ingroup solidarity’ word such as ‘hero’ mean at that point in time? And would people engage with it for the same reason as after the invasion? Given that the introduction of this concept is half of the contribution of this manuscript (the replication of prior findings in the Ukraine-Russia war being the other half), I think a more careful measurement approach is called for.

- Thank you, and we fully agree with this comment. This is why we have now added the classifier-based results, which do not rely on a rather subjective classification of

war-related terms, but rather are more generally applicable to all kinds of content (in Ukrainian and Russian) that might reflect ingroup solidarity or outgroup hostility. Doing so, we find that our initial conclusions hold up well (and, interestingly, that the dictionary-based approach turned out to be fairly accurate). We now mention this on Pages . See pages 7 and 8 (Materials and Methods) for an extensive explanation of the new methodology.

5. Related to this, I looked for, but could not find a report on the baserates of the various classes of words. Reporting baserates could help shed more light on, or even falsify the assumption implied in the previous paragraph: that all bellicose words are more frequent after the outbreak than before, simply because there is a war going on at that point.

- This is a good point; we now provide frequency plots of language categories over time (see below and Supplementary Information, Figure S10). Briefly put, we find that descriptive ingroup and outgroup language are fairly common before the invasion, but ingroup solidarity and outgroup hostility less so (around 10%). After the invasion, however, we see a substantial increase in the total volume of content reflecting both ingroup solidarity and outgroup hostility. This further adds to the validity of our findings; even though both anti-outgroup and pro-ingroup language are commonly found on Ukrainian social media after the 2022 invasion, only pro-ingroup language is a meaningful predictor of engagement.

6. Another validity concern is that, while the possibility of bots contributing to the dataset is mentioned in a few places, no attempt seems to have been taken to remove bot activity from the dataset. Since the influence of bots is explicitly mentioned in the interpretation of the strikingly ‘deviant’ engagement pattern on Pro-Russian Twitter (Fig 2A), I wonder to what extent other findings may actually be artefacts of bot activity.

- Thank you, and we agree; this was an oversight on our part. We estimate the risk that our findings are substantially influenced by bot activity to be rather low; our results are more or less identical across different platforms (Facebook and Twitter), which have varying levels of bot activity. We now note this on page 7. Unrelated to this specific issue, we also changed the main analysis methodology to account for the fact that multiple posts may come from the same news sources using mixed effects linear regressions with news sources as the random effect (instead of just linear regression, which we did originally). This improved methodology did not change any of our main findings significantly but did correct the issue you pointed out in Figure 2A.

7. A final concern is that I find the connection between the findings and the broader literature of affective polarization somewhat underdeveloped. In particular it is unclear whether the results are regarded as a cause, consequence, or symptom of affective polarization? What is the connection between the various classes of words (even if we go by the labels assigned to them) and the theory?

- Thank you! We agree that the initial manuscript was not sufficient in its discussion of the relevance of our findings for the broader literature; we now situate our study within the literature on intergroup emotions theory and realistic group conflict theory (see pages 1 and 2), which overlap with the literature on affective polarisation; in our manuscript, we are unable to speak to whether perceptions of individual social media users vis-a-vis other groups become more affectively polarised. Instead, we focus on

how language use, which can be a signal of how groups relate to one another, changes over time. We now clarify this on pages 1 and 2 as well as in the discussion (page 7).

8. As a minor sidenote, while I can see why the authors argue that a broader perspective on affective polarization than just based on partisanship is needed to fully understand it, I think this also makes the need for affective polarization as a separate construct superfluous. Affective polarization is, at least to my knowledge, a term introduced by political scientists to apply the broader concepts of ingroup favoritism and outgroup hostility derogation from Social Psychology / Social Identity Theory to the (US) political partisan context. By broadening affective polarization to include international relations, it seems to me that we are back at the original concepts from Social Identity Theory.

- Thank you; we agree with this, and we also agree that our results are not able to speak to affective polarization specifically. In the introductory section (see pages 1 and 2), we have now made changes to our original theoretical framework. Specifically, we now discuss two competing theoretical approaches (realistic conflict group theory and intergroup emotions theory); briefly put, our results speak strongly in favour of intergroup emotions theory, which predicts that ingroup rhetoric should become more salient during periods of intergroup conflict. Please also see our revised discussion on page 7.

In conclusion, while I think the manuscript indeed adds to the existing body of evidence on these findings, and while I welcome the broadening of the US-centric scope of this literature, I think the manuscript does not really advance the understanding of why these patterns occur. While I in principle like the research and welcome the findings, I think the methodological approach and interpretation are currently too limited for this outlet.

- Thank you again for your very insightful comments. We hope that our revisions and additional data, along with a substantial expansion of our theoretical contributions, have improved our initial manuscript. We look forward to any other comments you may have.

Reviewers' Comments:

Reviewer #1:

Remarks to the Author:

I would like to thank the authors for taking my earlier comments so seriously. I think the paper is greatly improved in terms of transparency – the words included in the dictionaries are now clearly available to readers – and because the paper now (largely – see below) shows the robustness of the results to using a more traditional state-of-the-art approached to classification using a supervised machine learning approach. I am now much more favorably inclined towards publishing the paper in Nature Communications.

I must admit it is still a bit difficult to get through the findings in the supplemental appendices – might I suggest that the authors use descriptive variable names in the tables instead of the names of the variables from their datasets? With this caveat in mind, I have identified four issues with the data/data interpretation that I think warrant addressing.

The first is the least serious, but in looking at Figure 4 – which is great for conveying the information in the paper! – I can't tell if in-group solidarity by the time we get to the summer of 2022 is actually statistically distinct from the period in the summer of 2021? Clearly everything else falls after the invasion so something is going on, but how do we know if the post war is what is different for ingroup solidarity vs. the period in the fourth quarter of 2021 being an anomaly on the low side? Is it possible that the lead up to the war saw a decrease in in-group solidarity engagement in an effort to ramp down engagement with talk that could have been seen as somehow making the possibility of war more likely? I'm probably just splitting hairs here and maybe I missed something in the appendix, but it might be worth addressing this point, i.e., that there is a sharp discontinuity in the trend of interest around late August/September 2021. If you could do statistical tests to show that the post-war trend is different from summer of 2021 trend, that would be helpful I think.

Second, Figure S7 looks very different from all the other figures – why is out-group hostility going up after the invasion? Is it possible the legend is just wrong?? (In the other figures, in-group solidarity is blue). Or is something very different happening on Twitter with the Dictionary method? If so, does that say something about the Dictionary Method? And if this is the case, then it needs to be addressed in the text of the paper, because the claim in the middle of page 9 that the results are largely the same across dictionaries and BERT-NLI models would not hold, nor would the claim in footnote 1 on p.4 of similar findings.

Something very strange is happening in Figure S10, Panel B1. While we see some spikes in other figures (especially around the invasion), this panel shows four sharp drops in pro-Ukrainian Twitter posts that are not simply the sort of thing you might see if there was data missing on a single day, because each time there is then a gradual increase until the next sharp drop. There seems to be no "real world" explanation that I can think of for why pro-Ukrainian tweets from news sources would drop in this way. I have a few thoughts. The first would be to check the data pipeline that was used to create this figure. The second would be to check the code for making the figure. The third is that maybe these tweets were being amplified by bots that were getting caught in crackdowns periodically and then replaced by new bots? I have no idea how you would be able to demonstrate this was the case, but I really think you need to address what is going on here – hopefully it is just a coding error?

Finally, and most seriously, in the text of the paper Study 3 is presented as corroborating the results of Study 2. The authors note that after the invasion ingroup solidarity predicts more engagement than outgroup hostility, which is reflected in Colum 2 of Table S3. However, my read of Table S3 is that this a *lower** level of engagement than in-group solidarity before the invasion (14% vs. 29% more than baseline). If I am not mistaken, this would then be a quite different finding from Study 2 and would cut against one of the main arguments of the paper that the

invasion increased engagement with in group solidarity posts? In fact, in-group is more than out-group before the invasion, and then both drop after the invasion, so there does not seem to even be in relative change between the two.

Reviewer #2:

Remarks to the Author:

My main concern with the original manuscript was with the ingroup/outgroup classification schemes and lack of transparency in statistical modeling. I thought the authors made an excellent effort to re-classify ingroup/outgroup affective language, showed their new methods has increased precision and replicated the results. The test of assumptions in statistical models is also now available and looks reasonable. Finally, the authors improved the precision of conclusions drawn in the abstract and discussion. I thank the authors for their efforts and think the manuscript is much improved. I have no further comments.

Reviewer #3:

Remarks to the Author:

Review of NCOMMS-23-01909A, a revised manuscript, I was reviewer 3 previously. The main changes are an improved method to classify posts, an additional dataset, and more theoretical elaboration of the author's thinking and interpretations.

Although I believe that the first two changes have substantially improved the methodological side of the research (but note that I am not an expert on supervised machine learning techniques, so please rely on other reviewers for a definitive assessment), I'm afraid my view on the contribution made in the manuscript hasn't changed: I continue to think that the topic is interesting and societally relevant, but that the methodological approach is fundamentally too limited to provide a strong conclusion about the research question.

1. It remains unclear what engagement with this content reflects in terms of human psychology. In your response to my original point comment on this issue (#3 in my original review), you mention the new classifier and that you ran many robustness checks. These methodological changes do not address the concern, however. Let me illustrate with the following statement from your article:

"However, realistic group conflict theory suggests that both ingroup solidarity and outgroup hostility should increase in importance during intergroup conflict, but this is different from what we observed. Despite outgroup hostility becoming much more commonly expressed on Ukrainian Facebook and Twitter after the invasion as compared to before (see Figure S10), people did not engage with it much." (p. 7).

Thus, you describe two findings: after the invasion, (a) production of outgroup hostility language goes up and (b) outgroup hostility language does not invite more engagement. Based on this, you infer that RCT is "wrong".

However, this interpretation depends on whether RCT predicts that conflict increases the production of outgroup hostility (in which case RCT is supported), or an increase in engagement with outgroup hostility (in which case you are right). But in fact, RCT is silent on whether the animosity caused by conflict should increase the one, the other, or both. Moreover, the reduced engagement with these posts may reflect other motives, such as a motive to avoid exposing oneself or others to the atrocities of war, which are likely to be linked to stories about a war enemy.

When commenting on measurement validity (#4 in my original review), you also point to the new classifier method. But although your algorithmic coding of solidarity/hostility goes a long way

toward making the measurement more accurate and reliable, it does little to improve the connection between your measure and the underlying psychological concepts that you suggest it is related to.

Thus, a fundamental gap remains between the theory on human psychology and the noisy measurement based on ambiguous engagement on social media. Given that your research question is about these motives (ie hostility versus solidarity), I think any conclusion that you can draw based on these data alone, no matter how sophisticated the algorithms, will be tentative at best (see also comment #4 below).

2. To add more interpretational depth, you have elaborated more on the (social) psychological theories related to your research. However, it seems that you might have misinterpreted social identity theory and (intergroup) emotions theory.

On intergroup emotion theory, you write: “[...] intergroup emotions theory suggests that one’s emotional experiences depend on salient group memberships, their associated group norms, and the strength of one’s group identification. Intergroup emotions – and the knowledge that these emotions are shared with others – thus shape intergroup processes by strengthening ingroup identity”

You then continue by contrasting intergroup emotion theory with social identity theory.

2a. This quote misrepresents intergroup emotion theory because it reverses the causality. IET predicts that stronger identification with a group enhances the emotions experienced on behalf of that group (i.e., group-related concerns; see work by Smith, Mackie, Seger from the 00s), not the other way around. Your deduction that “In other words, intergroup emotions theory predicts that expressions of ingroup solidarity should become more important and popular than expressions of outgroup hostility after the outbreak of intergroup conflict.” is therefore false. In fact, all group-based emotions should be experienced more strongly after an outbreak of intergroup conflict, because competition increases social identity salience (see integrated threat theory), which makes people more emotionally responsive to all group-level concerns.

2b. Moreover, intergroup emotion theory does not contrast with social identity theory. It is an elaboration of the affective side of social identification (and social identity theory) and therefore closely connected, if not integral to it.

3. Throughout, more precise conceptual language is needed. In particular, behaviors are often confounded with motivations and emotions, e.g., “after the November 2015 Paris terrorist attacks, one study found no evidence of increased outgroup hostility, but instead observed a higher level of ingroup solidarity among French citizens” (p. 1)

Other examples of conceptual imprecision:

- ingroup favoritism and outgroup derogation are more commonly seen as ‘treatments’ (ie behaviors), than as ‘views’ – they are certainly not emotions.
- in setting up for the contribution, you write that “there is a dearth of research into whether group identity salience is stable over time, or whether how group identity is expressed is subject to external shocks.” (p. 2) But you actually fill neither gap, i.e., you don’t measure group identity salience nor identity expression.

4. While much of the manuscript seeks to make claims about human psychology that I feel are not warranted (see #1), the last paragraphs of the article are more explicitly written at the level of the findings: “Nevertheless, whether due to platform algorithms and moderation or human psychology (or both), our empirical findings suggest that expressions of ingroup solidarity were likely to receive considerably more engagement than outgroup hostility on Ukrainian social media during the first six months of the 2022 Russo-Ukrainian war”

This bit, and the conclusion that follows it (p. 8), are conclusions that believe are warranted based on the data. Any interpretations in terms of human psychology are tentative at best, and should be

reframed to better reflect this, as well as to consider alternative interpretations.

Minor:

5. I continue to find the terms “descriptive ingroup/outgroup language” confusing. In several places, you also call the same language “Ukrainian and Russian identity language”, which I find not only confusing, but also inappropriate because there are many reasons other than identity for mentioning one’s own or another country. E.g., on p.5 you write that “Ukrainian and Russian identity language became less predictive of engagement. Controlling for all else, descriptive outgroup words drastically dropped” (p. 5). By using the term “identity language”, it also becomes an interpretational issue, because this statement can only be true if there is no identity-related language other than the descriptive ingroup/outgroup words.

I realize that this is how Rathje used these terms, but I would encourage you to be critical of their approach and adjust their terminology to improve accuracy and prevent confusion. I personally think “ingroup/outgroup mention” might be more appropriate terms.

6. Fig 3A is referred to from Study 2, but the caption says it belongs to Study 1.?

Dear reviewers and editors,

Thank you so much for your highly insightful and helpful comments. We appreciate the time and effort put into re-reading our updated manuscript.

We have now made substantial changes to the second version, most notably by (1) changing the terminology of “descriptive ingroup/outgroup language” to “ingroup/outgroup mentions,” (2) correcting accidental mislabeled legends and points of confusion, and, most importantly, (3) substantially clarifying the theoretical foundation of our work grounded in testing the predictions of Intergroup Emotions Theory that “research on emotions toward the in-group and out-group in conflict situations offers the potential to test the hypothesis that in-group positivity drives conflict processes to a greater extent than out-group negativity” (Mackie & Smith, 2018, p. 27).

We believe that our computational approach of combining social media data with precise natural language models offers unique benefits such as time granularity and external validity and is complementary to other methods like survey studies, especially since operationalizing social identity and intergroup emotions theories on digital and social media is a new emerging area. We would like to note that our theoretical position, stemming from the predictions of Intergroup Emotions Theory, now completely coheres with the measurement constructs. We have also paid extra attention to ensuring that the fact that our findings are correlational and not experimental is evident throughout the paper.

We have responded to each of your comments, highlighted in yellow below. We hope that our manuscript is now ready to be accepted for publication in *Nature Communications*, and we are very grateful for your comments, which have greatly improved this study.

Warm regards,

The authors

Mackie, D. M., & Smith, E. R. (2018). Intergroup Emotions Theory: Production, Regulation, and Modification of Group-Based Emotions. In *Advances in Experimental Social Psychology* (Vol. 58, pp. 1–69). Elsevier.
<https://doi.org/10.1016/bs.aesp.2018.03.001>

Reviewer #1 (Remarks to the Author):

I would like to thank the authors for taking my earlier comments so seriously. I think the paper is greatly improved in terms of transparency – the words included in the dictionaries are now clearly available to readers – and because the paper now (largely – see below) shows the robustness of the results to using a more traditional state-of-the-art approach to classification using a supervised machine learning approach. I am now much more favorably inclined towards publishing the paper in Nature Communications.

I must admit it is still a bit difficult to get through the findings in the supplemental appendices – might I suggest that the authors use descriptive variable names in the tables instead of the names of the variables from their datasets? With this caveat in mind, I have identified four issues with the data/data interpretation that I think warrant addressing.

- Thank you! We appreciate the suggestion, and we agree; we have now changed the variable names in the supplement to be more descriptive and clear. In addition, after a suggestion by Reviewer 3, we have changed the “descriptive ingroup language” and “descriptive outgroup language” to “ingroup mentions” and “outgroup mentions”, respectively. This, we believe, is a clearer description of the concepts that these two terms refer to.

The first is the least serious, but in looking at Figure 4 – which is great for conveying the information in the paper! – I can't tell if in-group solidarity by the time we get to the summer of 2022 is actually statistically distinct from the period in the summer of 2021? Clearly everything else falls after the invasion so something is going on, but how do we know if the post war is what is different for ingroup solidarity vs. the period in the fourth quarter of 2021 being an anomaly on the low side? Is it possible that the lead up to the war saw a decrease in in-group solidarity engagement in an effort to ramp down engagement with talk that could have been seen as somehow making the possibility of war more likely? I'm probably just splitting hairs here and maybe I missed something in the appendix, but it might be worth addressing this point, i.e., that there is a sharp discontinuity in the trend of interest around late August/September 2021. If you could do statistical tests to show that the post-war trend is different from summer of 2021 trend, that would be helpful I think.

- Thank you for pointing this out. It is completely expected that the period from the end of August to the beginning of September in Ukraine would have higher engagement with ingroup solidarity content because that is when the celebrations of Ukrainian independence take place (Ukrainian Independence Day is the 24th of August). This is a highly salient event for Ukrainian ingroup solidarity, which we would expect to increase engagement with solidarity content independent of any conflict. We have added a sentence explaining this to the manuscript in the legend to Figure 4 (page 6): “The rise in importance of ingroup solidarity around end of August 2021 coincides with the

celebrations of the Ukrainian Independence Day on August 24 – a highly identity-salient event supposed to elicit solidarity.”

Second, Figure S7 looks very different from all the other figures – why is out-group hostility going up after the invasion? Is it possible the legend is just wrong?? (In the other figures, in-group solidarity is blue). Or is something very different happening on Twitter with the Dictionary method? If so, does that say something about the Dictionary Method? And if this is the case, then it needs to be addressed in the text of the paper, because the claim in the middle of page 9 that the results are largely the same across dictionaries and BERT-NLI models would not hold, nor would the claim in footnote 1 on p.4 of similar findings.

- You are right – the legend in Figure S7 is wrong: ingroup solidarity should be in light blue (as in all the other figures), showing the same effect as in all the other figures and methods. The legend has now been fixed (see below).

Something very strange is happening in Figure S10, Panel B1. While we see some spikes in other figures (especially around the invasion), this panel shows four sharp drops in pro-Ukrainian Twitter posts that are not simply the sort of thing you might see if there was data missing on a single day, because each time there is then a gradual increase until the next sharp drop. There seems to be no “real world” explanation that I can think of for why pro-Ukrainian tweets from news sources would drop in this way. I have a few thoughts. The first would be to check the data pipeline that was used to create this figure. The second would be to check the code for making the figure. The third is that maybe these tweets were being amplified by bots that were getting caught in crackdowns periodically and then replaced by new bots? I have no idea how you would be able to demonstrate this was the case, but I really think you need to address what is going on here – hopefully it is just a coding error?

- Thank you. We agree that the graph looks a bit odd; this is because of an unavoidable limitation of our data collection method for Twitter for this study in particular. We collected our news source data from Twitter using the non-academic API in four tranches (August 2021, January 2022, June 2022, and September 2022), as was stated in the manuscript methods section (page 8):

Twitter news sources data was collected at four timepoints (August, 2021, and January, June, and September, 2022) by retrieving the 3200 most recent posts for each account using the R package “rtweet”.

This means that there is less data available just after data collection dates, as that data is the “oldest” at that point and falls outside of the rate limit of 3200 posts per page. For example, in January 2022, we collected data between August 2021 and January 2022. Because the API first retrieves the news tweets, we get 3200 tweets closest to January 2022, and if the page has posted more than that since August, some of the content does not get returned. Moreover, with older Twitter data, some of the posts might have gotten deleted or blocked. This leads to the phenomenon as shown in the graph. This could theoretically be mitigated by recollecting the data from the entire time period at once and re-running the analyses. However, this is not possible due to Twitter’s recent API changes, which means that we would have to pay a rather exorbitant fee for a much lower data volume. Therefore, our alternative mitigation procedure is twofold: 1) we compare the Twitter data to the Facebook data (which doesn’t have this issue), which both show highly similar proportions of our variables of interest (the four different identity language categories), suggesting that the collected data is sufficiently representative (see Figure S10, compare Figures A2 and B2); if these patterns are so consistent across platforms, we deem it unlikely that the discrepancy in volume in Twitter data has a meaningful influence on our findings, and 2) we collected the data for Study 3 (geolocated Twitter data) in one sitting using the academic API (before Twitter’s recent API changes), and therefore this dataset does not suffer from the same limitations (see Figure S10, Panel C1). Our findings here hold up well, although the effect is weaker (likely because that dataset includes *all* Ukrainian Twitter original posts, not just those related to news; see below). In sum, we appreciate your pointing this out, and ideally, we would recollect some of our data to triple-check our findings, but this is not possible at present. However, we are confident that our mitigation procedures are sufficient, and we are not concerned that this issue impacts the robustness of our findings. We have also added a line in this revision to the methods section after the description of the data collection procedure (page 8) saying: “(see the temporal distribution of the data in the SI Figure S10).”

Finally, and most seriously, in the text of the paper Study 3 is presented as corroborating the results of Study 2. The authors note that after the invasion ingroup solidarity predicts more engagement than outgroup hostility, which is reflected in Column 2 of Table S3. However, my read of Table S3 is that this is a *lower* level of engagement than in-group solidarity before the invasion (14% vs. 29% more than baseline). If I am not mistaken, this would then be a quite different finding from Study 2 and would cut against one of the main arguments of the paper that the invasion increased engagement with in-group solidarity posts? In fact, in-group is more than out-group before the invasion, and then both drop after the invasion, so there does not seem to even be a relative change between the two.

- Thank you for bringing this up; the explanation for this was an omission on our part, which we have now clarified. Firstly, the main claim we are making in this study is that ingroup solidarity drives engagement on social media during intergroup conflict more than outgroup hostility, not that there is more engagement with ingroup solidarity after the beginning of the conflict than before it. So the period before the invasion is not relevant here. Secondly, the post-level RoBERTa classifier of pro-Ukrainian content used in Study 3 was trained and evaluated on data from after the invasion exclusively (30,000 posts that are very different from posts occurring before the invasion and mostly talk explicitly about the war) and therefore does not necessarily generalize to the posts from before the invasion. This was originally explained on page 6 of the manuscript:

Unlike the previous two studies, where we used the orientation of the account of origin to classify posts, here we used a fine-tuned RoBERTa model to classify each post as either pro-Ukrainian or pro-Russian independent of the account that posted it. **We then only analyzed the posts identified as pro-Ukrainian and posted after the invasion (as the pro-Ukrainian classifier was trained on data exclusively from after the invasion).**

We have now explained this in Footnote 3 (page 7) in the manuscript and in the supplement (in the description of Tables S3 and S7):

The RoBERTa classifier used in Study 3 to label posts as pro-Ukrainian or pro-Russian was trained and evaluated exclusively on the data from after the invasion and thus does not necessarily generalize to the prior period. Therefore, we cannot make any claims concerning the predictive power of these four variables for Twitter engagement for the period before the invasion. However, we provide this data for completeness (see SI Tables S3 and S7).

Reviewer #2 (Remarks to the Author):

My main concern with the original manuscript was with the ingroup/outgroup classification schemes and lack of transparency in statistical modeling. I thought the authors made an excellent effort to re-classify ingroup/outgroup affective language, showed their new methods has increased precision and replicated the results. The test of assumptions in statistical models is also now available and looks reasonable. Finally, the authors improved the precision of conclusions drawn in the abstract and discussion. I thank the authors for their efforts and think the manuscript is much improved. I have no further comments.

- Thank you so much for your input and very useful comments. We really appreciate it.

Reviewer #3 (Remarks to the Author):

Review of NCOMMS-23-01909A, a revised manuscript, I was reviewer 3 previously. The main changes are an improved method to classify posts, an additional dataset, and more theoretical elaboration of the author's thinking and interpretations.

Although I believe that the first two changes have substantially improved the methodological side of the research (but note that I am not an expert on supervised machine learning techniques, so please rely on other reviewers for a definitive assessment), I'm afraid my view on the contribution made in the manuscript hasn't changed: I continue to think that the topic is interesting and societally relevant, but that the methodological approach is fundamentally too limited to provide a strong conclusion about the research question.

- Thank you for your highly insightful comments, which motivated us to look critically at our original manuscript and the theoretical underpinnings of our findings. Our methodological approach of combining social media data with language models to detect the fundamental intergroup emotions, particularly in the context of operationalizing Social Identity Theory (SIT) and Intergroup Emotions Theory (IET) on digital and social media, is an emerging area that presents novel challenges but also opportunities to get more external validity as compared to traditional lab studies. The ways in which we operationalize and measure the IET are different to how you might ask people about this in a survey, but they also avoid the fundamental challenges of conducting surveys with participants who reside in areas of active military conflict (Moss et al., 2019). We view the computational and survey-based approaches as complementary, and our theoretical position (as refined in the new version of the manuscript and elaborated below) now completely coheres with the measurement constructs proposed in SIT and IET research. We hope that these two approaches will together help advance the field's understanding of identity-based emotional expressions during intergroup conflict.
- That said, we certainly agree that the theoretical contributions of this work could have been phrased in a more robust manner in the original manuscript, which we have now done. You were entirely correct that realistic group conflict theory is not a useful framework that helps us better understand identity expressions during intergroup conflict. We also completely agree that our original phrasing suggested that intergroup emotions theory is separate from social identity theory, whereas it should be seen as an extension of it. As we explain in more detail below, we have now revisited the literature on intergroup emotions theory. Mackie & Smith (2018, p. 27) have argued the following:

From the preponderance of evidence, we suggest that positive emotions toward the in-group are more likely to drive intergroup bias and discrimination, compared to negative emotions toward the

out-group. Importantly, the absence of positive emotions is clearly distinguishable (conceptually and empirically) from the presence of negative emotions, meaning that in-group positivity and out-group negativity can be independently measured. Thus, **research on emotions toward the in-group and out-group in conflict situations offers the potential to test the hypothesis that in-group positivity drives conflict processes to a greater extent than out-group negativity.**

In the previous versions of our manuscript, our phrasing was indeed imprecise in that it was not clear that what we are looking at are *emotional identity expressions* during intergroup conflict, as outlined by Mackie & Smith (2018) in the above quote. We conceptualize ingroup solidarity and outgroup hostility in this manner, i.e., as intergroup emotions of positivity towards the ingroup and negativity towards the outgroup (Mackie & Smith, 2018), in our case on social media. We operationalized these conceptual definitions – which we used to label posts and train the classifier in our manuscript – the following way (see the “Ukraine Social Identity Code Book.pdf” on our OSF page):

- **Solidarity with Ukraine or Ukrainians:** The quote is expressing solidarity, liking, or unity of Ukraine or Ukrainians, for example it praises Ukraine, mentions Ukrainians as competent, good people or Ukraine as a great, strong or united nation.
- **Hostility towards Russia or Russians:** The quote is expressing hostility, derogation or disliking of Russia or Russians, for example it criticises Russia, mentions Russians as incompetent, immoral people or the Russian Federation as a bad, weak or failing nation.

Our intention here was to capture specifically emotional expressions of ingroup and outgroup identity. Our study is thus a test of the above-proposed hypothesis by Mackie & Smith, although we fully agree that this was not clear enough in our original wording. We have now made several substantial changes in our introduction and discussion sections to reflect these points. Please see the following changes (page 2):

Given the key role of emotions in shaping the diffusion of (news) content on social media^{3, 4}, our investigation adopts the intergroup emotions theory framework^{18, 28}. This theory extends social identity theory into the realm of emotions, suggesting that salient group identifications can influence emotions whereby events (such as a violent conflict) are appraised in terms of their implications for the ingroup. Shared intergroup emotions can potentially influence intergroup processes by reinforcing ingroup identity¹⁸. Notably, intergroup emotions theorists suggest that “research on emotions toward the in-group and out-group in conflict situations offers the potential to test the hypothesis that in-group positivity drives conflict

processes to a greater extent than out-group negativity” [18, p. 27]. To test this theoretical prediction, we examine the importance of ingroup solidarity versus outgroup hostility in the context of the 2022 full-scale invasion of Ukraine by the Russian Federation.

1. It remains unclear what engagement with this content reflects in terms of human psychology. In your response to my original point comment on this issue (#3 in my original review), you mention the new classifier and that you ran many robustness checks. These methodological changes do not address the concern, however. Let me illustrate with the following statement from your article:

“However, realistic group conflict theory suggests that both ingroup solidarity and outgroup hostility should increase in importance during intergroup conflict, but this is different from what we observed. Despite outgroup hostility becoming much more commonly expressed on Ukrainian Facebook and Twitter after the invasion as compared to before (see Figure S10), people did not engage with it much.” (p. 7). Thus, you describe two findings: after the invasion, (a) production of outgroup hostility language goes up and (b) outgroup hostility language does not invite more engagement. Based on this, you infer that RCT is “wrong”.

However, this interpretation depends on whether RCT predicts that conflict increases the production of outgroup hostility (in which case RCT is supported), or an increase in engagement with outgroup hostility (in which case you are right). But in fact, RCT is silent on whether the animosity caused by conflict should increase the one, the other, or both. Moreover, the reduced engagement with these posts may reflect other motives, such as a motive to avoid exposing oneself or others to the atrocities of war, which are likely to be linked to stories about a war enemy.

When commenting on measurement validity (#4 in my original review), you also point to the new classifier method. But although your algorithmic coding of solidarity/hostility goes a long way toward making the measurement more accurate and reliable, it does little to improve the connection between your measure and the underlying psychological concepts that you suggest it is related to.

Thus, a fundamental gap remains between the theory on human psychology and the noisy measurement based on ambiguous engagement on social media. Given that your research question is about these motives (ie hostility versus solidarity), I think any conclusion that you can draw based on these data alone, no matter how sophisticated the algorithms, will be tentative at best (see also comment #4 below).

- Thank you for bringing up this excellent point. We agree that the previous version of our manuscript did not describe our work’s theoretical foundations well enough. We have now revised our introduction and discussion sections to stand on firm theoretical footing. You are also right that RCT is not the right

framework, and we've removed mentions of it from our manuscript. Instead, we have made it more clear that our investigation leverages the concepts of ingroup solidarity and outgroup hostility, as described by Mackie & Smith (2018), and tests the robustness of their predictions regarding different emotion-based expressions of group identity during intergroup conflict on social media (see the above comment). To this end, we have rewritten paragraphs 2-4 in the introduction (pp1-2):

Social identity theory⁷ postulates that people derive part of their self-identity from important memberships in social groups. Once individuals categorize themselves into a social group, they often adopt the identity of that group and start comparing themselves (the "ingroup") to others (the "outgroup"). Although social identity theory has been studied for many decades⁸, much less is known about the psychological processes that contribute to social identification around political issues on digital and social media^{9, 10}. In fact, the influence of specific social identity processes online appears to hinge on the intergroup context, as indicated by varying polarization effects observed in social media studies across different countries¹¹⁻¹⁴. Moreover, research about the role of media and polarization in non-US contexts is lacking, with as much as 86% of studies coming from the US¹⁵. While some studies have investigated how identity-based motivations are expressed on social media^{4, 5, 16, 17}, there exists a notable research gap in directly comparing the fundamental intergroup emotions of outgroup hostility and ingroup solidarity¹⁸ as predictors of online engagement⁶. Bridging this gap is crucial for understanding the social and emotional processes driving intergroup conflict on social media and formulating interventions to address polarization. In short, to date, conditions under which identity expressions on social media are driven by ingroup solidarity versus outgroup hostility remain underspecified⁶.

In the context of the present-day United States, negative outgroup cues seem more effective than positive ingroup cues in persuading partisans^{5, 19, 20}, although, in general, there appears to be a consensus among scholars of social identity theory that ingroup-favoring motivations should matter more than outgroup-derogation^{21, 22}. Moreover, many studies suggest that expressions of ingroup solidarity might be more important than outgroup hostility after severe shocks or in the presence of outgroup threats^{16, 23-27}. For instance, after the November 2015 Paris terrorist attacks, one study²⁵ found no evidence of an increase in opinions indicating outgroup hostility but instead observed a higher level of views consistent with ingroup solidarity

among French citizens. Garcia and Rimé¹⁶ observed a similar effect in the expressions of positive and negative emotions on Twitter after these attacks.

These findings raise important questions about the conditions under which different group identity expressions gain traction on social media during major news events. Given the key role of emotions in shaping the diffusion of (news) content on social media^{3, 4}, our investigation adopts the intergroup emotions theory framework^{18, 28}. This theory extends social identity theory into the realm of emotions, suggesting that salient group identifications can influence emotions whereby events (such as a violent conflict) are appraised in terms of their implications for the ingroup. Shared intergroup emotions can potentially influence intergroup processes by reinforcing ingroup identity¹⁸. Notably, intergroup emotions theorists suggest that “research on emotions toward the in-group and out-group in conflict situations offers the potential to test the hypothesis that in-group positivity drives conflict processes to a greater extent than out-group negativity” [18, p. 27]. To test this theoretical prediction, we examine the importance of ingroup solidarity versus outgroup hostility in the context of the 2022 full-scale invasion of Ukraine by the Russian Federation.

- Moreover, the type of behavioral data and quantitative analysis presented in our manuscript offers complimentary evidence to other social psychological approaches, providing a higher level of external validity at the expense of a lower degree of experimental control. Computational investigations of SIT and IET enabled by the recent advances in machine learning and artificial intelligence are integral to bringing these theories into the 21st century, connecting them with the emerging technology that influences our daily lives, and have been recognized in numerous psychological papers as beneficial to our understanding of the psychological concepts operationalized. For instance, papers like Garcia and Rimé (2019, *Psychological Science*), Brady et al. (2017, *PNAS*), Rathje et al. (2021, *PNAS*), and others underscore how big data analysis of psychological constructs such as emotions, morality, and social identity can be done through language analysis of social media data. For instance, David Garcia and Bernard Rimé (2019, p. 617) studied the “social sharing of emotion” on Twitter after the 2015 Paris terror attacks. In their words, they “predict participation in the social sharing of emotion following a traumatic event to enhance social belonging, shared beliefs, positive affect, and prosocial attitudes”. We extend this framework to social identity theory (via intergroup emotions theory): our argument is that expressions of positive and negative emotions can be related to ingroups and outgroups, and that a traumatic event (in our case, the outbreak of the full-scale invasion) influences not only how these emotions are expressed, but

also how message receivers react to such expressions. Like Garcia and Rimé (2019), we use social media data, which has the advantage of being naturalistic in the sense that it reflects voluntary expressions not elicited by the researcher (with the downside that there is less experimental control), as a measure of how emotions are expressed. However, unlike Garcia and Rimé (2019), we do not only classify the use of emotions on social media (in their case, negative and positive affect, as well as anxiety, anger, and sadness; in our case, more specific emotions directed at ingroups and outgroups) but also receivers' *reactions* to these emotions being expressed, in the form of engagement (likes, retweets, et cetera). Moreover, Brady et al. (2017, *PNAS*, p. 7313) used a methodology similar to ours with dictionaries and social media data to study what psychological variables help the diffusion of moral ideas in networks, offering "insights into how people are exposed to moral and political ideas through social networks, thus expanding models of social influence and group polarization as people become increasingly immersed in social media networks" and clarifying how "moral emotions contribute to moral contagion" (p. 7316). These findings helped psychologists better understand human behavior in the modern world and refine cognitive models (e.g., Van Bavel & Pereira, 2018, *Trends in Cognitive Sciences*).

- However, we have paid a lot of attention to ensure that it is clear throughout the paper that our findings are correlational and not experimental (e.g., see discussion on page 7):

We offer several interpretations of our findings. Intergroup emotions theory^{18, 28} suggests that one's personal emotional experiences depend on the salient group membership and its associated norms. Based on empirical evidence, Mackie et al.¹⁸ hypothesized that "ingroup positivity drives conflict processes to a greater extent than outgroup negativity" [18, p.27]. In line with this hypothesis, we find **correlational evidence** that ingroup solidarity drives more engagement than outgroup hostility on social media during conflict. **However, we cannot make any claims about the existence or direction of any causal effect, as further experimental studies are required to isolate the processes.**

2. To add more interpretational depth, you have elaborated more on the (social) psychological theories related to your research. However, it seems that you might have misinterpreted social identity theory and (intergroup) emotions theory.

On intergroup emotion theory, you write: "[...] intergroup emotions theory suggests that one's emotional experiences depend on salient group memberships, their associated group norms, and the strength of one's group identification. Intergroup emotions – and the knowledge that these emotions are shared with others – thus shape intergroup processes by strengthening ingroup identity"

You then continue by contrasting intergroup emotion theory with social identity theory.

2a. This quote misrepresents intergroup emotion theory because it reverses the causality. IET predicts that stronger identification with a group enhances the emotions experienced on behalf of that group (i.e., group-related concerns; see work by Smith, Mackie, Seger from the 00s), not the other way around. Your deduction that “In other words, intergroup emotions theory predicts that expressions of ingroup solidarity should become more important and popular than expressions of outgroup hostility after the outbreak of intergroup conflict.” is therefore false. In fact, all group-based emotions should be experienced more strongly after an outbreak of intergroup conflict, because competition increases social identity salience (see integrated threat theory), which makes people more emotionally responsive to all group-level concerns.

- Thank you for pointing this out; your comments have motivated us to dig deeper into the theoretical underpinnings of our research. We agree that our original manuscript was not careful enough in its discussion of social identity literature. We have, therefore, substantially expanded and clarified our literature review (see pages 1-2 and the comments above). With respect to your point about IET: The predictions we tried to articulate were based on the question coming from intergroup emotions theory of whether ingroup or outgroup targeted emotions are more important during conflict (see page 26 of Mackie & Smith, 2018). As mentioned above, on page 27, Mackie & Smith (2018) argue that:

From the preponderance of evidence, we suggest that positive emotions toward the in-group are more likely to drive intergroup bias and discrimination, compared to negative emotions toward the out-group. Importantly, the absence of positive emotions is clearly distinguishable (conceptually and empirically) from the presence of negative emotions, meaning that in-group positivity and out-group negativity can be independently measured. Thus, research on emotions toward the in-group and out-group in conflict situations offers the potential to test the hypothesis that in-group positivity drives conflict processes to a greater extent than out-group negativity.

This last sentence is exactly what our work is trying to accomplish. We have now carefully rephrased our manuscript (see above) to more clearly reflect this point.

2b. Moreover, intergroup emotion theory does not contrast with social identity theory. It is an elaboration of the affective side of social identification (and social identity theory) and therefore closely connected, if not integral to it.

- Thank you for bringing this to our attention; we completely agree and have updated the manuscript to reflect this more clearly (see paragraph 1 on page 2):

Given the key role of emotions in shaping the diffusion of (news) content on social media^{3, 4}, our investigation adopts the intergroup emotions theory framework^{18, 28}. This theory extends social identity theory into the realm of emotions, suggesting that salient group identifications can influence emotions whereby events (such as a violent conflict) are appraised in terms of their implications for the ingroup. Shared intergroup emotions can potentially influence intergroup processes by reinforcing ingroup identity¹⁸. Notably, intergroup emotions theorists suggest that “research on emotions toward the in-group and out-group in conflict situations offers the potential to test the hypothesis that in-group positivity drives conflict processes to a greater extent than out-group negativity” [18, p. 27]. To test this theoretical prediction, we examine the importance of ingroup solidarity versus outgroup hostility in the context of the 2022 full-scale invasion of Ukraine by the Russian Federation.

3. Throughout, more precise conceptual language is needed. In particular, behaviors are often confounded with motivations and emotions, e.g., “after the November 2015 Paris terrorist attacks, one study found no evidence of increased outgroup hostility, but instead observed a higher level of ingroup solidarity among French citizens” (p. 1)

Other examples of conceptual imprecision:

- ingroup favoritism and outgroup derogation are more commonly seen as ‘treatments’ (ie behaviors), than as ‘views’ – they are certainly not emotions.
- in setting up for the contribution, you write that “there is a dearth of research into whether group identity salience is stable over time, or whether how group identity is expressed is subject to external shocks.” (p. 2) But you actually fill neither gap, i.e., you don’t measure group identity salience nor identity expression.

- Thank you for this comment. Indeed, although our work directly studies social media engagement with expressions of ingroup solidarity and outgroup hostility, the original manuscript lacked precision in using these terms, which we used to sometimes refer to behaviors and sometimes to emotions. SIT mostly uses the terms ingroup favoritism and outgroup derogation to refer to behaviors (Tajfel & Turner, 1979). IET, as you mentioned, extends these concepts to the domain of affect, where people usually refer to them as ingroup love and outgroup hate (Mackie et al., 2000). In our work, we opted for using ingroup solidarity and outgroup hostility as the labels for these emotions when measuring the emotional expressions of group identity with our classifiers because these terms, we believe, better capture the complexities of a society in conflict. However, we also originally used the

same terms to refer to the behaviors predicted by SIT. In order to attain more conceptual precision and clarity, we have now updated the main text to have a clear distinction between the two throughout, using ingroup favoritism/outgroup derogation to refer to behaviors and ingroup solidarity/outgroup hostility to refer to emotions (see especially pages 1-2 pasted above and highlighted in the manuscript).

- We have also elaborated on the quote you highlighted to make the language more precise (page 1): “For instance, after the November 2015 Paris terrorist attacks, one study²⁵ found no evidence of an increase in **opinions indicating outgroup hostility** but instead observed a higher level of **views consistent with ingroup solidarity** among French citizens. Garcia and Rimé¹⁶ observed a similar effect **in the expressions of positive and negative emotions on Twitter** after these attacks.”

4. While much of the manuscript seeks to make claims about human psychology that I feel are not warranted (see #1), the last paragraphs of the article are more explicitly written at the level of the findings: “Nevertheless, whether due to platform algorithms and moderation or human psychology (or both), our empirical findings suggest that expressions of ingroup solidarity were likely to receive considerably more engagement than outgroup hostility on Ukrainian social media during the first six months of the 2022 Russo-Ukrainian war”

This bit, and the conclusion that follows it (p. 8), are conclusions that believe are warranted based on the data. Any interpretations in terms of human psychology are tentative at best, and should be reframed to better reflect this, as well as to consider alternative interpretations.

- Thank you. We agree that the psychological interpretation in our original manuscript was unsatisfactory, and we have reframed the paper to more clearly articulate both the psychological framework we are using (SIT and IET) and the extent to which we can interpret our findings (see our comments above in response to your point 1). The combination of social media data and precise natural language models offers a unique extent of external validity and empirical granularity that we could not have obtained with a traditional lab study or a survey, especially when the participants are involved in active conflict. We have also elaborated on the tentativeness of any conclusion that can be drawn from such data and added alternative explanations in the discussion section (see pages 7 and 8).

We offer several interpretations of our findings. Intergroup emotions theory^{18, 28} suggests that one’s personal emotional experiences depend on the salient group membership and its associated norms. Based on empirical evidence, Mackie et al.¹⁸ hypothesized that “ingroup positivity drives conflict processes to a greater extent than outgroup negativity” [18, p.27]. In line

with this hypothesis, we find **correlational evidence** that ingroup solidarity drives more engagement than outgroup hostility on social media during conflict. **However, we cannot make any claims about the existence or direction of any causal effect, as further experimental studies are required to isolate the processes.** Nonetheless, the increase in engagement with pro-Ukrainian content relative to anti-Russian content coincided with a marked decrease in positive sentiments towards Russians among Ukrainians (see Fig. 1). Despite outgroup hostility becoming much more commonplace on Ukrainian Facebook and Twitter after the invasion as compared to before (see SI Figure S10)⁴, people did not engage with it much. Specifically, while mentioning the outgroup serves as a good predictor of social media engagement (i.e., group members talking about other groups, but not necessarily in a negative manner) when no outright intergroup conflict is present, outgroup hostility predicts social media engagement far less well (especially during intergroup conflict). This contradicts expectations based on Rathje et al.⁵ and prior findings on dehumanization in the media^{49–51}. Since the United States is a highly polarized country, where polarization is primarily driven by negative partisanship^{19, 20}, our results suggest that what types of identity language drive online engagement may be context-dependent and not universal across cultures⁵²

On the other hand, it is possible that especially hostile content is algorithmically discouraged by social media platforms, but if this is true, then it would have to be the case for both Twitter and Facebook to a similar extent. Even though our study explored the dynamics of engagement over time, making the effects less dependent on specific news events, we cannot know if and when the underlying algorithms have changed and how that influenced the results. For instance, a recent experimental study suggests that Facebook's algorithmic feed might have led people in the US to see more content classified as uncivil or containing slurs during the 2020 elections but had little effect on measures of issue and affective polarization⁵³. However, Facebook and Twitter have been reported to demote and ban Ukrainian pages posting content about the war⁵⁴. Another possibility is that Ukrainian social media users are reluctant to share explicitly derogatory content aimed at the Russian people, as opposed to the Russian government³⁰, or that people are afraid to express hostility for personal safety reasons. However, this does not explain why outgroup hostility predicted a lot of "Angry" reactions after the start of the invasion but became gradually less predictive

starting in May 2022 (see SI Figure S9). At the same time, expressions of ingroup solidarity remained important throughout, potentially sustaining the group's efforts toward victory¹⁸.

Minor:

5. I continue to find the terms “descriptive ingroup/outgroup language” confusing. In several places, you also call the same language “Ukrainian and Russian identity language”, which I find not only confusing, but also inappropriate because there are many reasons other than identity for mentioning one's own or another country. E.g., on p.5 you write that “Ukrainian and Russian identity language became less predictive of engagement. Controlling for all else, descriptive outgroup words drastically dropped” (p. 5). By using the term “identity language”, it also becomes an interpretational issue, because this statement can only be true if there is no identity-related language other than the descriptive ingroup/outgroup words.

I realize that this is how Rathje used these terms, but I would encourage you to be critical of their approach and adjust their terminology to improve accuracy and prevent confusion. I personally think “ingroup/outgroup mention” might be more appropriate terms.

- Thank you for this great suggestion! We completely agree that ingroup/outgroup mentions make more sense than descriptive ingroup/outgroup and have updated the manuscript, figures, and the SI to use this terminology.

6. Fig 3A is referred to from Study 2, but the caption says it belongs to Study 1.?

- Thank you; you are right. We have now fixed the legend to be more descriptive (page 5): “Figure 3 (A) Ingroup and outgroup mentions and negative, positive, and moral emotional language as predictors of engagement after the invasion on pro-Ukrainian Facebook and Twitter.”

Reviewers' Comments:

Reviewer #1:

Remarks to the Author:

I thank the authors again for the seriousness with which they addressed my previous comments. In regard to the four points I made after the last round of revisions:

1) Figure 4 – Thanks for adding the clarification around August 2021 – I think that will be useful information for the reader in interpreting the Figure. Also, I agree that is a good explanation for the surge in 2021, which in turn makes the claim of an increase post-invasion even more evident.

2) Figure S7 – ok! Glad to hear it was just an error with the legend and that it has now been corrected.

3) Figure S10 – Ok, I understand your explanation – as well as why you can not go back and rehydrate given the new policies at Twitter. That being said, the figure is still kind of misleading in the sense that it purports to show changes in pro-Ukrainian tweets over time, when in reality what it is really showing is an artifact of your data collection process (as you helpfully explain in the current response to reviewers). I see three potential options here: (i) just drop the figure b/c it does not really display what you want it to display; (ii): keep the figure, but add a much lengthier explanation in the caption along the lines of what you put in your response to reviewers so that it is explicit why we see the patterns we do; or (iii) maybe redo the figure just to show the small window around the invasion, a period of time during which there does appear to be a discontinuity in the data that is independent of the overall trend due to the data collection effects? To be clear, I would still recommend adding the lengthier explanation of the data collection effect if you adopt option (iii) so that the reader is fully informed.

4) Study 3 as a corroboration of Study 2: Thanks for the explanation in the response to reviewers. I think in my previous review I was responding to the line on p.6 that reads “We find that the results from Study 2 are replicated in a non-news-specific dataset”, which could be read as including the changes from pre-invasion to post-invasion as well. You are absolutely correct, though, that in the remainder of the paragraph you do focus only on the relative engagement across different types of engagement post-invasion, and I appreciate you taking the time to emphasize that in your response. While I think adding Footnote 3 was helpful, it still gets at this point in an indirect way – by discussing what can be estimated and what can not be estimated – as opposed to explicitly just stating that Study 3 is not intended to be used to replicate the findings regarding changes from pre-invasion to post-invasion. So my recommendation would be to add one more sentence in the discussion of Study 3 on p.6 that just states that explicitly – I think that would just reduce the chance of someone reading it the way I did originally.

I am now happy to recommend publication of the article, although would suggest that the authors consider my suggestion for points 3&4.

Reviewer #2:

None

Reviewer #3:

Remarks to the Author:

Review of NCOMMS-23-01909B, a revised manuscript. I was reviewer 3 previously.

I will first compliment you on the revision. The theorizing is more coherent and accurate than in the previous versions. Moreover, I appreciate the changes in terminology and the clearer separation of theorizing and empirical findings.

Although I remain of the opinion that psychological inferences from behavior alone are problematic, I will not raise that issue again. I seem to be alone in considering this an unfixable issue, and will defer to the editor here.

But given the nature of this outlet, and given that you seek to investigate social psychological constructs, I believe the review of the relevant social psychological literature needs to meet a high threshold – not only to contextualize your findings, but also to avoid misunderstandings in research that will build on yours.

1. In this regard, in framing your contribution, you heavily emphasize a quote from Mackie & Smith (2018), which is useful but unfortunately, at the same time, misinterpreted. In this statement, M&S argue that ingroup-love emotions are more important CAUSES of conflict than outgroup-hate emotions. That is, they argue that the more negative behavior toward other groups compared to our own group (e.g., discrimination) is more commonly caused by ingroup-love than outgroup hate. However, your research does not focus on causes of conflict. Instead, the conflict is a given (i.e., an external event), solidarity and hostility are only studied as consequences of the conflict. Moreover your focus is on the “victims” of conflict, and not the initiators/perpetrators, as M&S focus on.

I believe an accurate application of the IET quote to the situation would therefore be to investigate whether expressions of ingroup solidarity and/or outgroup hostility spike in pro-Russian media just before the invasion?

Given that cause and effect are reversed in your research compared to the theorizing, the question is: what DO your findings tell us about relations between conflict, solidarity and hostility? I think you could argue that you investigate the time course of (engagement with) ingroup solidarity and outgroup hostility both pre-outbreak, during outbreak, and post-outbreak. I think you need to broaden your review and more clearly separate causes and consequences of conflict to build a relevant theoretical framework. The M&S quote is most relevant pre-conflict; the rally around the flag bits from previous versions (which are more applicable to the post-outbreak situation) were more useful for the during/post-outbreak stages.

2. In a related vein, you write that “there appears to be a consensus among scholars of social identity theory that ingroup-favoring motivations should matter more than outgroup-derogation”. This is a very limited reading of the literature. For instance, there is an extensive literature on dehumanization that shows that outgroup-derogation “matters” a lot. E.g., the atrocities in WW2-Germany are commonly understood as driven by (morality-based) outgroup derogation/dehumanization. The cited sources don’t support the argument either. For instance, although Weisel & Böhm (2015) indeed show that ingroup love is a more common cause of ‘discrimination’, except when the intergroup relation is based on morality, where outgroup hate is more potent. I think the Russia-Ukraine conflict easily qualifies as a morality-based conflict. ‘Common’ isn’t the same as important. I think it would be helpful to discuss conditions under which solidarity or derogation may surface.

3. Although conceptual clarity has improved, there is still ambiguity around key concepts.

3a. As noted before, ingroup solidarity and outgroup hostility are not emotions by any common conceptualization of emotions. Love and hate are. I think the labels used by a classifier should not stand in the way of accurate terminology.

3b. Engagement is construed in many different ways in the manuscript and the rebuttal. Already in the first two points of your rebuttal you construe the main DV, engagement, as both “identity-based emotional expressions” and “identity expressions”. Later, you also refer to a quote from Garcia and Rimé, who construe it (I think more appropriately) as “the social sharing of emotion”. Note that these are completely different things, e.g., an (identity-based) emotion expression (e.g., anger) is something different than an identity expression (e.g., that I am a social psychologist),

which is again different from socially sharing emotions (e.g., interacting about our shared anger to build a common worldview, invite social support, etc). Interestingly, on p. 7, you write "Despite outgroup hostility becoming much more commonplace on Ukrainian Facebook and Twitter after the invasion as compared to before (see SI Figure S10), people did not engage with it much" which nicely illustrates the difference. Although people were EXPRESSING more hostility, other people did not ENGAGE with it. Thus, engagement is distinct from expression of emotions. In your study, it is primarily the media outlets who are expressing the emotions, which makes it unclear what the users are doing. It would be helpful to either construe the DV in a clear and coherent way, or refrain from construing it at all and interpreting the findings in terms of different construals.

4. Two more specific points about your contribution statement:

- "less is known about the psychological processes that contribute to social identification around political issues on digital and social media" (p.1) although the statement is true, you don't fill the gap, i.e., you don't provide insight into social identification as an outcome. You could argue that your study correlates of social identity.
- "Bridging this gap is crucial for understanding the social and emotional processes driving intergroup conflict on social media and formulating interventions to address polarization." (p. 1)
(a) the conflict is on the ground, not on social media; (b) the war is not polarization, unless we take polarization to be a synonym for all types of conflict; (c) I have difficulty seeing how your findings may help address polarization

I would really like to be more supportive of this work. But as a social psychologist, I am worried how a publication in this outlet may introduce enduring misunderstandings about important theories from my field. I hope you will see the constructive intent of my comments.

Dear reviewers,

Thank you kindly for your very helpful comments. We have done our best to address all of them in detail. Please find our responses highlighted in yellow below.

Warm regards,

The authors

Reviewer #1 (Remarks to the Author):

I thank the authors again for the seriousness with which they addressed my previous comments. In regard to the four points I made after the last round of revisions:

- Thank you so much for all the time and effort you put into your comments on this and previous revisions! Your comments have made the paper orders of magnitude better. We have implemented your new suggestions to the best of our abilities and elaborate point by point below.

1) Figure 4 – Thanks for adding the clarification around August 2021 – I think that will be useful information for the reader in interpreting the Figure. Also, I agree that is a good explanation for the surge in 2021, which in turn makes the claim of an increase post-invasion even more evident.

- Thank you for your help on this!

2) Figure S7 – ok! Glad to hear it was just an error with the legend and that it has now been corrected.

- Thank you for spotting this error.

3) Figure S10 – Ok, I understand your explanation – as well as why you can not go back and rehydrate given the new policies at Twitter. That being said, the figure is still kind of misleading in the sense that it purports to show changes in pro-Ukrainian tweets over time, when in reality what it is really showing is an artifact of your data collection process (as you helpfully explain in the current response to reviewers). I see three potential options here: (i) just drop the figure b/c it does not really display what you want it to display; (ii): keep the figure, but add a much lengthier explanation in the caption along the lines of what you put in your response to reviewers so that it is explicit why we see the patterns we do; or (iii) maybe redo the figure just to show the small window around the invasion, a period of time during which there does appear to be a discontinuity in the data that is independent of the overall trend due to the data collection effects? To be clear, I would still recommend adding the lengthier explanation of the data collection effect if you adopt option (iii) so that the reader is fully informed.

- Thank you, we completely agree. We originally included the figure for transparency so that the reader is informed about when the majority of the data is from. We were not trying to show the changes before and after the invasion with this figure. We choose to follow your advice number (ii) and keep the figure while adding the longer explanation below it so that we are as transparent as possible. Thank you again for this suggestion. The explanation:
 - **“Note for Supplementary Fig. 10 B1:** We collected our news source data from Twitter using the non-academic API in four tranches (August 2021, January 2022, June 2022, and September 2022) by retrieving the 3200 most recent posts for each account using the R package “rtweet”.

This means that there is less data available just after data collection dates, as that data is the “oldest” at that point and falls outside of the rate limit of 3200 posts per page. For example, in January 2022, we collected data between August 2021 and January 2022. Because the API first retrieves the news tweets, we get 3200 tweets closest to January 2022, and if the page has posted more than that since August, some of the content does not get returned. Both the Twitter data and the Facebook data (which doesn't have this issue) show highly similar proportions of our variables of interest (the four different identity language categories), suggesting that the collected data is sufficiently representative (compare Supplementary Figures A2 and B2). It is, therefore, unlikely that the discrepancy in volume in Twitter data has a meaningful influence on our findings.”

4) Study 3 as a corroboration of Study 2: Thanks for the explanation in the response to reviewers. I think in my previous review I was responding to the line on p.6 that reads “We find that the results from Study 2 are replicated in a non-news-specific dataset”, which could be read as including the changes from pre-invasion to post-invasion as well. You are absolutely correct, though, that in the remainder of the paragraph you do focus only on the relative engagement across different types of engagement post-invasion, and I appreciate you taking the time to emphasize that in your response. While I think adding Footnote 3 was helpful, it still gets at this point in an indirect way – by discussing what can be estimated and what can not be estimated – as opposed to explicitly just stating that Study 3 is not intended to be used to replicate the findings regarding changes from pre-invasion to post-invasion. So my recommendation would be to add one more sentence in the discussion of Study 3 on p.6 that just states that explicitly – I think that would just reduce the chance of someone reading it the way I did originally.

- Thank you for your suggestion! We agree, and we added a sentence on page 6 to emphasize this (in violet color in the updated manuscript):
 - “Thus, this study is not intended to replicate the findings regarding changes from pre-invasion to post-invasion, but only the engagement patterns post-invasion.”

I am now happy to recommend publication of the article, although would suggest that the authors consider my suggestion for points 3&4.

- Thank you so much for the thoughtful suggestions you have given throughout; they truly made the manuscript much better. We are very happy with your recommendation and have implemented your suggestions for points 3&4 in the revised manuscript. Thank you again for your time!

Reviewer #3 (Remarks to the Author):

Review of NCOMMS-23-01909B, a revised manuscript. I was reviewer 3 previously.

I will first compliment you on the revision. The theorizing is more coherent and accurate than in the previous versions. Moreover, I appreciate the changes in terminology and the clearer separation of theorizing and empirical findings.

- Thank you so much for your time and comments on the theoretical framework underpinning our work. Your comments on our previous versions of the manuscript have drastically helped improve the conceptual clarity and contribution of the paper, and we are extremely grateful to you for that.

Although I remain of the opinion that psychological inferences from behavior alone are problematic, I will not raise that issue again. I seem to be alone in considering this an unfixable issue, and will defer to the editor here.

- Thank you for this; it's an interesting discussion for sure.

But given the nature of this outlet, and given that you seek to investigate social psychological constructs, I believe the review of the relevant social psychological literature needs to meet a high threshold – not only to contextualize your findings, but also to avoid misunderstandings in research that will build on yours.

- As a team comprised of both computational and social psychologists ourselves, we completely agree with the need to be precise and thorough in the theoretical underpinnings and have implemented your suggestions to the best extent possible in the revised manuscript. However, we do want to respectfully note that *Nature Communications* is a broad interdisciplinary journal aimed at a general science audience, and as such less concerned with field-specific jargon and scholarly debates of primary interest to a specific sub-discipline of psychology. As you and R2 note, our key contribution is using unique behavioral data to explore how group-based expressions co-vary with a major intergroup conflict situation. We also want to make sure we are not engaging in *HARKing* (hypothesizing after the results are known). We are not suggesting you are implying we do so at all, but we are cognizant of the need to not continually adjust our theoretical framing to better fit the results. In fact, we strongly think the framing is accurate based on our understanding of the literature. We also did not have strong predictions about the role of ingroup solidarity versus outgroup hate, we merely leveraged the quote from Mackie and Smith (2018) expressing the need to independently measure these constructs and test whether ingroup love

can trump outgroup hate during intergroup conflict processes as previous research has suggested, which is exactly what we do in the paper.

However, to leave no room for doubt and to get further clarification on the quote in question, we contacted colleagues Diane Mackie and Eliot Smith and highlight their responses (Personal Communication, March 13) where relevant to help inform the discussion below. We have done our best to address any remaining points of contention in this final revision. Indeed, we detail below the changes made and have highlighted them in violet in the main manuscript. We realize that we may not reach full consensus on this issue, but we hope that our responses and revisions sufficiently assuage your remaining concerns about the theoretical framing of the paper and that our findings indeed make a useful contribution to IET. We would like to thank you very warmly for all your help and feedback.

1. In this regard, in framing your contribution, you heavily emphasize a quote from Mackie & Smith (2018), which is useful but unfortunately, at the same time, misinterpreted. In this statement, M&S argue that ingroup-love emotions are more important CAUSES of conflict than outgroup-hate emotions. That is, they argue that the more negative behavior toward other groups compared to our own group (e.g., discrimination) is more commonly caused by ingroup-love than outgroup hate. However, your research does not focus on causes of conflict. Instead, the conflict is a given (i.e., an external event), solidarity and hostility are only studied as consequences of the conflict. Moreover your focus is on the “victims” of conflict, and not the initiators/perpetrators, as M&S focus on.

- We must respectfully disagree with the reviewer here. Mackie & Smith (2018) do not only focus on *causes* of conflict. In fact, they refer to “conflict situations” (p. 27) broadly construed in their review and they do not separate out the theoretical predictions of Intergroup Emotions Theory in this way either. On the contrary, in section (3.4) that directly follows the quote we use in the paper, M&S speak of the “*multidirectionality* of processes” noting how emotions can drive behavior but *also* how intergroup conflict can in turn impact social identification and emotions, specifically appealing to a dynamic reciprocal understanding of causes and effects (i.e. not a unilateral distinction between causes and consequences).

It turns out that M&S do not argue that ingroup-love emotions are more important causes of conflict than outgroup-hate emotions at all. When we queried them about the quote in question, Mackie & Smith clarified over e-mail to us that

“The idea of ingroup positivity being more influential than outgroup negativity comes from Brewer and Greenwald & Pettigrew and is not a hypothesis derived

from IET. We mentioned this issue in our Advances chapter mainly to point out that measuring emotions toward the ingroup and the outgroup offers a clear method to address this question. We mainly wanted to point out that measuring intergroup emotions seems like an ideal way to test that hypothesis. (And as of 2017 or so when we wrote the chapter, very little research had taken that approach.)”

Indeed, the actual quote in the paper merely mentions that “Research on emotions toward the in-group and out-group in conflict situations offers the potential to test the hypothesis that in-group positivity drives conflict processes to a greater extent than out-group negativity”.

This is exactly what we do in the paper: we test the hypothesis that in-group positivity drives conflict processes to a greater extent than out-group negativity.

Of course, one instantiation of IET certainly covers a situation in which intergroup emotions are the cause of discriminative behavior but to suggest that this is the only interpretation of IET is not correct. For instance, the example M&S provide to exemplify their quote covers a complex conflict situation in New Zealand where cross-sectional data is used to predict support for conflict-relevant policies - a situation that doesn't disentangle and cannot speak to causes vs effects. Moreover, earlier work by Mackie, Maitner, & Smith (2009) clearly refers to conflict processes in general (from conflict initiation to reconciliation) and specifically mentions that “ongoing structural intergroup relations elicit group-based emotions” (p.1) - which closely reflects our current case study.

Lastly, in our reading, it is also not correct that M&S exclusively focus on perpetrators, there's plenty of work where emotions expressed by the victims are also examined (e.g., see Leonard, Mackie, & Smith, 2011) so IET can also apply to emotions experienced by the “victims”. We acknowledge that we don't have data on the perpetrator (your next point), but we strongly think that M&S's reference to simultaneous causality and emotions experienced by either group across a range of conflict stages provides room for a much broader interpretation of IET. In fact, we now explicitly acknowledge (per the reviewers' concern) that the hypothesis could be investigated from the perspective of the perpetrators (Russia) but that our case study presents a novel opportunity to investigate engagement with in-group love over out-group hate among Ukrainians, which we now highlight as a useful theoretical contribution to IET (while also acknowledging in the discussion that future research should contrast these findings to pro-Russian media). We hope the reviewer agrees that IET is a broad

tent that can accommodate testing of its hypotheses across all conflict stages for multiple groups, which is certainly in line with M&S own views, as they write that our views are “consistent with the emphasis in our Advances paper on complex, multidirectional causation among emotions, behaviors, and the surrounding intergroup situation.”

I believe an accurate application of the IET quote to the situation would therefore be to investigate whether expressions of ingroup solidarity and/or outgroup hostility spike in pro-Russian media just before the invasion?

- This is an interesting point, but as mentioned above, we don't believe the quote is applied exclusively to perpetrators and causes (there's no suggestion of this in the chapter in which the quote appears and M&S confirmed over e-mail as well that IET is not committed to any such specific predictions), but it is a very interesting study suggestion from the reviewer nonetheless. However, as we mention in the manuscript on p. 3, Facebook and Twitter were banned in Russia shortly after the invasion, so the data would not be representative and we cannot retrieve it, especially now that Musk has closed Twitter's API/made it unaffordable for academic research. Accordingly, we assume that the reviewer meant pro-Russian media *within* Ukraine before the invasion. As mentioned in the paper (Study 1), outgroup mentions are the strongest predictor of engagement pre-invasion among pro-Russian news sources on Facebook. Because Study 1 is meant as a conceptual replication of Rathje et al. (2021)'s US study (before the 2022 invasion), we only include the dictionary analyses using the same categories as in their study, which are ingroup and outgroup mentions and negative, positive, and moral emotional affect. In this study, we indeed find that outgroup mentions generate the most engagement for both pro-Ukrainian and pro-Russian news sources, broadly replicating Rathje et al. (2021).

We then choose to only focus on pro-Ukrainian data for our time series analyses because most of the pro-Russian news sources stopped posting on Facebook and Twitter as the platforms were banned in Russia shortly after the invasion, making any comparison infeasible. We understand that the reviewer believes that the pre-invasion pattern re outgroup mentions is reversed from what IET would predict (i.e. higher engagement with ingroup content pre-invasion) but as mentioned by M&S, IET does actually *not* have specific predictions about ingroup love vs outgroup hate at any given time point and our contribution fills the research gap identified by M&S exactly by investigating how these processes unfold over time during a unique intergroup conflict. Also, as the war in Ukraine started more than 8 years before the 2022 invasion and both groups have not felt positively towards each other since 2014 (see Fig 1), talking about causes vs

consequences and aggressor versus victim is very complex. We further elaborate on this in response to your next point.

Given that cause and effect are reversed in your research compared to the theorizing, the question is: what DO your findings tell us about relations between conflict, solidarity and hostility? I think you could argue that you investigate the time course of (engagement with) ingroup solidarity and outgroup hostility both pre-outbreak, during outbreak, and post-outbreak. I think you need to broaden your review and more clearly separate causes and consequences of conflict to build a relevant theoretical framework. The M&S quote is most relevant pre-conflict; the rally around the flag bits from previous versions (which are more applicable to the post-outbreak situation) were more useful for the during/post-outbreak stages.

- Again, the M&S quote is not specific to any time point. IET recognizes the complexity of intergroup conflict (especially in non-Western contexts) and therefore (perhaps wisely) refrains from committing to any particular predictions in this regard. Now, M&S do offer some personal thoughts over e-mail partially consistent with the reviewers point that *“is useful to distinguish the initial stages of a conflict from full-fledged or open warfare”*. We agree but note that this is exactly what we have done in the manuscript all along: we compare engagement with identity expressions pre-conflict to the progression of fully-fledged warfare. The reason why we think the ingroup solidarity finding during the invasion is significant for IET is because we have to consider the complicated nature of the meaning of “pre-invasion” in this context. This is an ongoing intergroup conflict, and the escalation of violence is what we are looking at here as a break-point, but the initial cause of the conflict happened in 2014 when Russia annexed Crimea, if not earlier, so it is not really sensible to try to determine an obvious starting point as the “cause” of the conflict in our time series data. Like other intergroup conflicts that sometimes go back many decades (e.g., Israel-Palestine), the situation is complex. For example, was Israel the victim or the aggressor? Maybe initially the victim but now the aggressor? Or historically the aggressor all along? Or perhaps the victim all along if we go back to slavery in ancient Egypt? Unlike some conflicts (e.g., consistent unidirectional oppression of one minority by a majority group), these are not causal processes we can easily disentangle as both sides are involved in experiencing and expressing conflict and threat to identities. The most we can hope for in real-life behavioral examinations (as opposed to controlled social psych lab studies) is exactly what you state above (and broadly echoed by R2); “that we investigate the time course of (engagement with) ingroup solidarity and outgroup hostility both pre-outbreak, during outbreak, and post-outbreak.” However, there is no post-outbreak really (the war is ongoing and now in its 11th year, as most Ukrainians and

international organizations would consider it (e.g., Kyiv Independent; WSJ; DW), and the intergroup conflict started much earlier during the Russian Empire and throughout the USSR times (Plokhly, 2015). We find that on Ukrainian social media ingroup solidarity gets most engagement pre-invasion but even more so during conflict escalation when identity is highly salient (and the predictiveness of outgroup hostility drops to almost zero during this period). Perhaps in some ways, the reviewer could think of the invasion as the “cause” at least in terms of activating social identities and strong emotional expressions associated with discriminative intergroup behavior. Please also note that we are now less committed to attributing our finding as evidence for that specific quote and focus more on just exploring how our novel finding helps inform IET and the broader literature on ingroup love vs outgroup hate.

Plokhly, S. (2015). *The gates of Europe: A history of Ukraine*. Basic Books.

2. In a related vein, you write that “there appears to be a consensus among scholars of social identity theory that ingroup-favoring motivations should matter more than outgroup-derogation”. This is a very limited reading of the literature. For instance, there is an extensive literature on dehumanization that shows that outgroup-derogation “matters” a lot. E.g., the atrocities in WW2-Germany are commonly understood as driven by (morality-based) outgroup derogation/dehumanization. The cited sources don’t support the argument either. For instance, although Weisel & Böhm (2015) indeed show that ingroup love is a more common cause of ‘discrimination’, except when the intergroup relation is based on morality, where outgroup hate is more potent. I think the Russia-Ukraine conflict easily qualifies as a morality-based conflict. ‘Common’ isn’t the same as important. I think it would be helpful to discuss conditions under which solidarity or derogation may surface.

- Thank you for this suggestion, we agree and following M&S’s feedback, we also acknowledge that we perhaps came off too strong on claiming consensus. We have now re-phrased this as “some support” for the hypothesis, and we also agree that it is helpful to mention in the introduction the conditions under which solidarity or derogation are more likely to occur to add nuance to our literature review and discussion section. We have now done so. Please see below and p.1: “More generally, major conflicts such as World War II have generally been explained by focusing on outgroup derogation and dehumanization^{20, 21}.”

3. Although conceptual clarity has improved, there is still ambiguity around key concepts.

3a. As noted before, ingroup solidarity and outgroup hostility are not emotions by any common conceptualization of emotions. Love and hate are. I think the labels used by a classifier should not stand in the way of accurate terminology.

- Thank you, we explain the labels of emotions in the introduction such that “ingroup solidarity” is equivalent to “ingroup love” and “outgroup hostility” is equivalent to “outgroup hate” in our manuscript for clarity. Please see page 1:
 - “[...] there exists a notable research gap in directly comparing the fundamental intergroup emotions of outgroup hostility (or outgroup hate) and ingroup solidarity (or ingroup love)¹⁸ as predictors of online engagement⁶.”

3b. Engagement is construed in many different ways in the manuscript and the rebuttal. Already in the first two points of your rebuttal you construe the main DV, engagement, as both “identity-based emotional expressions” and “identity expressions”. Later, you also refer to a quote from Garcia and Rimé, who construe it (I think more appropriately) as “the social sharing of emotion”. Note that these are completely different things, e.g., an (identity-based) emotion expression (e.g., anger) is something different than an identity expression (e.g., that I am a social psychologist), which is again different from socially sharing emotions (e.g., interacting about our shared anger to build a common worldview, invite social support, etc). Interestingly, on p. 7, you write “Despite outgroup hostility becoming much more commonplace on Ukrainian Facebook and Twitter after the invasion as compared to before (see SI Figure S10), people did not engage with it much” which nicely illustrates the difference. Although people were EXPRESSING more hostility, other people did not ENGAGE with it. Thus, engagement is distinct from expression of emotions. In your study, it is primarily the media outlets who are expressing the emotions, which makes it unclear what the users are doing. It would be helpful to either construe the DV in a clear and coherent way, or refrain from construing it at all and interpreting the findings in terms of different construals.

- Thank you for bringing this consideration to our attention. Firstly, we would like to clarify that our dependent variable throughout our analyses is the log-transformed amount of engagement a post receives, where engagement is defined as the sum of all platform-specific reactions. Explicitly for Twitter this is: $DV = \log(\text{retweet} + \text{favorites} + 1)$. For Facebook: $DV = \log(\text{shares} + \text{likes} + \text{comments} + \text{angry} + \text{wow} + \text{haha} + \text{sad} + \text{love} + 1)$. Social media engagement has been used to study identity-based and emotion-based drivers of online virality (e.g., Brady et al. 2017; Rathje et al. 2021, but importantly not Garcia & Rimé, 2019, which only looks at the *usage* of different emotions in online discussions, not users’ engagement with them). In our study, following again Brady et al. (2017) and Rathje et al. (2021), we assume that engagement with posts containing various types of identity-based emotional expressions (e.g.,

ingroup solidarity, outgroup hostility) implies some level of agreement: a social media user who sees a post saying “Glory to Ukraine!” (for example) and clicks “retweet” or “love” can be reasonably assumed to agree with the content of this post. Similarly, if a user sees a post that says “Russian bastards bombed a village” and clicks “like”, they are assumed to express agreement with Russians being referred to as “bastards”.

- However, to your point, you are right that in the previous versions of our manuscript, we referred to our key *independent* variables of interest (ingroup solidarity and outgroup hostility, primarily), in different ways (e.g., “identity expressions” and “identity-based emotional expressions”), and we agree that this inconsistency leads to a lack of clarity. In line with your recommendation, we have therefore revised the descriptions of what these IVs mean throughout the manuscript. To be specific, we now refer to ingroup solidarity (or ingroup love) and outgroup hostility (or outgroup hate) as “expressions of intergroup emotions”, in line with Mackie & Smith (2018). Our DV (engagement) approximates user *agreement with* expressions of intergroup emotions, which is a measure of overall popularity or viral potential.

4. Two more specific points about your contribution statement:

- “less is known about the psychological processes that contribute to social identification around political issues on digital and social media” (p.1) although the statement is true, you don’t fill the gap, i.e., you don’t provide insight into social identification as an outcome. You could argue that you study correlates of social identity.

- Thank you for pointing this out. We have now changed the sentence to say (p,1): “Although social identity theory has been studied for many decades⁸, less is known about the correlates of social identity on digital and social media.”

- “Bridging this gap is crucial for understanding the social and emotional processes driving intergroup conflict on social media and formulating interventions to address polarization.” (p. 1) (a) the conflict is on the ground, not on social media; (b) the war is not polarization, unless we take polarization to be a synonym for all types of conflict; (c) I have difficulty seeing how your findings may help address polarization

- Thank you for your comments. We have done our best to address each of your points below.
- RE part (a): This is not correct: Russia has used social media to conduct foreign influence campaigns in Ukraine and the US, at least since the 2014 Crimea annexation and the 2016 US elections, respectively (e.g., see Lange-Ionatanishvili et al. (2015) and Boyte (2017) for Ukraine; US Senate Report (2019) and Eady (2023) for the US). In fact, the Russian annexation of Crimea is used as a prime example of so-called “Hybrid warfare” that involves the skillful

use of emerging technologies such as social media (Rácz, 2015). Valery Gerasimov, Chief of the General Staff of the Russian Armed Forces, also developed the so-called “Gerasimov Doctrine”, which explicitly considers the information space to be part of the (kinetic) battlefield. Moreover, media have always played a role in conflicts through, for instance, propaganda campaigns; there are whole scientific journals dedicated to examining the role of media in war, such as the *Journal of Information Warfare and Media*, *War & Conflict*. As a specific example: listeners of a radio station that urged violence were much more likely to engage in murder, perpetrating genocide toward the Tutsi minority in Rwanda (Yanagizawa-Drott, 1994). So to summarize, the conflict is playing out both on the ground and on social media.

- We have added this point in one sentence in the discussion section on p.7: “Our findings also have real-world implications, as modern warfare is waged in the digital space as well as on the kinetic battlefield. In particular, Russia has used social media to conduct foreign influence campaigns in Ukraine and the US, at least since the 2014 Crimea annexation and the 2016 US elections, respectively⁵². In fact, the Russian annexation of Crimea is one of the most significant examples of so-called “Hybrid warfare” that involves the skillful use of emerging technologies such as social media.^{31, 52} Moreover, our findings may indicate that the 2022 invasion provoked a further consolidation of Ukrainian national identity, in line with other research showing that the 2014 Euromaidan revolution boosted Ukrainian national (civic) identity.^{31, 36, 37”}
- RE part (b): In our understanding, war implies polarization to some degree because, when two groups are at war – which is “a state of armed conflict between two countries or groups” – it often correlates with a more negative perception, feelings and attitudes toward the other group. In our specific case, as Fig. 1 in the manuscript shows, “after the 2022 invasion, as few as 2% of Ukrainians and 23% of Russians expressed a positive attitude toward the other country, a dramatic decrease from 83% and 74% positive opinions just a decade earlier,³²⁻³⁴ indicating increased intergroup tensions and polarization.” (p.2).
- RE part (c): Influential previous studies such as Brady et al. (2017) and Rathje et al. (2021) have found that more hostile content (e.g., negative or outrageous posts and outgroup-related posts) tends to go viral, which has led some people to blame social media for polarization (e.g., Van Bavel et al., 2021). Our study presents a counter-example: we find that ingroup solidarity drives more engagement on social media than outgroup hostility during the conflict, while outgroup hostility does not drive engagement. Therefore, it shows that: (1) Social media CAN differentially propagate ingroup solidarity content and not hostile content. (2) The intergroup context surrounding the social media content (the

Ukrainian context, in our case) that is studied might have an effect on whether hostility or solidarity goes viral. (3) In the Ukrainian context of high polarization (see the point above), solidarity drives more engagement on social media, suggesting that it is not just hostility that matters for polarization (even though most of past literature on social media and polarization has focused on hostile content). Our findings are thus broadly in line with Brewer's (1999) argument that "Discrimination can be motivated solely by ingroup preference..." (p. 431), we note that Ukrainians are not just victims but discriminate against Russians too (again, as noted in the point above and Fig 1). Therefore, our study highlights the need for future research to create social media interventions that target ingroup solidarity instead of outgroup hostility to causally test how each type of content increases or decreases the levels of affective polarization experienced on these platforms.

- We have added this explanation in the manuscript on p.8:
"In the Ukrainian context of high polarization (see Fig.1), solidarity drives more engagement on social media, suggesting that it is not only negative emotions like outrage and hostility that matter, even though the negative emotions have been the primary focus of past studies^{4,5}. Therefore, our study highlights the need for future research to create social media interventions that independently target ingroup solidarity and outgroup hostility to causally test how each type of content increases or decreases the levels of affective polarization experienced on these platforms."

Rácz, A. (2015). *Russia's hybrid war in Ukraine: breaking the enemy's ability to resist*. Finnish Institute of International Affairs.

US Senate Report of the Select Committee on Intelligence United States Senate on Russian Active Measures Campaigns and Interference in the 2016 U.S. Election, Volume 2: Russia's Use of Social Media with Additional Views (2019).

Lange-Ionatamishvili, E., Svetoka, S., & Geers, K. (2015). Strategic communications and social media in the Russia Ukraine conflict. *Cyber war in perspective: Russian aggression against Ukraine*, 103-111.

Eady, G., Paskhalis, T., Zilinsky, J., Bonneau, R., Nagler, J., & Tucker, J. A. (2023). Exposure to the Russian Internet Research Agency foreign influence campaign on Twitter in the 2016 US election and its relationship to attitudes and voting behavior. *Nature communications*, 14(1), 62.

Boyte, K. J. (2017). An analysis of the social-media technology, tactics, and narratives used to control perception in the propaganda war over Ukraine. *Journal of Information Warfare*, 16(1), 88-111.

Van Bavel, J. J., Rathje, S., Harris, E., Robertson, C., & Sternisko, A. (2021). How social media shapes polarization. *Trends in Cognitive Sciences*, 25(11), 913-916.

I would really like to be more supportive of this work. But as a social psychologist, I am worried how a publication in this outlet may introduce enduring misunderstandings about important theories from my field. I hope you will see the constructive intent of my comments.

- Thank you so much for your constructive comments, we really appreciate them. We are also social psychologists and agree with you on the importance of terminology and coherent social theories. We have done our best in this version of the manuscript to address all of your concerns but it is possible that we continue to hold different views on the same literature. Nonetheless, we think our novel findings make a useful contribution to IET and we've made clear where they align and diverge from well-known psychological theories of intergroup conflict. We hope the reviewer can now support publication.

Reviewer #2's comments on rebuttal to other reviewers' concerns:

In the last round of review, I was satisfied with the authors response to my methodological concerns. For this round, I read very carefully Reviewer #3's concerns and the authors' response (responses to Reviewer #1 were good).

- Thank you so much for your time and effort in reviewing our updated manuscript! We really appreciate your previous methodological comments and the current comment on the social-psychological framing. Please see our response to reviewer 3 above.

I was not entirely convinced by the authors' response to R3, and the new framing of the paper as one that can push forward Intergroup Emotions Theory predictions does not land very well. The new edit states, "Research on emotions toward the in-group and out-group in conflict situations offers the potential to test the hypothesis that in-group positivity drives conflict processes to a greater extent than out-group negativity"

Ultimately, I think R3 is correct that the paper is not testing the idea that ingroup positivity drives conflict processes.

However, I think the paper can still be successfully framed as a cross-cultural exploration of how group-based expressions covary with a major conflict situation in Russia.

- Thank you for this comment! We have updated the framing of the paper to be more in line with your suggestion. Please see the violet changes in the introduction and the response to reviewer 3.

Given the framing that is very social identity heavy, I would also appreciate a citation of previous work that causally tested the impact of group identity salience on moral emotion expressions

<https://osf.io/preprints/osf/dgt6u>

- Thank you so much for this great suggestion! We have cited the paper in the introduction section.

Reviewers' Comments:

Reviewer #1:

Remarks to the Author:

I thank the authors again for the seriousness with which they addressed my previous comments. In regard to the four points I made after the last round of revisions:

1) Figure 4 – Thanks for adding the clarification around August 2021 – I think that will be useful information for the reader in interpreting the Figure. Also, I agree that is a good explanation for the surge in 2021, which in turn makes the claim of an increase post-invasion even more evident.

2) Figure S7 – ok! Glad to hear it was just an error with the legend and that it has now been corrected.

3) Figure S10 – Ok, I understand your explanation – as well as why you can not go back and rehydrate given the new policies at Twitter. That being said, the figure is still kind of misleading in the sense that it purports to show changes in pro-Ukrainian tweets over time, when in reality what it is really showing is an artifact of your data collection process (as you helpfully explain in the current response to reviewers). I see three potential options here: (i) just drop the figure b/c it does not really display what you want it to display; (ii): keep the figure, but add a much lengthier explanation in the caption along the lines of what you put in your response to reviewers so that it is explicit why we see the patterns we do; or (iii) maybe redo the figure just to show the small window around the invasion, a period of time during which there does appear to be a discontinuity in the data that is independent of the overall trend due to the data collection effects? To be clear, I would still recommend adding the lengthier explanation of the data collection effect if you adopt option (iii) so that the reader is fully informed.

4) Study 3 as a corroboration of Study 2: Thanks for the explanation in the response to reviewers. I think in my previous review I was responding to the line on p.6 that reads “We find that the results from Study 2 are replicated in a non-news-specific dataset”, which could be read as including the changes from pre-invasion to post-invasion as well. You are absolutely correct, though, that in the remainder of the paragraph you do focus only on the relative engagement across different types of engagement post-invasion, and I appreciate you taking the time to emphasize that in your response. While I think adding Footnote 3 was helpful, it still gets at this point in an indirect way – by discussing what can be estimated and what can not be estimated – as opposed to explicitly just stating that Study 3 is not intended to be used to replicate the findings regarding changes from pre-invasion to post-invasion. So my recommendation would be to add one more sentence in the discussion of Study 3 on p.6 that just states that explicitly – I think that would just reduce the chance of someone reading it the way I did originally.

I am now happy to recommend publication of the article, although would suggest that the authors consider my suggestion for points 3&4.

Reviewer #2:

None

Reviewer #3:

Remarks to the Author:

Review of NCOMMS-23-01909B, a revised manuscript. I was reviewer 3 previously.

I will first compliment you on the revision. The theorizing is more coherent and accurate than in the previous versions. Moreover, I appreciate the changes in terminology and the clearer separation of theorizing and empirical findings.

Although I remain of the opinion that psychological inferences from behavior alone are problematic, I will not raise that issue again. I seem to be alone in considering this an unfixable issue, and will defer to the editor here.

But given the nature of this outlet, and given that you seek to investigate social psychological constructs, I believe the review of the relevant social psychological literature needs to meet a high threshold – not only to contextualize your findings, but also to avoid misunderstandings in research that will build on yours.

1. In this regard, in framing your contribution, you heavily emphasize a quote from Mackie & Smith (2018), which is useful but unfortunately, at the same time, misinterpreted. In this statement, M&S argue that ingroup-love emotions are more important CAUSES of conflict than outgroup-hate emotions. That is, they argue that the more negative behavior toward other groups compared to our own group (e.g., discrimination) is more commonly caused by ingroup-love than outgroup hate. However, your research does not focus on causes of conflict. Instead, the conflict is a given (i.e., an external event), solidarity and hostility are only studied as consequences of the conflict. Moreover your focus is on the “victims” of conflict, and not the initiators/perpetrators, as M&S focus on.

I believe an accurate application of the IET quote to the situation would therefore be to investigate whether expressions of ingroup solidarity and/or outgroup hostility spike in pro-Russian media just before the invasion?

Given that cause and effect are reversed in your research compared to the theorizing, the question is: what DO your findings tell us about relations between conflict, solidarity and hostility? I think you could argue that you investigate the time course of (engagement with) ingroup solidarity and outgroup hostility both pre-outbreak, during outbreak, and post-outbreak. I think you need to broaden your review and more clearly separate causes and consequences of conflict to build a relevant theoretical framework. The M&S quote is most relevant pre-conflict; the rally around the flag bits from previous versions (which are more applicable to the post-outbreak situation) were more useful for the during/post-outbreak stages.

2. In a related vein, you write that “there appears to be a consensus among scholars of social identity theory that ingroup-favoring motivations should matter more than outgroup-derogation”. This is a very limited reading of the literature. For instance, there is an extensive literature on dehumanization that shows that outgroup-derogation “matters” a lot. E.g., the atrocities in WW2-Germany are commonly understood as driven by (morality-based) outgroup derogation/dehumanization. The cited sources don’t support the argument either. For instance, although Weisel & Böhm (2015) indeed show that ingroup love is a more common cause of ‘discrimination’, except when the intergroup relation is based on morality, where outgroup hate is more potent. I think the Russia-Ukraine conflict easily qualifies as a morality-based conflict. ‘Common’ isn’t the same as important. I think it would be helpful to discuss conditions under which solidarity or derogation may surface.

3. Although conceptual clarity has improved, there is still ambiguity around key concepts.

3a. As noted before, ingroup solidarity and outgroup hostility are not emotions by any common conceptualization of emotions. Love and hate are. I think the labels used by a classifier should not stand in the way of accurate terminology.

3b. Engagement is construed in many different ways in the manuscript and the rebuttal. Already in the first two points of your rebuttal you construe the main DV, engagement, as both “identity-based emotional expressions” and “identity expressions”. Later, you also refer to a quote from Garcia and Rimé, who construe it (I think more appropriately) as “the social sharing of emotion”. Note that these are completely different things, e.g., an (identity-based) emotion expression (e.g., anger) is something different than an identity expression (e.g., that I am a social psychologist),

which is again different from socially sharing emotions (e.g., interacting about our shared anger to build a common worldview, invite social support, etc). Interestingly, on p. 7, you write "Despite outgroup hostility becoming much more commonplace on Ukrainian Facebook and Twitter after the invasion as compared to before (see SI Figure S10), people did not engage with it much" which nicely illustrates the difference. Although people were EXPRESSING more hostility, other people did not ENGAGE with it. Thus, engagement is distinct from expression of emotions. In your study, it is primarily the media outlets who are expressing the emotions, which makes it unclear what the users are doing. It would be helpful to either construe the DV in a clear and coherent way, or refrain from construing it at all and interpreting the findings in terms of different construals.

4. Two more specific points about your contribution statement:

- "less is known about the psychological processes that contribute to social identification around political issues on digital and social media" (p.1) although the statement is true, you don't fill the gap, i.e., you don't provide insight into social identification as an outcome. You could argue that your study correlates of social identity.
- "Bridging this gap is crucial for understanding the social and emotional processes driving intergroup conflict on social media and formulating interventions to address polarization." (p. 1)
(a) the conflict is on the ground, not on social media; (b) the war is not polarization, unless we take polarization to be a synonym for all types of conflict; (c) I have difficulty seeing how your findings may help address polarization

I would really like to be more supportive of this work. But as a social psychologist, I am worried how a publication in this outlet may introduce enduring misunderstandings about important theories from my field. I hope you will see the constructive intent of my comments.

Reviewer #2 (Remarks to the Author):

I read the revised manuscript and I am happy with the edits. The authors have done a better job at contextualizing their findings and they did about as good of a job as I think they can in addressing R3's comments about social identity theory and IET. I do not have further comments - thanks for an interesting study!

- Thank you for all your comments and help with this study!

Reviewer #3 (Remarks to the Author):

Review of NCOMMS-23-01909C

First of all, I appreciate your thoughtful engagement with my concerns. Although I accept that some of the issues that I've raised are perhaps a matter of interpretation, I'm afraid other issues remain unresolved, most importantly that I continue to find the conceptualization and terminology imprecise. However, I feel it would be unreasonable to continue repeating these concerns (some of which may be impossible to resolve, in particular the fundamental concern that motives cannot be inferred from behavior alone, see #4 below), and expect that it will yield a different outcome.

Thus, realizing that this is the third revision, that I am the only reviewer still having concerns about the manuscript, that I do find the data themselves interesting, and that the data clearly come with a 'best before'-date, my intention is to not extend the review process too much. I would therefore like to suggest only some specifications and explications of the most contentious conceptual issues, so that the reader can better evaluate the connection between your findings and the literature, and be more aware of potential limitations of your interpretation.

- Thank you so much for all of your comments and concerns. We really appreciate your thoughtful engagement with our manuscript and comments. We have addressed each of your final points below in the main text.

1. I appreciate that you've reached out to M&S to provide more context to their quoted hypothesis that "in-group positivity drives conflict processes to a greater extent than out-group negativity". I accept that in hindsight, my interpretation of the quote was too narrow. I stand corrected: the quote applies to the victim's perspective as well, it applies broadly to all conflict-related processes, and the processes are likely bidirectional. However, it remains unclear to me how the quote connects to the current RQ. The quote deals with Emotions (X) and Conflict-related processes (Y). In the interest of clear conceptualization, it would be helpful if you could be more explicit about which of the

following three interpretations you have in mind. Note that each of these lead to follow-up questions.

1a. $Y =$ outbreak/conflict-related events, causing $X =$ media expressions: While this is, theoretically, the most plausible relation (aside from the anthropomorphizing of media), the focus of your paper is on explaining engagement. Thus, if this is your interpretation, how does it connect to the DV?

1b. $Y =$ outbreak/conflict-related events, causing $X =$ social media users' experiences, as 'measured' by engagement: In this case, I believe you need the assumption (currently implicit) that social media users 'agree' (engage) with content that is emotionally similar to their own (group-based) emotions. It would be helpful to make that assumption explicit, or better, make a case for why you believe this to be the case. Caveat: I believe this assumption is problematic (see #3 and #4 below), and that this interpretation is difficult to reconcile with the baserate findings, which show that ingroup solidarity expressions and outgroup hostility expressions increase similarly post-outbreak (in particular supplemental Fig 2.10, panel C2).

1c. $X =$ media expressions, $Y =$ engagement: In this case, I think it would be helpful to explicate that engagement is considered a "conflict-related process", and explain why engagement can elicit/sustain/resolve/impact (or in another way relate to) conflict.

1d. There might be another interpretation that I'm overlooking. In that case, I think it would be particularly helpful to clarify your thinking.

- Thank you so much for constructively engaging and recognizing there's room for diverse views on this and explicitly stating your interpretations. We have removed the direct quote, and, more generally, we would like to refrain from making strong causal claims given the correlational nature of our data. Nevertheless, we completely agree with the need to make our ideas as explicit as possible. Our interpretation is that patterns of engagement with ingroup solidarity or outgroup hostility covary with important conflict-related events. As per the reviewers' request, we are now more specific about what this means.
- For example, we have now defined engagement specifically in the manuscript (p1):
 - "Social media platforms run algorithms that utilize user signals such as shares, likes, and other reactions, known as engagement metrics, to show users content that maximizes these metrics⁸. The interplay between algorithmic content recommendation and human behavior makes social media a complex system with emergent properties^{8,9}. Although social media platforms limit access to their recommender algorithms,

researchers can investigate what characteristics of human behavior (e.g., the content users post on a platform) tend to generate more engagement.”

- We later say that “we investigate social media engagement (operationalized as the sum of all platform-specific reactions)” on page 2.
- We also explain in more detail what we aimed to investigate explicitly on page 2 (please also see our reply to point #4 below):
 - “In our work, we study whether social media posts gain more engagement if they express ingroup solidarity and outgroup hostility after the 2022 full-scale invasion of Ukraine to explore the broader theoretical prediction that ingroup solidarity can be associated with more engagement than outgroup hostility^{25, 27, 34} and to advance research on social media engagement predictors in non-Western contexts more generally.”
- We also explicitly state our interpretation with respect to IET, M&S, and the broader literature (e.g., Brewer, 1999; Greenwald & Pettigrew, 2014) on page 5:
 - With regard to our work, users might experience intergroup emotions through emotional contagion by being exposed to posts that express emotions (e.g.,⁵⁸), which increases their willingness to perform group affiliative behaviors (e.g., engaging with such posts) and group identification^{34, 59}. In the context of an intergroup conflict, ingroup solidarity might increase the sense of group identity and willingness to perform affiliative or group-beneficial behaviors more than outgroup hostility, resulting in higher engagement rates with ingroup posts.

2. When applying intergroup emotion theory to the current RQ, you’re taking ideas that are developed to explain (causes/consequences of) emotional experiences to explain consequences of emotional expressions (at least, if your interpretation is either 1b or 1c). These not only deal with different phenomena (experience vs. expression), but also a different causal order (i.e., experiences as a consequence of group-related concerns vs. others’ emotional expressions as cues triggering engagement), and a generalization from one level of analysis to another (internal emotional processes versus emotional processes between ‘people’). Given how little work there is on social effects of emotional expressions in inter-group relations (for rare exceptions, see e.g., De Vos et al., 2013; Livingstone et al., 2011; Shore et al., 2019; and work by Eran Halperin –which together show that social effects of emotion expressions are a topic in their own right, and require different explanations), I understand why IET is a useful theory here, but I think it would be helpful to be more explicit that you’re making such generalizations.

- Thank you so much for this comment. It is indeed helpful to state our generalization explicitly here. Briefly, our interpretation of our correlational results with respect to IET is that:

A [users read news media / other's posts that express emotions] → B [users experience intergroup emotions] → C [users behave in more group-affiliated ways (e.g., engage with posts) as described by IET].

- We are basing A→B on emotional contagion literature and the Facebook emotional experiment where Facebook causally manipulated the amount of emotional content presented to users and found that users in the reduced positive emotions condition indeed produced less positive emotional content and more negative emotional content and the reverse held for users in the reduced negative emotions condition (<https://www.pnas.org/doi/full/10.1073/pnas.1320040111>; see more e.g. <https://journals.plos.org/plosone/article?id=10.1371/journal.pone.0142390>).
- We are taking B→C from the IET claim that intergroup emotions can differentially influence intergroup behavior such as by increasing affiliative behavior like displaying symbols of group identity (i.e., user emotions regarding their group can influence engagement) as well as Bar-Tal et al (p1; <https://spssi.onlinelibrary.wiley.com/doi/pdf/10.1111/j.1540-4560.2007.00518.x>), “collective emotions play a pivotal role in shaping individual and societal responses to conflicting events”^j.
- We also take from M&S and the broader literature (e.g., Brewer, 1999; Greenwald & Pettigrew, 2014) the interpretation that ingroup solidarity should increase the sense of group identity and affiliative or group-beneficial behavior more (the quote that “in-group positivity drives conflict processes to a greater extent than out-group negativity”), leading to the inference that users should engage with ingroup solidarity more with [engagement with ingroup solidarity content] being the group-affiliative behavior.
- We would like to highlight again that our data is correlational and we are not making any causal claims, merely describing the empirical correlational results we found and providing one possible interpretation with respect to IET and the broader social identity literature. Further experimental research is needed to establish causality and rule out third-variable effects. We explicitly mention our reasoning on page 5:
 - “With respect to our work, users might experience intergroup emotions through emotional contagion by being exposed to posts that express emotions (e.g.,⁵⁸), which increases their willingness to perform group affiliative behaviors (e.g., engaging with such posts) as well as group identification^{34, 59}. In an intergroup conflict context, ingroup solidarity might increase the sense of group identity and willingness to perform affiliative or group-beneficial behaviors more than outgroup hostility, resulting in higher engagement rates.”

- Please see our reply to your point 4 for a more detailed discussion of our assumptions.

3. You make claims about ‘importance’ and what ‘matters’. Although the data clearly show that posts containing ‘ingroup solidarity’ (vs. outgroup hostility) attract more engagement post-outbreak, I still feel claims about ‘importance’ are problematic. Do you mean statistically important? Important in terms of real-world consequences? Important in terms of attracting readership? And for whom is it important, for media, for social media users, for the world? E.g., one way of reading this statement is that you take engagement a measure of ‘success’ (impact), and thus that “it is more important FOR NEWS OUTLETS to signal solidarity than negativity about outgroups, because this generates more engagement with their publications”. Is this what you have in mind? To avoid misunderstandings, it would be helpful to be more specific where you make judgements of “importance” based on the findings.

- Thank you, we agree that we need to be more explicit about what we mean by importance. In the manuscript, by “importance” we simply meant how explanatory a variable is, i.e., the effect size. We agree that this is confusing and unnecessary, so we have removed all mentions of importance from the paper and replaced them with the specific meaning within that context.
- Nevertheless, we think that the findings are important for social media users, news outlets and the social media platforms themselves, for a variety of reasons. The fact that ingroup solidarity gains engagement, more so than outgroup hostility, during conflict is important for social media users because it means that dominant posts on the platforms (those that travel more widely) are expressing ingroup solidarity and not outgroup hostility, which may contribute to a more positive atmosphere (e.g., see Ferrara & Yang 2015 on positive-emotional contagion: <https://journals.plos.org/plosone/article?id=10.1371/journal.pone.0142390>). It also matters for news outlets, as you mentioned, to know which posts to write to get more engagement. Moreover, it is important for social media platforms as a way to gauge if their algorithms are promoting unnecessarily hostile content. We have now made this clear in the manuscript on page 6:
 - Overall, the fact that ingroup solidarity gains engagement during conflict, more so than outgroup hostility, may be important for social media users as it may contribute to a more positive atmosphere^{58, 66} and for social media content creators to write posts that get traction. Moreover, it is important for social media platforms as a way to gauge if their algorithms are promoting unnecessarily hostile content.

4. Finally, given that this review process will be published alongside the paper, I would like to reiterate my concern (see first and second review round) about the fundamental limitations of inferring psychological states from observed behavior alone. I am not reiterating this concern in the expectation that it will lead to changes in the manuscript; this is mainly intended for readers who might consult this review alongside the paper.

I believe that 'engaging' with a media post expressing ingroup solidarity can reflect many things, including a state of heightened ingroup solidarity (i.e., an expression of a current state), a state of low ingroup solidarity (i.e., an expression of a desired state), an attempt to boost ingroup solidarity (i.e., motivated behavior), or no change in these experiences/motives at all, but instead something unrelated (e.g., a shift in the balance in online versus offline behavior, where conflict makes 'hostile' expressions more exclusively reserved for offline exchanges and only 'solidarity' expressions find their way into the online world). I certainly do not think it is a direct read-out of the experience of the particular emotion contained in the post that a social media user is engaging with. Jensen (2017) has published a great critique on inferring psychological 'happiness' from the use of happy words on social media, including comments on validity, representativeness, and the state-behavior connection that are highly related to the concerns that I've raised in this review process (although it isn't very detailed on the important distinction between emotional experiences and emotional expressions, which have no 1-to-1 relation, see e.g., work on display rules),

<https://doi.org/10.1371/journal.pone.0180080>

- Thank you so much for this detailed explanation. We have paraphrased your comments and included them in the discussion section of the manuscript on page 6 to bring the readers' attention to these potential concerns.
 - "Our interpretation of our work with respect to psychological literature relies on the assumptions that (1) social media users experience emotions after reading relevant emotional posts (i.e., emotional contagion) and that (2) they behave (i.e., engage with the posts) because of their emotional experiences. Although these assumptions are somewhat supported by previous studies (see ref.⁵⁸ for (1) and ref.¹⁵ for (2)), there are alternative explanations for why a user would engage with a post. In fact, a user liking an ingroup solidarity post may reflect a range of emotional states, including the user's current state, the user's desired state, an attempt to boost morale through motivated behavior, a change in online versus offline behavioral patterns, or it could be unrelated to the post's emotional content. Future experimental research in controlled laboratory and field settings is necessary to validate and clarify our findings further."